# Constructing Industrial-Scale Optimization Modeling Benchmark

Zhong Li [* 1]  Hongliang Lu [* 2]  Tao Wei [* 2]  Yuxuan Chen [* 2]
Wenyu Liu [2]  Yuan Lan [3]  Fan Zhang [3]  Zaiwen Wen [2]

## Abstract

Optimization modeling underpins decision-making in logistics, manufacturing, energy, and finance, yet translating natural-language requirements into correct optimization formulations and solver-executable code remains labor-intensive. Although large language models (LLMs) have been explored for this task, evaluation is still dominated by toy-sized or synthetic benchmarks, masking the difficulty of industrial problems with $10^3$–$10^6$ (or more) variables and constraints. A key bottleneck is the lack of benchmarks that align natural-language specifications with reference formulations/solver code grounded in real optimization models. To fill in this gap, we introduce MIPLIB-NL, built via a structure-aware reverse construction methodology from real mixed-integer linear programs in MIPLIB 2017. Our pipeline (i) recovers compact, reusable model structure from flat solver formulations, (ii) reverse-generates natural-language specifications explicitly tied to this recovered structure under a unified model–data separation format, and (iii) performs iterative semantic validation through expert review and human–LLM interaction with independent reconstruction checks. This yields 223 one-to-one reconstructions that preserve the mathematical content of the original instances while enabling realistic natural-language-to-optimization evaluation. Experiments show substantial performance degradation on MIPLIB-NL for systems that perform strongly on existing benchmarks, exposing failure modes invisible at toy scale.

*Equal contribution  [1]Great Bay University [2]Peking University [3]Huawei Technologies Co., Ltd. Correspondence to: Zaiwen Wen <wenzw@pku.edu.cn>.

*Proceedings of the $43^{rd}$ International Conference on Machine Learning*, Seoul, South Korea. PMLR 306, 2026. Copyright 2026 by the author(s).

## 1. Introduction

Optimization models play a central role in modern decision-making across logistics, manufacturing, energy systems, finance, and machine learning (Singh, 2012; Antoniou & Lu, 2007). Despite their impact, translating a natural-language (NL) problem description into a correct and executable optimization model remains a highly specialized task, requiring substantial domain expertise and modeling experience (Huang et al., 2025a; Jiang et al., 2025). Recent advances in large language models (LLMs) have therefore sparked growing interest in *natural-language-to-optimization* (NL-to-Opt) modeling, where optimization models are constructed directly from natural-language specifications. Such systems could significantly lower the barrier to applying optimization techniques in practice (Xiao et al., 2025).

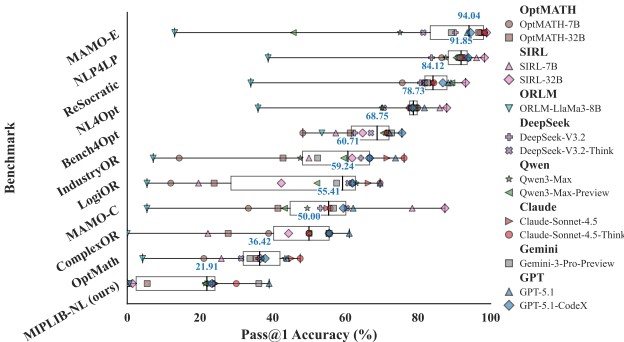

*Figure 1.* Accuracy distributions of state-of-the-art LLM-based methods for optimization modeling across existing benchmarks and MIPLIB-NL. Each boxplot shows the distribution of accuracies across methods for a given benchmark. Individual markers represent method-level performance and blue annotations indicate the median accuracy. MIPLIB-NL remains substantially more challenging, with generally lower performance across methods.

Therefore, a growing body of work has proposed prompting strategies, agent-based frameworks, and supervised fine-tuning approaches for LLM-based optimization modeling (Ramamonjison et al., 2022; Jiang et al., 2025; AhmadiTeshnizi et al., 2024; Xiao et al., 2024; Huang et al., 2025a; Lu et al., 2025; Liu et al., 2026). However, progress in this area has been evaluated almost exclusively on a small collection of existing benchmarks. Figure 2 reveals that these benchmarks are predominantly toy-sized or synthetic (either

by Human or LLMs), often involving only 4-120 variables and constraints with limited structural complexity. As a result, performance on many existing datasets has largely saturated as shown in Figure 1, creating the impression that NL-to-Opt modeling is close to being solved.

This evaluation regime stands in sharp contrast to real-world optimization practice, where industrial optimization models are rarely small or flat (Bi et al., 2024). Instead, they are typically defined through collections of variable groups (indexed) and constraint families (compositional)—such as flow conservation over network nodes, capacity constraints over resources, or assignment constraints over entity pairs—resulting in models with thousands to hundreds of thousands of variables and constraints. Such structural and scale-related challenges are largely absent from current benchmarks, leaving the behavior of LLM-based systems on realistic optimization models poorly understood. We argue that this gap is fundamentally *data-centric*. Rather than reflecting limitations of model architectures alone, it stems from the absence of a principled methodology for constructing NL-to-Opt benchmarks grounded in real industrial optimization models. Most existing datasets are created via forward generation from simplified templates or manually authored word problems, which flatten or omit the indexed, compositional structure that characterizes practical models. Without datasets that faithfully reflect industrial modeling practice, it is difficult to assess whether LLMs can genuinely perform optimization modeling beyond toy settings.

To address this limitation, we introduce MIPLIB-NL, a new benchmark constructed by *structure-aware reverse generation* from MIPLIB 2017 (Gleixner et al., 2021). MIPLIB 2017 is a widely used benchmark repository in the optimization community, consisting of real mixed-integer linear programs (MILP) submitted by practitioners and curated over decades for solver evaluation. Unlike synthetic benchmarks, MIPLIB instances embody the scale, structure, and modeling patterns encountered in industrial applications, but are distributed only as symbolic solver-level formulations without natural-language descriptions.

Starting from these solver-level formulations, we develop a reverse construction methodology that recovers their latent structural organization—variable groups and constraint families—and uses this structure to generate natural-language problem descriptions and structured data specifications. This process yields MIPLIB-NL, consisting of 223 carefully selected one-to-one reconstructions that faithfully preserve the mathematical structure and numerical properties of their corresponding MIPLIB 2017 instances. Across this benchmark, instances adopt a unified model–data separation format (see Appendix B for details), enabling natural-language descriptions to remain compact even for industrial-scale models (see Figure 10 for evidence).

Using MIPLIB-NL, we evaluate a range of state-of-the-art NL-to-Opt systems, including supervised fine-tuned models, and direct prompting with recent general-purpose LLMs. While these systems often achieve high accuracy on existing toy benchmarks, their performance degrades substantially on our industrial-scale instances (Figure 1). This discrepancy indicates that prior evaluations largely overestimate current optimization modeling capabilities and highlights the need for benchmarks grounded in real industrial models.

Our contributions are fourfold. (i) We identify *dataset design*, rather than model architecture alone, as a key bottleneck in NL-to-Opt evaluation, showing that existing benchmarks underestimate the challenges posed by large-scale industrial structure (Section 2.2). (ii) We propose a structure-aware reverse construction methodology that derives natural-language optimization instances directly from real industrial MILP models (Sections 3.1–3.4). (iii) We release MIPLIB-NL, the first NL-to-Opt benchmark fully grounded in MIPLIB 2017, emphasizing industrial realism beyond toy or synthetic formulations (Section 3.5). (iv) Using MIPLIB-NL, we present a comprehensive evaluation on state-of-the-art LLM-based optimization modeling methods, revealing systematic performance degradation and failure modes that are hidden by existing benchmarks (Section 4). Together, these contributions establish a principled foundation for advancing NL-to-Opt evaluations beyond toy benchmarks.

## 2. Preliminary

### 2.1. LLMs for Optimization Modeling

Recent advances in LLMs have motivated growing interest in NL-to-Opt modeling, where a verbal problem specification is translated into a formal mathematical optimization model and executable solver code (Xiao et al., 2025). A range of systems have been proposed, including NL4Opt (Ramamonjison et al., 2022), LLMOPT (Jiang et al., 2025), OptiMUS (AhmadiTeshnizi et al., 2024), Chain-of-Experts (CoE) (Xiao et al., 2024), ORLM (Huang et al., 2025a), OptMath (Lu et al., 2025), SIRL (Chen et al., 2025), etc. These approaches span prompt-based, agentic, and fine-tuned frameworks, and demonstrate that LLMs can generate correct optimization models for small or moderately structured problems. However, reported progress is tightly coupled to the datasets used for evaluation. As we argue in this work, strong performance on existing benchmarks does not necessarily indicate robust modeling capability on realistic industrial optimization problems, motivating an examination of dataset design.

### 2.2. Existing NL-to-Optimization Benchmarks

The development of NL-to-Opt systems has been closely tied to the availability of benchmark datasets, which are

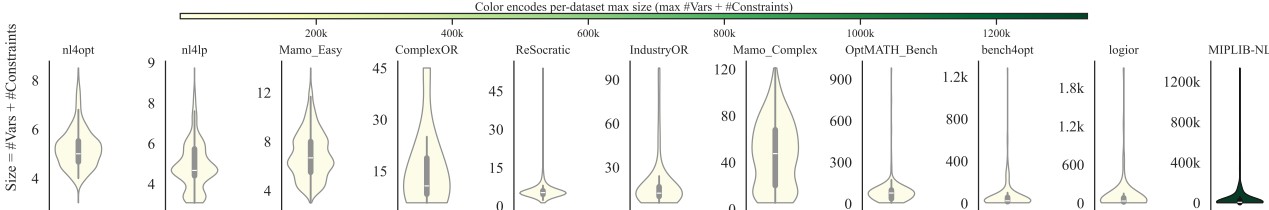

*Figure 2.* Distributions of problem sizes across optimization benchmarks, measured as the total number of variables plus constraints per instance. Each violin represents the per-instance size distribution of a dataset. A shared color scale encodes dataset scale, where warmer colors indicate larger maximum instance sizes. Compared to prior benchmarks, MIPLIB-NL (ours) spans a markedly larger scale and exhibits a substantially heavier upper tail, reflecting the presence of many large-scale industrial instances. While the violin shape is compressed near the lower range due to the extreme scale variation, a bucketed analysis in Fig. 5 further reveals that a significant fraction of MIPLIB-NL instances lie well beyond the scale of existing benchmarks.

summarized in Table 2 in Appendix A. These datasets include NL4Opt (Ramamonjison et al., 2022), IndustryOR (Huang et al., 2025a), ComplexOR (Xiao et al., 2024), MAMO (Huang et al., 2025b), NLP4LP (Ahmadi­Teshnizi et al., 2024), OptiBench (Wang et al., 2024), Re­Socratic (Yang et al., 2025b), OptMath (Lu et al., 2025), Lo­giOR (Yang et al., 2025a), Bench4Opt (Wang et al., 2025), OptiTrust (Lima et al., 2025), and Step-Opt (Wu et al., 2025), etc. Recent work has also extended optimization-modeling benchmarks to multimodal settings, as in MM-OptBench, where problem specifications combine text with visual arti­facts (Li et al., 2026b). While these datasets have played an important role in early progress, recent surveys and follow-up analyses (Xiao et al., 2025; Yang et al., 2025b) reveal several systematic limitations that constrain realistic evalua­tion.

**Limited scale.** As shown in Figure 2, most existing bench­marks are dominated by toy-sized instances, typically in­volving only tens of decision variables and constraints. Even datasets described as "large-scale" rarely exceed a few hun­dred variables (Liang et al., 2026), whereas real industrial optimization models routinely involve $10^3$–$10^6$ variables and constraints or more (Gleixner et al., 2021).

**Structural fidelity.** Many benchmarks rely on simplified or synthetic formulations that flatten constraint families into surface-level descriptions. This under-represents the com­positional, loop-based structures (Table 6) that characterize industrial optimization models in domains such as energy systems, production planning, and network design. As a re­sult, systems can achieve high benchmark accuracy without demonstrating genuine structural modeling competence.

**Verifiability and noise.** Several widely used datasets treat manually authored formulations or solver outputs as ground truth with limited auditing. Subsequent studies (Xiao et al., 2025; Yang et al., 2025b) report substantial annotation noise, including infeasible models, missing constraints, and incor­rect objective directions, obscuring true correctness and inflating reported performance (see Figure 11 for details).

This reliability concern is sharpened by OptArgus, which shows that matching a reference objective value alone may miss structural hallucinations and benchmarks consistency auditing across problem descriptions, symbolic models, and solver implementations (Li et al., 2026a).

These limitations together have led to an evaluation regime in which modern LLMs appear highly capable on existing benchmarks, while their behavior on realistic, industrial-scale optimization models remains largely unexplored.

### 2.3. Problem Setting and a Data-Centric Perspective

In this work, we study the task of NL-to-Opt modeling. For­mally, a problem instance is specified by a natural-language description $\mathbf{x}^{\mathrm{NL}} \in \mathcal{X}_{\mathrm{NL}}$ that describes an optimization task and presents relevant data either as part of the descrip­tion itself or as separate structured inputs (e.g., parameter tables or external files). The goal of NL-to-Opt model­ing is to generate (i) a mathematical optimization model and (ii) a solver-executable implementation, together de­noted by $\mathbf{x}^{\mathrm{Opt}} \in \mathcal{X}_{\mathrm{Opt}}$, such that the resulting model is *semantically consistent* with the original specification $\mathbf{x}^{\mathrm{NL}}$. Equivalently, NL-to-Opt modeling aims to learn a mapping $f : \mathcal{X}_{\mathrm{NL}} \rightarrow \mathcal{X}_{\mathrm{Opt}}$, where semantic consistency requires that $\mathbf{x}^{\mathrm{Opt}}$ defines a mathematical model that is semantically equivalent to the one described in natural language.

Our reverse construction (i.e., Opt-to-NL) is motivated by MIPLIB 2017 (Gleixner et al., 2021), an authoritative and widely used repository of mixed-integer linear programs in the optimization community. Crucially, MIPLIB repre­sents decades of accumulated modeling effort, curated and validated by the optimization community, and consists of instances derived from real industrial applications and val­idated by state-of-the-art solvers. However, these models are distributed exclusively in algebraic formats (e.g., `.mps`) and lack natural-language specifications, making them un­suitable for direct NL-to-Opt evaluation.

Therefore, rather than collecting natural-language specifi­cations directly, we adopt a *data-centric construction per-*

*spective* based on the reverse direction, namely Opt-to-NL generation, which allows NL-to-Opt benchmarks to directly benefit from the scale, realism, and long-term curation of MIPLIB. Specifically, we begin with a set of optimization instances in MIPLIB 2017 (denoted as $\mathcal{X}_{\text{Opt}}^{\text{real}} = \{ \mathbf{x}_i^{\text{Opt}} \}$,) and construct corresponding natural-language specifications via a controlled reverse mapping

$$g : \ \mathcal{X}_{\text{Opt}}^{\text{real}} \ \rightarrow \ \mathcal{X}_{\text{NL}},$$

yielding paired instances $\left( \mathbf{x}_i^{\text{NL}}, \mathbf{x}_i^{\text{Opt}} \right)$ that are suitable for NL-to-Opt evaluation.

By starting from real optimization models and reverse-generating natural-language descriptions, we adopt an Opt-to-NL perspective that grounds benchmark construction in authoritative industrial formulations. This perspective enables the construction of benchmarks that are large-scale, structurally faithful, and closely aligned with real optimization practice, and directly motivates the structure-aware reverse construction pipeline introduced in the next section.

# 3. Constructing Industrial-Scale Benchmark via Structure-Aware Reverse Generation

This section describes how we construct MIPLIB-NL by reverse-generating natural-language (NL) descriptions and structured data specifications from mathematical optimization models in MIPLIB 2017 (provided in *.mps* format). Our objective is to build a large-scale benchmark that faithfully reflects the structure, scale, and modeling practices of real industrial optimization problems, while remaining suitable for rigorous evaluation of NL-to-Opt systems.

## 3.1. Overview of the Construction Pipeline

Figure 3 summarizes our structure-aware reverse construction pipeline. Starting from MIPLIB 2017 models in `.mps` format, we systematically transform each industrial optimization model into a NL-to-Opt benchmark instance through the following stages. **(1) Structural abstraction**: We recover indexed variable groups and repeated constraint families from flat MPS formulations, yielding a compact loop-based structural *scaffold* (Section 3.2); **(2) Structure-aware Opt-to-NL generation**: Using this scaffold as a fixed semantic backbone, experts design deterministic NL blueprints that translate variables, objectives, and constraint families into faithful NL descriptions, with optional non-semantic linguistic polishing (Section 3.3); **(3) Semantic validation**: We validate semantic sufficiency via independent NL-to-Opt reconstruction and expert-guided review, ensuring that each instance fully specifies the intended optimization model (Section 3.4). All stages operate on a unified instance representation with strict model–data separation, enabling scalable construction, validation, and reproducible

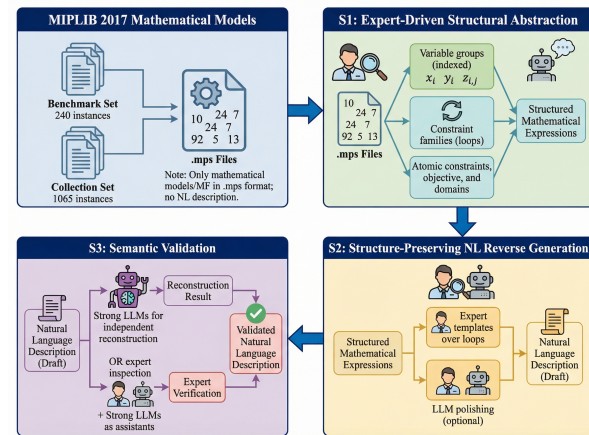

*Figure 3.* Structure-aware reverse construction pipeline: starting from MIPLIB 2017 math models, we perform (1) expert-driven structural abstraction of (often large) MPS formulations, (2) structure-preserving Opt-to-NL reverse generation via expert-designed blueprints, and (3) semantic validation via independent NL-to-Opt reconstruction with human–LLM interaction.

evaluation. Details of the schema and artifact layout to store generated instances are described in Appendix B.

## 3.2. Structural Abstraction from MPS Math Models

**From atomic constraints to constraint families.** Prior work (Ramamonjison et al., 2022; Li et al., 2023) on NL-to-Opt has primarily characterized optimization models through *atomic constraint types* (also referred to as semantic constraint types), i.e., semantic categories that describe the algebraic form of individual constraints, such as bounds, linear sums, capacity limits, ratios, and simple logical relations. These atomic types provide a useful vocabulary for mapping NL statements to algebraic expressions and are effective for small or textbook-scale problem instances. Table 5 in Appendix D.1 summarizes the commonly used atomic constraint types adopted in prior work.

However, atomic constraint types alone substantially underestimate the complexity of real industrial optimization models. In practice, constraints rarely appear as isolated algebraic expressions (Bi et al., 2024). Instead, they arise as large *constraint families*: repeated instantiations of the same semantic rule over one or more index domains, such as products, locations, time periods, commodities, or graph structures. The same atomic constraint type (e.g., a capacity or summation constraint) may therefore induce radically different modeling structures depending on how it is indexed, aggregated, or coupled across indices.

**Loop-based structure as the source of scale and complexity.** Constraint families are generated through *loop-based modeling constructs* that specify how atomic constraint templates are instantiated over index sets. These loop structures

determine not only the number of constraints that appear at the solver level, but also how different constraint families interact, couple across dimensions (e.g., jointly indexed by products, locations, and time), or evolve over time. When expanded into MPS form, such loop-based structures may result in thousands or even millions of algebraic rows, obscuring the underlying generative logic of the model.

As a result, recovering meaningful modeling structure from an MPS file is a *model interpretation* problem rather than a purely syntactic parsing task. Although the MPS format preserves algebraic correctness, it discards the loop-based structure through which constraints were originally generated, as well as the semantic roles of variables and indices. Identifying this structure therefore requires domain knowledge about how industrial MILPs are typically formulated.

**Stage 1: Expert-driven loop abstraction.** Figure 4 summarizes the process of expert-driven loop abstraction: experts triangulate *on-model signals* from the MPS (e.g., naming regularities and incidence structure) with *off-model evidence* (e.g., limited MIPLIB metadata) to form and iteratively refine a structural hypothesis, and then validate it against the expanded algebraic rows. Our approach centers on recovering a *loop-based structural scaffold* that explicitly represents how constraint families are generated from atomic modeling rules. For each MIPLIB instance, OR experts analyze the formulation to identify: (i) indexed variable groups; (ii) constraints families corresponding to repeated instantiations of the same algebraic relation; and (iii) a small number of non-repeating (atomic) constraints. Rather than treating individual algebraic rows as primitive objects, we group constraints that share the same algebraic form and modeling role into a loop-based structure template. This abstraction yields a compact structural representation that preserves the essential modeling logic while remaining independent of problem scale. It enables us to (i) express large industrial models concisely in NL without enumerating constraints, and (ii) ensure consistent alignment between NL, mathematical formulations, and solver code.

Although the notation used to describe a recovered loop may vary across experts, the underlying mathematical semantics are fixed by the original MPS rows and coefficients. We therefore treat loop abstraction as a compact representation of solver-level algebra, not as a free-form annotation. In practice, a drafted loop scaffold can be expanded back into algebraic rows and checked for exact equivalence against the original formulation, which reduces subjectivity to presentation choices rather than model semantics.

**Summary of observed loop-based structures.** Across MIPLIB-derived instances, we observe a rich set of recurring loop-based structural scaffolds—including nested loops, subset-indexed loops, temporal or recursive coupling, sliding-window aggregation, pairwise (complete-

graph) instantiation, and extra-dimension replication—that are largely absent from existing NL-to-Opt benchmarks. Canonical forms of these loop-based structural scaffolds, together with a systematic taxonomy (Table 6), are provided in Appendix D.2. Additional empirical observations (Appendix D.3), representative examples (Appendix D.6), and the augmented application taxonomy (Table 4) arising from this structural analysis are also provided in Appendix D.

### 3.3. Structure-Preserving NL Reverse Generation

**Stage 2: Opt-to-NL reverse generation.** Once the loop-based structural scaffold has been extracted and validated, the remaining task is to express this structure as a NL problem description suitable for NL-to-Opt evaluation. Crucially, at this stage the *mathematical structure is already fixed*. NL generation does not involve discovering new variables, constraints, or objectives, but translating an expert-validated scaffold into a faithful and readable problem narrative. This stage consists of the following three steps.

**(1) Application context identification.** To produce a faithful and readable NL description, each recovered scaffold is associated with an application scenario. In most cases, this is already determined during or immediately after Stage 1 by combining MIPLIB metadata, variable and constraint naming patterns, data sources, and the recovered loop structure. For a small minority of instances—many of which were explicitly described in MIPLIB as artificially generated—the original application domain is unclear; in these cases, we assign a natural scenario that is consistent with the recovered variables, objectives, and constraint families.

The criterion is therefore structural clarity and semantic recoverability, rather than whether the MIPLIB source is real or synthetic. If a real-world-tagged instance is heavily obfuscated or lacks recoverable business logic, we exclude it instead of forcing a generic application template; conversely, a synthetic instance with a clear combinatorial structure may receive a faithful natural scenario.

**(2) Expert-designed NL blueprints.** For each application scenario, OR experts further design NL blueprints that map: (a) indexed variable groups to semantic roles (e.g., production quantities, flow decisions, facility-opening indicators); (b) the objective to optimization intent and direction; (c) each constraint family to a loop-based NL clause (e.g., "for each facility and time period"); and (d) data dependencies to explicit file and parameter references. These blueprints are constructed directly from the extracted scaffold and the associated application-level interpretation obtained in Stage 1. Therefore, the generated *raw NL* description is structurally complete: every variable group and constraint family in the scaffold has a corresponding linguistic realization, and no modeling decisions are left implicit.

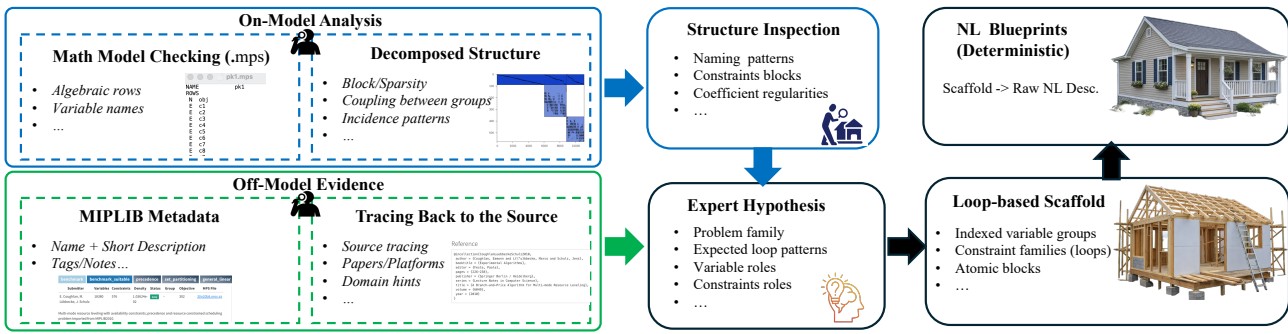

*Figure 4.* Expert-driven structural understanding and loop scaffold construction. To recover high-level modeling logic from industrial MPS files (Stage 1), we adopt an expert-in-the-loop process. OR experts combine *on-model analysis* (variable naming patterns, constraint blocks, coefficient regularities, sparsity and incidence structure) with *off-model evidence* (e.g., MIPLIB metadata, source papers, and domain hints if available) to form hypotheses about problem families, variable roles, and constraint semantics. These hypotheses are refined through inspection of decomposed model structure and validated against algebraic rows in the MPS file, yielding indexed variable groups, constraint families (loops), and atomic blocks. The resulting loop-based scaffold serves as the foundation for deterministic NL blueprint construction and structure-preserving Opt-to-NL generation (Stage 2).

**(3) LLM-assisted linguistic polishing.** To improve readability and stylistic naturalness, we optionally apply LLM-based polishing to the raw NL under strict semantic constraints. The LLM is forbidden from introducing new variables or constraints, inventing coefficients, or modifying data references. Its role is limited to surface-level rephrasing (e.g., lexical choice and sentence flow), while semantics remain fixed by the scaffold and blueprint.

### 3.4. Semantic Validation

**Stage 3: semantic validation.** After structural abstraction (Stage 1) and Opt-to-NL reverse generation (Stage 2), the remaining question is whether the constructed instance—including the NL description, external data files, and parameter specifications—*faithfully and sufficiently encodes* the intended optimization problem derived from the original MPS model. Validation therefore focuses on semantic adequacy of the *entire instance specification*.

**Primary validation: independent NL-to-Opt reconstruction.** For each constructed instance, we first perform an *independent reconstruction* test using multiple state-of-the-art large language models (LLMs). Each model is provided only with the following files: (1) the `instance.json`—which specifies the NL problem description (abstract and optional concrete forms), instance-level parameters, and semantic descriptions of all external data sources, and (2) the referenced data files.

For each constructed instance, we evaluate reconstruction using multiple strong LLMs under a fixed sampling budget. For a given LLM, we generate up to 8 independent reconstruction attempts with different random seeds (i.e., Pass@8). Each attempt requires the model to (a) reconstruct a mathematical formulation of the optimization problem and (b) produce solver-executable code that reads the provided

data. An instance is considered *successfully reconstructed* by a given LLM if *at least one* of its 8 attempts produces a model whose solver outcome (e.g., optimal objective value or solver status, when available) matches that of the released instance under the same solver configuration. We adopt an *existential* validation criterion: success by a single LLM is sufficient evidence that the NL and data specification is semantically sufficient, while failure by all tested LLMs does not by itself invalidate the instance.

**Secondary validation: expert-driven human–LLM interaction.** For instances where none of the tested LLMs succeed in independent reconstruction, we do not conclude that the instance is incorrect. Instead, such cases enter a secondary validation phase centered on *expert review with human–LLM interaction*.

Our experts perform targeted manual inspection, focusing on: (a) alignment between the NL description and the extracted loop scaffold, (b) correctness and completeness of variable roles, objective intent, and constraint families, and (c) consistency between NL statements, external data files, and parameter values. Given the scale of industrial MPS models, this review is structural rather than exhaustive. Importantly, this process also serves to identify and correct potential errors introduced during the reverse construction stage. In cases where inconsistencies or omissions are traced back to flaws in the generated NL specification or data alignment, we revise the instance accordingly before inclusion in the benchmark. During this validation process, strong LLMs are used as interactive assistants rather than autonomous generators. Experts iteratively refine prompts, clarify modeling assumptions, and guide the LLMs toward a correct interpretation of the specification. In many cases (see Appendix E for representative examples), such expert-guided interaction enables LLMs to successfully reconstruct and solve the in-

stance; these instances are retained in the benchmark after validation or revision.

In our construction logs, 245 candidate instances entered the full validation pipeline. Among them, 163 passed the primary independent reconstruction stage. The remaining 82 cases entered secondary expert-driven validation; 60 were retained after expert–LLM review and revision, while 22 were discarded. This breakdown illustrates the complementary roles of automatic reconstruction checks and expert-guided semantic validation.

**Infeasible and open instances.** A small number of constructed instances correspond to MIPLIB models that are infeasible or remain open, for which no certified optimal objective value exists. For these instances, solver-outcome matching is not applicable. We retain such instances (8 infeasible and 15 open) in the benchmark after expert validation confirms that their NL descriptions, data specifications, and recovered structure are semantically complete and faithful to the original models. However, because standard quantitative metrics rely on objective-value comparison, these instances are excluded from the experimental evaluation (see Appendix E.3 for validation details).

### 3.5. The MIPLIB-NL Benchmark: Properties and Characteristics

Based on the structure-aware reverse construction procedure described above, we construct and release the benchmark MIPLIB-NL, consisting of 223 instances that are one-to-one reverse-generated from selected models in MIPLIB 2017. We emphasize that MIPLIB 2017 contains over 1,300 models in total; due to the substantial manual effort required for structure recovery and semantic validation, we focus on a representative subset of instances that can be reliably translated. The main sources of exclusion are summarized in Appendix I. Each instance is a *triply aligned specification*, consisting of (i) a natural-language problem description (with separately specified data files), (ii) a mathematical optimization model, and (iii) a solver-executable implementation. By construction, the decision variables, constraints, objective, and numerical parameters are preserved exactly (up to systematic renaming), ensuring that the optimal solution and solver behavior match those of the original MIPLIB formulation. As a result, MIPLIB-NL provides a faithful natural-language interface to canonical, solver-validated industrial optimization problems. The dataset is publicly released at https://github.com/optsuite/MIPLIB-NL.

**Problem scale and structural compression.** Figure 5 (top) shows the distribution of model size in MIPLIB-NL, measured by the total number of variables and constraints. While smaller instances are present, a substantial fraction lies in the large and very large regimes, yielding a heavy-tailed

scale distribution characteristic of industrial optimization models generated via loop-based instantiation. Figure 5 (bottom) reports the corresponding compression ratios between the flattened MPS representation and the structured natural-language specification with model–data separation. Across all scale buckets—and increasingly for larger instances—the natural-language and data representation remains markedly more compact, indicating that solver-level size inflation mainly arises from flattening structured, repetitive models into explicit MPS form. More details are given in Appendix C. Together, these properties fundamentally distinguish MIPLIB-NL from prior NL-to-Opt benchmarks: instead of emphasizing isolated semantic constraint types or small toy models, MIPLIB-NL foregrounds large-scale structure, indexing, and constraint interaction as first-class challenges, enabling more realistic and discriminative evaluation of NL-to-Opt systems and directly motivating the experimental analysis in the next section.

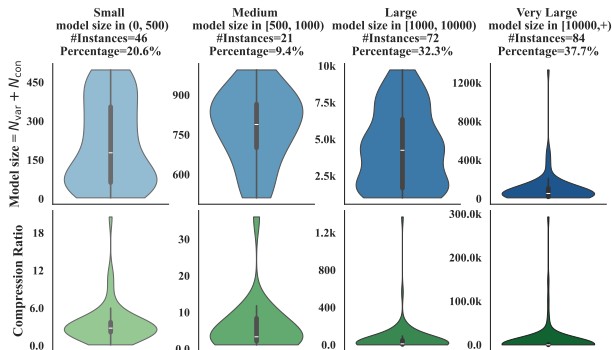

*Figure 5.* Distribution of model size and compression ratio across scale buckets on MIPLIB-NL. The top row shows the distribution of problem size, measured as the total number of variables and constraints. The bottom row reports the corresponding compression ratios, defined as $\mathrm{CR} = \frac{\text{MPS file size}}{\text{NL file size} + \text{auxiliary data file size}}$, which capture how much more compact the structured natural-language description with model–data separation is compared to the flattened, solver-ready MPS representation. Buckets are defined by model size, and annotations indicate the number and proportion of instances in each bucket. The results reveal that, even as instance scale grows substantially, the separated NL–data representation remains markedly more compact, highlighting the importance of scale-aware evaluation for large optimization problems.

## 4. Experiments

We conduct a set of experiments to demonstrate the utility of MIPLIB-NL as a realistic evaluation benchmark and to assess how existing NL-to-Opt systems perform when confronted with industrial-scale optimization problems.

### 4.1. Evaluation Overview

**Evaluation and Analysis Questions.** Our experiments separate benchmark-validation axes from a diagnostic analysis

*Table 1.* Pass@1 accuracy (%) of different NL-to-Opt systems and bare LLMs across datasets.

| | Method | NL4Opt | MAMO-E | MAMO-C | ComplexOR | IndustryOR | NLP4LP | ReSocratic | OptMath | Bench4Opt | LogiOR | MIPLIB-NL(Ours) |
|---|---|---|---|---|---|---|---|---|---|---|---|---|
| **Fine-Tuned Models** | OptMATH-Qwen2.5-7B | 79.91 | 97.06 | 33.33 | 38.89 | 14.29 | 86.52 | 75.68 | 21.08 | 48.30 | 11.96 | 0.00 |
| | OptMATH-Qwen2.5-32B | 79.44 | 96.70 | 41.44 | 27.78 | 42.86 | 92.13 | 81.89 | 35.54 | 61.36 | 23.91 | 5.50 |
| | SIRL-Qwen2.5-7B | 86.00 | 98.50 | 78.40 | 22.20 | 50.00 | 96.10 | 89.10 | 25.90 | 57.38 | 19.60 | 0.48 |
| | SIRL-Qwen2.5-32B | 87.90 | 98.90 | 87.40 | 44.40 | 61.90 | 98.30 | 93.10 | 36.70 | 64.77 | 42.40 | 1.43 |
| | ORLM-LlaMa3-8B | 77.10 | 88.81 | 42.34 | 0.00 | 40.48 | 83.71 | 68.73 | 5.42 | 53.54 | 13.04 | 0.58 |
| **Bare LLMs (Direct Prompting)** | DeepSeek-V3.2-685B | 77.57 | 90.28 | 53.15 | 55.56 | 64.29 | 83.71 | 80.65 | 36.14 | 62.55 | 61.96 | 23.81 |
| | DeepSeek-V3.2-Think-685B | 70.56 | 81.47 | 60.36 | 50.00 | 66.67 | 91.01 | 81.89 | 31.33 | 67.05 | 60.87 | 22.38 |
| | Qwen3-Max (No-Thinking) | 70.09 | 75.05 | 49.55 | 55.56 | 47.62 | 87.64 | 84.12 | 43.37 | 71.02 | 63.04 | 21.43 |
| | Qwen3-Max-Preview (Thinking) | 78.97 | 84.22 | 43.24 | 50.00 | 59.52 | 90.45 | 89.33 | 36.75 | 70.45 | 52.17 | 21.43 |
| | Claude-Sonnet-4.5 (No-Thinking) | 78.04 | 98.35 | 54.96 | 55.56 | 71.43 | 92.13 | 84.12 | 45.18 | 72.16 | 66.30 | 24.23 |
| | Claude-Sonnet-4.5 (Thinking) | 78.04 | 98.35 | 55.86 | 50.00 | 76.19 | 92.70 | 84.37 | 47.59 | 71.59 | 69.57 | 30.00 |
| | Gemini-3-Pro-Preview | 78.97 | 89.17 | 56.76 | 55.56 | 52.38 | 91.57 | 82.63 | 33.73 | 72.73 | 57.61 | 36.19 |
| | GPT-5.1 | 81.78 | 93.58 | 62.16 | 61.11 | 73.81 | 93.82 | 88.34 | 43.98 | 72.16 | 69.57 | 39.05 |
| | GPT-5.1-CodeX | 78.50 | 94.50 | 59.46 | 55.56 | 66.67 | 93.82 | 86.85 | 37.95 | 75.57 | 61.96 | 23.33 |
| | Avg | 78.78 | 91.78 | 55.60 | 44.44 | 56.29 | 90.97 | 83.63 | 34.33 | 65.76 | 48.14 | 17.85 |

question: **RQ1.** How do existing NL-to-Opt systems and bare LLMs perform on MIPLIB-NL compared with prior toy-scale benchmarks? **RQ2.** How does modeling performance vary with problem scale? and **RQ3.** What error modes dominate at large scale? RQ1 and RQ2 establish the capability picture under the new benchmark regime, while RQ3 analyzes the failure patterns exposed by that regime.

**Experimental Setup.** We evaluate a broad spectrum of existing NL-to-Opt approaches, covering two primary categories of methods: (i) fine-tuned models trained via supervised learning and/or reinforcement learning, and (ii) bare LLMs evaluated via direct prompting. For bare LLMs, we further distinguish between standard inference (*no-thinking*) and explicit reasoning-enabled variants (*thinking*), when available. In total, we evaluate 14 representative systems. All methods are tested on a diverse collection of benchmarks, including 10 widely used NL-to-Opt datasets as well as our newly introduced MIPLIB-NL. We report multiple complementary evaluation metrics to capture both modeling correctness and end-to-end usability, including Pass@$N$ accuracy ($N = 1$ and 8) and solver code executability. Complete experimental details—including dataset composition, baseline configurations, metric definitions, and diagnostic protocols—are deferred to Appendix F to save space.

### 4.2. Overall Performance across Datasets (RQ1)

Table 1 reports Pass@1 accuracy across a range of existing NL-to-Opt benchmarks and our proposed MIPLIB-NL. Rows correspond to representative NL-to-Opt systems, including fine-tuned models and bare LLMs evaluated via direct prompting, while columns correspond to different datasets. In addition to Pass@1, we also evaluate Pass@8 accuracy (Table 8), and separately report solver code executability statistics under single-sample (Table 9) evaluation setting; full results and analysis are provided in Appendix H.

**Main takeaway.** A consistent pattern emerges across all method classes. Systems that achieve high accuracy on existing toy-scale benchmarks often suffer dramatic performance drops on MIPLIB-NL, in many cases by tens

of percentage points. Notably, this degradation affects not only bare LLMs but also fine-tuned models. These results indicate that strong performance on existing benchmarks substantially overestimates current NL-to-Opt capabilities, and that MIPLIB-NL exposes modeling challenges that are largely invisible under prior evaluation regimes. Appendix H further includes a family-held-out LoRA check on Qwen2.5-7B-Instruct (Table 10). The result shows that difficult in-domain supervision can substantially improve a smaller model, especially in executability, but still leaves a large gap to frontier models. Thus, MIPLIB-NL is useful not only for evaluation but also for studying supervision and scaling effects in industrial-scale modeling.

### 4.3. Scale-Dependent Performance (RQ2)

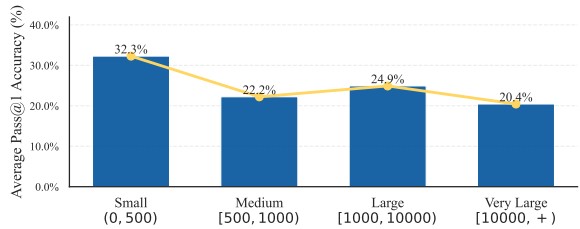

*Figure 6.* Performance across scale buckets on MIPLIB-NL, where scale (x-axis) is defined as the total number of variables and constraints. While overall accuracy declines as scale increases, the trend is not strictly monotonic, reflecting interactions between scale, problem structure, and domain composition.

To examine the effect of problem scale, we stratify MIPLIB-NL instances by model size, defined as the total number of variables and constraints, into four buckets: *small* ($< 500$), *medium* (500–999), *large* (1000–9999), and *very large* ($\geq 10000$). Figure 6 reports average Pass@1 accuracy across these groups. Overall performance declines as scale increases, but the trend is not strictly monotonic. Accuracy drops sharply from small to medium instances, shows partial recovery for large problems, and then decreases again for very large models. Despite this variability, large and very large instances remain more challenging than small

ones. These results indicate that increasing problem scale introduces structural and modeling difficulties that are absent from existing toy benchmarks, highlighting the need for evaluation on industrial-scale optimization problems.

### 4.4. Error Modes Revealed by MIPLIB-NL (RQ3)

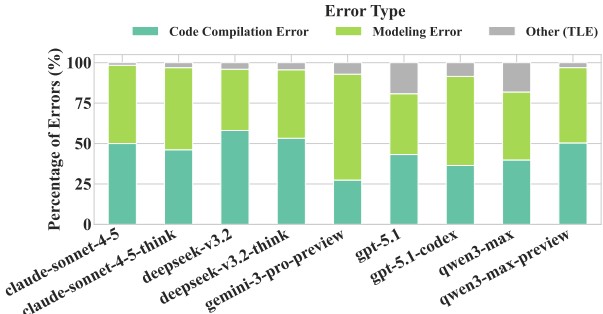

*Figure 7.* Error type distribution for each model on MIPLIB-NL, normalized over its failed instances. Modeling and execution-level errors dominate, with timeouts forming a minor residual.

Leveraging the unified schema and structural annotations provided by MIPLIB-NL, we categorize model failures into three broad classes: (i) *execution-level errors*, where generated solver code fails to run; (ii) *modeling-level errors*, such as incorrect variable typing, missing or mis-specified constraint families, or wrong objective direction; and (iii) *timeout errors*, where the solver fails to finish within the allotted time. As shown in Figure 7, failures are dominated by modeling-level errors (48.3%) and execution-level errors (45.0%), with a smaller share of timeouts (6.7%). The balance varies across model families: some exhibit a higher proportion of compilation/runtime failures, whereas others are more prone to modeling mistakes. Common modeling errors include omitted capacity or balance constraints, incomplete index sets in loop-based constraints, and incorrect coupling between decision variables and external data. These issues are less visible on toy benchmarks but become pervasive at scale, highlighting the value of MIPLIB-NL in exposing structurally grounded failure modes.

Beyond the coarse taxonomy in Figure 7, Appendix G further refines modeling and execution failures into subtype-level categories. Figure 18 provides the across-benchmark comparison: prior datasets are typically dominated by variable-domain/type or constraint-logic mistakes, whereas MIPLIB-NL is dominated by data/index coupling errors and incomplete-model failures. The pairwise distributional tests reported with Figure 18 are statistically significant, with the strongest contrast on modeling errors. Figure 19 provides a within-MIPLIB-NL relative-scale analysis: as MIPLIB-NL instances become larger relative to other MIPLIB-NL instances, execution failures shift from data-binding issues toward environment/resource bottlenecks, while modeling failures increasingly involve data/index coupling and incomplete model recovery. Taken together, Figures 7, 18, and 19 show that MIPLIB-NL does not merely make the task harder; it exposes a different and more structurally grounded failure regime.

Detailed case studies, including side-by-side comparisons of NL descriptions, generated formulations, and canonical models, are provided in Appendix G. Together, these analyses illustrate how MIPLIB-NL reveals failure modes that remain hidden under existing evaluation settings.

## 5. Conclusion

We introduced MIPLIB-NL, a large-scale NL-to-Opt benchmark constructed via structure-aware reverse generation from real industrial MILP instances in MIPLIB 2017. By recovering indexed variable groups and constraint families, MIPLIB-NL preserves industrial-scale structure while enabling compact natural-language specifications through model–data separation. Experiments show that state-of-the-art NL-to-Opt systems degrade sharply on MIPLIB-NL despite strong performance on existing toy benchmarks, revealing substantial gaps in handling large-scale structure and constraint interactions.

## Acknowledgments

The computational resources were supported by the Center for Intelligent Computing and Song-Shan Lake HPC Center (SSL-HPC) in Great Bay University, Dongguan, China. This work was supported in part by National Key Research and Development Program of China under the grant numbers 2024YFA1012900 and 2025YFA1017800, and the National Natural Science Foundation of China under the grant numbers 12331010 and 12288101. We also thank the anonymous reviewers for their valuable feedback.

## Impact Statement

This paper presents research aimed at advancing the field of machine learning, with a specific focus on natural language interfaces for optimization modeling and systematic evaluation of such systems. The primary contributions of this work are methodological and empirical, including the construction of benchmarks, and evaluation protocols, intended to support future research in this area.

As a result, the work does not directly introduce a deployable decision-making system, nor does it prescribe or automate real-world optimization decisions in operational settings. Any downstream applications of models evaluated or developed using the proposed benchmarks would depend on additional system design choices, domain expertise, and human oversight.

Potential broader impacts of this line of research are largely aligned with those commonly associated with advances in machine learning and optimization technologies. These include the possibility of lowering barriers to formulating optimization problems, which may improve accessibility and productivity in scientific, industrial, and educational contexts, but may also raise familiar concerns related to over-reliance on automated modeling tools or misapplication in high-stakes domains if used without appropriate validation.

We do not identify any unique or immediate ethical risks specific to the contributions of this paper beyond those already well understood in the broader machine learning literature.

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

# Appendix

# Table of Contents

# A. Dataset Statistics and Comparative Analysis

This appendix provides a unified statistical and comparative analysis of MIPLIB-NL, situating it within the landscape of existing natural-language-to-optimization (NL-to-Opt) benchmarks. Throughout this section, comparisons with prior datasets serve to contextualize the regime targeted by MIPLIB-NL and to clarify which aspects of modeling complexity are emphasized or abstracted away in existing benchmarks. **Benchmark design trade-offs.** Table 2 situates existing NL-to-Opt benchmarks within a broad design space, revealing systematic trade-offs between problem scale, structural fidelity, domain coverage, and the cost and reliability of annotation. Many benchmarks favor small or moderately sized formulations that simplify annotation and verification, but in doing so abstract away indexed constraint families, large-scale structure, or solver-level semantics that are central to realistic optimization models. Others emphasize linguistic diversity, instruction following, or domain narrative breadth, often relying on synthetic generation or reverse construction pipelines that limit solver verifiability, reproducibility, or end-to-end evaluation. By contrast, MIPLIB-NL is designed to occupy a complementary and underexplored region of this design space. By grounding all instances in solver-validated MIPLIB models and preserving indexed constraints, numerical structure, and problem scale, it prioritizes structural fidelity and evaluation reliability over surface-level linguistic variation. As a result, MIPLIB-NL enables analyses that isolate the effects of scale, structural complexity, domain coverage, and optimization type under consistent solver semantics.

This perspective directly motivates the dimension-wise analyses that follow, which examine how these design dimensions vary across benchmarks and how they influence the empirical behavior of current NL-to-Opt systems.

We therefore begin Appendix A with two complementary benchmark-level comparisons. Table 2 summarizes broad dataset design dimensions, while Figure 8 focuses specifically on the amount of modeling structure preserved in natural language. For the latter audit, we use four levels: L0 denotes flat or enumerative NL with no explicit family structure; L1 denotes simple indexed families; L2 denotes coupled or multi-index families; and L3 denotes advanced temporal, routing, sequencing, or similarly structured scaffolds.

*Table 2.* A structured comparison of NL-to-Opt datasets and benchmarks. We contrast their provenance, scale, optimization scope, domain coverage, available artifacts (e.g., NL, math, code, solutions), and documented or observed limitations that influence end-to-end evaluation and reproducibility.

| Dataset | Paper / Link | Affiliation | Year / Venue | Size | Opt. Types | Domains | Artifacts | Notes / Limitations |
|---|---|---|---|---|---|---|---|---|
| NL4Opt (LPWP) | NL4Opt / LPWP | Huawei | 2023 (NeurIPS Comp.) | 1101 (289 test) | IP / LP / MILP | 6 domains | NL descriptions | Elementary-level problems; no guaranteed optimal solutions; not end-to-end; survey reports ∼26% error rate; limited OR jargon and implicit constraints; numerical values embedded in text. |
| IndustryOR (ORLM) | ORLM | SUFE | 2025 (Operations Research) | 686 seeds → 32,481 train; 100 test | IP / LP / MILP (+ others) | 8 domains | NL; math (added); solver code; opt sol | Synthetic expansion; solver-limited; follow-up works report missing/incorrect info and substantial filtering; survey reports ∼54% error rate. |
| MAMO-EasyLP | MAMO | CUHK | 2025 (NAACL Findings) | 652 | MILP | Not specified | NL; LP file | Synthetic; solver-limited; survey reports ∼8% error rate. |
| MAMO-ComplexLP | MAMO | CUHK | 2025 (NAACL Findings) | 211 | LP / MILP | Not specified | NL; LP file | Synthetic; solver-limited; higher error rate than EasyLP (∼24%). |
| OptiBench | OptiBench | CUHK | 2025 (arXiv) | 816 | LP / MILP | 80 domains (16 classes) | NL; JSON data; LP model | Model–data separation; reverse generation; still limited scale and solver-dependent. |
| ReSocratic | ReSocratic (E-OPT) | HKUST | 2025 (ICLR) | 605 bench; 29k train | LP / NLP | Not specified | NL; Python code; results | Synthetic; no explicit math formulation; numerical data embedded in NL; context-length constrained; mostly very small problems. |
| ComplexOR | Chain-of-Experts | ZJU | 2024 (ICLR) | 18 | MILP | Supply chain / logistics | NL; math; solver code; opt sol | Extremely small benchmark; numerical data in NL; follow-up works corrected multiple instances. |
| NLP4LP-Easy | OptiMUS | Stanford | 2023–2024 (ICML) | 287 | LP | Retail, energy, etc. | NL; structured repr; code; opt sol | Small-scale; model–data separation; survey reports ∼22% error rate. |
| NLP4LP-Hard | OptiMUS | Stanford | 2023–2024 (ICML) | 68 | LP / MILP | Retail, energy, etc. | NL; structured repr; code; opt sol | Small but harder subset; semi-structured. |
| OptMATH-Train | OptMath | PKU | 2025 (ICML) | 150k + 50k aug. | IP / LP / MILP / NLP / SOCP | 9 domains | Reverse-generated NL–Math | Large-scale training set; reverse generation from MIPLIB. |
| OptMATH-Bench | OptMath | PKU | 2025 (ICML) | 166 | IP / LP / MILP / NLP / SOCP | 9 domains | NL–Math benchmark | Moderate benchmark size; used for standardized evaluation. |
| OptiTrust | OptiTrust | IBM | 2025 (arXiv) | 15k train | MILP | Not specified | Reverse-generated | Not open-sourced; limited reproducibility. |
| WIQOR | WIQOR | Capital One | 2025 (arXiv) | 1946 | IP / LP / MILP | Not specified | NL spec; canonical model; what-if NL | Focus on what-if analysis rather than end-to-end modeling. |
| Step-Opt-Instruct | Step-Opt | CAS | 2025 (arXiv) | 260 seeds; 4500 gen | Not specified | Not specified | NL; math; program | Instruction-style synthetic data; coverage details vary. |
| Large-Scale-OR | Lean-LLM-Opt | SJTU | 2025 (SSRN) | 80 test; 86 ref | IP / LP | 6 types | NL; type; math; data details | First attempt at large-scale benchmark; limited size and scope. |
| LogiOR | ORThought | ZJU | 2025 (arXiv) | 92 | LP / ILP / MILP / NLP | Logistics | NL; math; Gurobi code; opt sol | Open-source; focused domain; strong in-domain evaluation. |
| EOR | EOR | CityU | 2025 (ICLR) | 30 | Inferable (MILP-heavy) | SCM / finance / logistics | NL; math; solver code; opt sol | Based on IndustryOR; not publicly available. |
| ZimplOR* | Multi-AgentOPT | Huawei | 2024 (INFOR) | 70 | LP / MILP / QP | 15 domains | NL; specs; Zimpl code | Not public; strong specs but strict complexity caps. |
| ORQA | ORQA | Huawei | 2025 (AAAI) | 1513 | N/A (QA) | 20 domains | NL; MCQ; answers; reasoning | QA-style benchmark; not end-to-end modeling. |
| Bench4Opt | ORGEval | CUHKS | 2025 (ICLR-S) | 394 test | LP / MILP | 6 domains | NL; params; LP model | Partly MIPLIB-derived (mostly LLM generated); model–data separation. |
| **MIPLIB-NL** | **Ours** | **GBU & PKU** | **2026 (ICML)** | **223 (extensible)** | **MILP** | **Broad (9 major / 54 sub-types)** | **Unified JSON: NL + data + math + code + opt sol** | **Reverse-generated from solver-verified MIPLIB 2017; strict data–problem separation; large-scale, diverse, reproducible; enables realistic evaluation beyond toy problems.** |

Together, Table 2 and Figure 8 clarify the comparison with prior benchmarks. Some prior work does preserve loop-based structure, so the distinction is not a binary "loop versus no loop" claim. However, richer NL-preserved structural complexity remains rare in absolute count and appears at much smaller solver scales than in MIPLIB-NL, which is the regime targeted by our benchmark. The following subsections then analyze these differences by scale, NL description size, domain coverage, and benchmark reliability.

## A.1. Scale and Structural Complexity

We compare datasets along two tightly coupled dimensions of complexity: *optimization scale* at the solver level (Figure 9), and *description scale* at the natural-language level (Figure 10). Together, these dimensions determine whether a benchmark

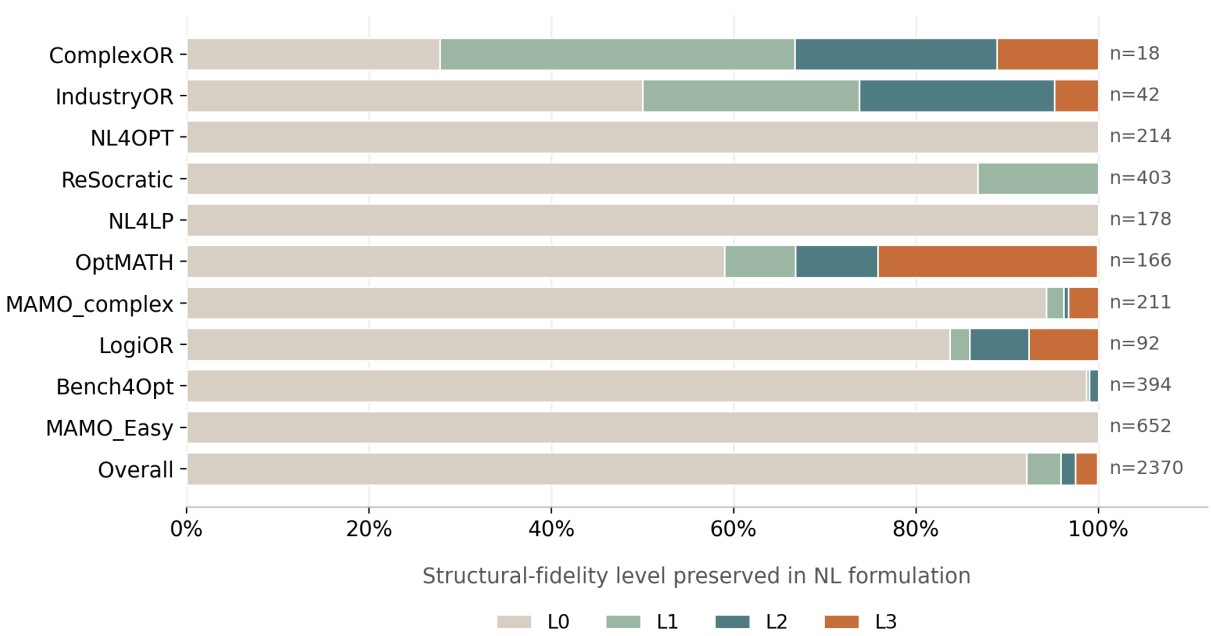

*Figure 8.* Audit of NL-preserved structural complexity across NL-to-Opt benchmarks. The comparison distinguishes flat descriptions (L0), simple indexed families (L1), coupled or multi-index families (L2), and advanced temporal/routing/sequencing scaffolds (L3). Overall, L0 accounts for 92.1%, while L2+L3 accounts for 4.1%, indicating that richer scaffolds exist but remain rare and concentrated in a small subset of datasets.

meaningfully reflects industrial optimization modeling practice.

**Optimization scale.** We first examine scale in terms of the number of decision variables and constraints, which directly reflect solver-level difficulty and the extent to which models rely on indexed constraint families.

Figure 9 contrasts the distributions of variable and constraint counts across benchmarks. Most existing NL-to-Opt datasets (e.g., NL4Opt, IndustryOR, MAMO-EasyLP, ReSocratic, ComplexOR, IndurstyOR) concentrate on problems with fewer than 100 variables and constraints, with even "large-scale" variants rarely exceeding a few hundred variables. In contrast, MIPLIB-NL occupies a qualitatively different regime. Median instances already involve on the order of $10^4$ variables, upper percentiles extend beyond $10^6$, and the largest instances exceed $10^7$ variables and constraints. This gap spans multiple orders of magnitude and reflects the industrial origins of MIPLIB-derived models rather than solver-limited synthetic construction.

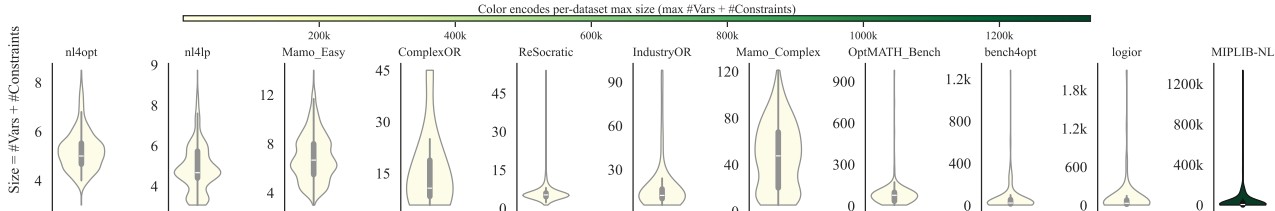

*Figure 9.* Distributions of problem sizes across optimization benchmarks, measured as the total number of variables plus constraints per instance. Each violin represents the per-instance size distribution of a dataset, with datasets ordered by their maximum size. A shared color scale encodes dataset scale, where warmer colors indicate larger maximum instance sizes. Compared to prior benchmarks, MIPLIB-NL spans a markedly larger scale and exhibits a substantially heavier upper tail, highlighting its suitability for evaluating large-scale optimization modeling.

At these scales, modeling challenges shift from algebraic transcription to *structural consistency*: index handling, constraint-family replication, and data binding errors become dominant failure modes (see Section 4.4 and Appendix G for details). Such

issues are largely invisible in toy-sized benchmarks, leading existing datasets to systematically overestimate performance under realistic modeling conditions.

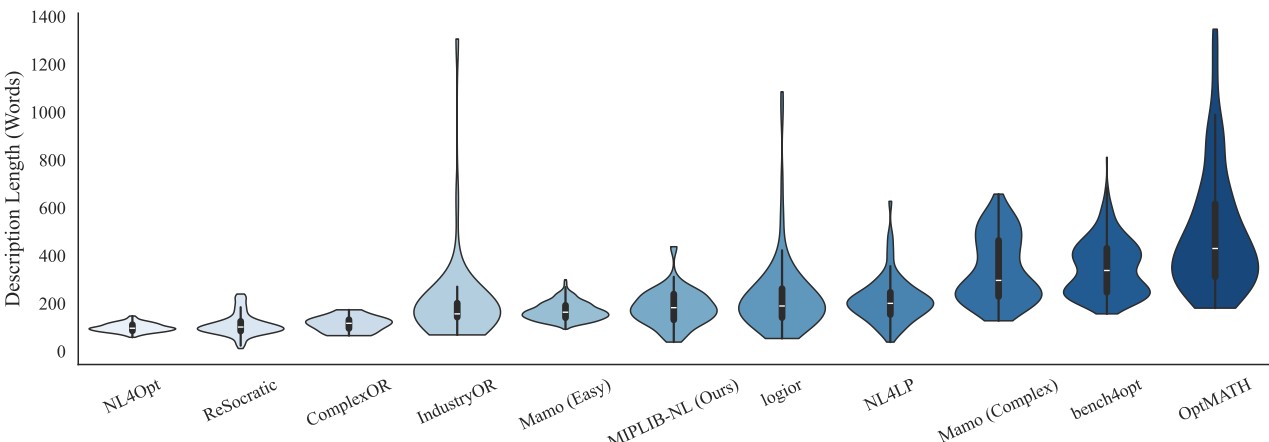

*Figure 10.* Natural-language description length across optimization benchmarks. Benchmarks that embed instance data directly into natural-language prompts exhibit rapidly increasing description length as problem complexity grows. Despite containing substantially more challenging optimization instances, **MIPLIB-NL** (ours) maintains moderate description length by adopting a model–data separation format, avoiding description lengths that would otherwise surpass the context limits of current large language models if instance data were embedded directly in natural-language prompts.

**Natural-language description scale.** Optimization scale alone does not tell the full story. A second, equally important dimension is how natural-language descriptions scale with problem size.

Figure 10 reports the distributions of natural-language description length across representative benchmarks. Existing datasets exhibit a strong coupling between problem size and NL length, largely because numerical data (indices, coefficients, instance-specific values) are embedded directly into text. As a result, NL descriptions grow rapidly with scale, imposing hard limits on instance size under realistic context constraints. By contrast, MIPLIB-NL enforces strict model–data separation. Natural-language descriptions specify *constraint families and loop-based structure*, while numerical data are stored externally. As a result, NL length remains largely independent of the number of variables and constraints, even as optimization scale increases by orders of magnitude.

**Implications.** Taken together, these comparisons highlight that scalability in NL-to-Opt is not primarily constrained by solver complexity, but by representational design. Benchmarks that entangle model structure with instance data conflate linguistic verbosity with modeling difficulty, whereas loop-based abstraction enables compact NL representations that faithfully scale to industrial-sized models.

### A.2. Domain Coverage and Problem-Type Diversity

We analyze problem-type coverage from two complementary perspectives: *application scenarios* and *optimization modeling taxonomy*.

At the scenario level, Table 3 categorizes instances by real-world application context, such as manufacturing, transportation, energy, healthcare, and agriculture. This view provides a descriptive summary of the semantic settings from which optimization problems are drawn. The composition of MIPLIB-NL directly reflects the content of MIPLIB 2017 and the availability of real industrial models, and therefore inherits the natural irregularities and domain biases present in that source.

Complementing this view, Table 4 analyzes coverage at the modeling level using a structured optimization taxonomy with nine core classes and extended subclasses, which are proposed by Lu et al. (2025). This taxonomy abstracts away surface-level narratives and instead captures how problems differ in decision structure, constraint families, and algorithmic formulation.

Jointly, these two perspectives show that MIPLIB-NL provides broad coverage not only of application narratives but also

*Table 3.* The application scenarios of the benchmarks, partially taken from Table 10 in Xie et al. (2025).

| Scenarios | NL4Opt | Mamo Easy | Mamo Complex | IndustryOR | NLP4LP | ComplexOR | OptiBench | OptMATH | Bench4Opt | LogiOR | MIPLIB-NL |
|---|---|---|---|---|---|---|---|---|---|---|---|
| Agriculture | 17 | 30 | 5 | 6 | 14 | 0 | 56 | 0 | 11 | 0 | 0 |
| Energy | 5 | 33 | 7 | 1 | 5 | 0 | 22 | 6 | 35 | 0 | 12 |
| Health | 45 | 49 | 53 | 3 | 49 | 2 | 41 | 1 | 20 | 0 | 6 |
| Retail | 16 | 47 | 37 | 11 | 21 | 1 | 40 | 5 | 0 | 1 | 1 |
| Environment | 8 | 40 | 0 | 0 | 9 | 0 | 12 | 0 | 0 | 1 | 6 |
| Education | 3 | 32 | 0 | 3 | 3 | 0 | 9 | 0 | 6 | 0 | 4 |
| Financial Services | 8 | 46 | 2 | 6 | 6 | 0 | 21 | 6 | 44 | 2 | 9 |
| Transportation | 51 | 73 | 76 | 18 | 48 | 7 | 87 | 50 | 94 | 67 | 58 |
| Public Utilities | 4 | 29 | 11 | 0 | 4 | 1 | 18 | 12 | 30 | 2 | 8 |
| Manufacturing | 61 | 71 | 8 | 45 | 68 | 6 | 230 | 57 | 90 | 14 | 23 |
| Software | 1 | 0 | 10 | 1 | 1 | 1 | 5 | 2 | 2 | 0 | 45 |
| Construction | 3 | 56 | 1 | 1 | 3 | 0 | 26 | 0 | 0 | 0 | 5 |
| Legal | 0 | 0 | 0 | 0 | 0 | 0 | 0 | 0 | 0 | 0 | 0 |
| Customer Service | 0 | 2 | 0 | 0 | 1 | 0 | 3 | 0 | 0 | 0 | 21 |
| Entertainment | 4 | 44 | 0 | 0 | 6 | 0 | 6 | 0 | 0 | 0 | 1 |
| Others | 4 | 100 | 1 | 5 | 4 | 0 | 29 | 27 | 62 | 5 | 24 |
| Sum | 230 | 652 | 211 | 100 | 242 | 18 | 605 | 166 | 394 | 92 | 223 |

of fundamentally different optimization structures.

## A.3. Summary of Comparative Findings

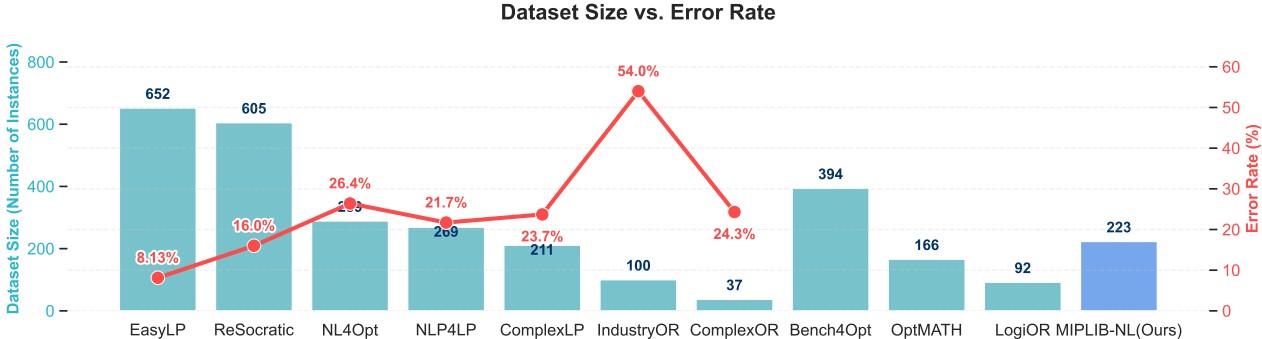

*Figure 11.* Dataset size versus error rate (reported by (Xiao et al., 2025; Yang et al., 2025b)) across existing NL-to-Opt benchmarks. Bars (left axis) report the number of instances in each dataset, while the line plot (right axis) shows the proportion of erroneous instances identified in prior audits. Despite moderate dataset sizes, several benchmarks exhibit high error rates, indicating substantial annotation noise in their purported ground-truth formulations. This reveals a critical disconnect between dataset scale and annotation reliability in existing NL-to-Opt benchmarks.

Figure 11 provides a complementary, dataset-level view of the design trade-offs summarized in Table 2. Notably, dataset size alone does not correlate with annotation correctness: multiple benchmarks with hundreds of instances exhibit error rates exceeding 20%–50%, suggesting that scaling up instance counts without systematic solver-level verification can substantially inflate noise in ground-truth formulations.

These findings reinforce that annotation reliability, rather than raw dataset size, is a bottleneck in realistic NL-to-Opt evaluation. In benchmarks constructed through manual authoring, instruction-style synthesis, or reverse generation without end-to-end solver validation, latent modeling errors can persist undetected and distort reported performance. As a result, empirical gains measured on such datasets may partially reflect robustness to annotation noise rather than true improvements in optimization modeling capability.

Taken together, the analyses in this appendix show that MIPLIB-NL occupies a qualitatively different evaluation regime from existing NL-to-Opt benchmarks. Rather than prioritizing instance count or linguistic surface diversity, MIPLIB-NL emphasizes large-scale optimization models characterized by indexed constraint families, rich loop-based structure, and strict separation between abstract model description and numerical data. This shift foregrounds structural reasoning, index consistency, and compositional model understanding as central challenges for NL-to-Opt systems.

**Verification and executability.** All instances in MIPLIB-NL are solver-executable. For the majority of benchmark-derived instances, optimal objective values are verified against reference solvers. Instances that are infeasible or have unknown optimal status are explicitly retained and labeled in the dataset to preserve modeling realism and reflect the full spectrum of industrial optimization practice. However, such instances are excluded from quantitative evaluation in the experimental sections, where performance metrics are computed only over instances with verified feasible and optimal solutions.

## B. Unified Instance Schema and Artifact Layout to Store Instances

Each instance in MIPLIB-NL is represented as a *self-contained, executable artifact* rather than a standalone text prompt or mathematical file. To support large-scale, verifiable, and reproducible optimization modeling, we adopt a unified instance schema that integrates natural language descriptions, external data tables, mathematical formulations, solver code, and verification logs. This appendix describes both the **logical schema** (expressed in `instance.json`) and the **physical artifact layout** (filesystem structure) that together define a complete instance.

### B.1. Design Principles

The instance schema is guided by the following principles:

- **Model–data separation.** Natural-language descriptions remain abstract and scalable, while large numerical tables are stored in external files.

- **Structural alignment.** Natural language, mathematical formulations, and solver code refer to the same index sets, parameters, and data files.

- **Scalability.** The schema supports instances with $10^3$–$10^7$ (even more) variables and constraints without exceeding context limits.

- **Auditability and reproducibility.** Every instance includes deterministic generator code, solver-executable implementations, and solver logs, enabling end-to-end verification and regeneration.

### B.2. Instance as a Self-Contained Artifact

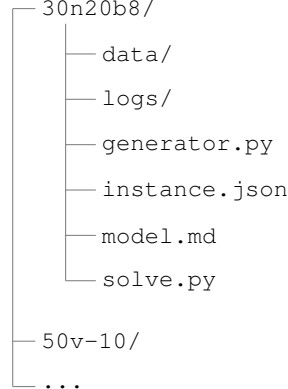

*Figure 12.* Structure of the dataset directories

Each instance is stored in its own directory and includes all components required for understanding, executing, and verifying the optimization model. Figure 12 illustrates a representative instance directory as follows:

- `instance.json`: unified schema linking all components.

- `model.md`: canonical mathematical formulation.

- `generator.py`: deterministic generator used to create or extend the instance.

- `solve.py`: solver-executable implementation.

- data/: external numerical tables (e.g., .csv).

- logs/: solver output and verification artifacts.

Among these files, three play a central and complementary role in defining an instance as a reusable, verifiable optimization artifact: instance.json specifies the logical schema and binds all components; model.md provides a solver-independent mathematical specification; and generator.py encodes the deterministic procedure by which concrete instances are constructed and scaled. The following subsections describe these three components in detail.

### B.3. Unified Instance Schema (`instance.json`)

```json
{
    "id": "nw04",
    "problem_type": {
        "major_category": "Transportation and Routing Optimization",
        "subcategory": "Aircraft Assignment Problem"
    },
    "abstract_problem": "Assume you are the crew scheduling manager for a major airline. You have a schedule of
N1 flight legs that must be operated in a specific period. Based on regulations and logistics, your team has
generated N2 possible crew pairings (a pairing is a sequence of flights a single crew can legally fly). The
operational cost for each pairing is listed in Table C1. The specific flight legs included in each pairing are
detailed in Table C2. Your goal is to select a subset of these pairings such that every single flight leg is
covered exactly once, minimizing the total operational cost.",
    "parameters": {
        "N1": 36,
        "N2": 87482
    },
    "files": {
        "files_1": {
            "path": "data/C1.csv",
            "description": "Table C1: The first column lists the Pairing IDs (variables), and the second column
shows the cost associated with assigning a crew to that pairing."
        },
        "files_2": {
            "path": "data/C2.csv",
            "description": "Table C2: This table defines the composition of each pairing. The first column
lists the Pairing ID, and the second column lists a Flight Leg ID included in that pairing. (This represents
the sparse constraint matrix)."
        }
    },
    "concrete problem":"……",
    "mathematical_formulation": "model.md",
    "optimal_value": 16862.0,
    "solver_code": "solve.py",
    "generator_code": "generator.py"
}
```

*Figure 13.* A compact instance illustration of our proposed json format.

As shown in Figure 13, each instance in MIPLIB-NL is represented by a unified instance.json file, which serves as a *structured index* binding natural language, numerical data, mathematical structure, and solver execution. Rather than embedding all information in a single artifact, the schema explicitly links to external data files and code, ensuring scalability, modularity, and structural consistency. Each instance contains the following top-level fields (field names shown in monospace):

- id: a globally unique identifier that links all artifacts associated with the instance.

- problem_type: taxonomy labels (major_category, subcategory) used for analysis, stratified evaluation, and controlled dataset construction.

- abstract_problem: a high-level natural-language description specifying the decision setting, objective intent, and major constraint families. Numerical values are omitted and referenced symbolically via parameters and files, allowing the description to remain compact and stable as scale increases.

- parameters: symbolic quantities controlling problem scale and structure (e.g., numbers of nodes, commodities, or time periods), which also serve as knobs for controlled instance variation.

- files: external data tables (typically .csv) that instantiate index sets and parameters referenced throughout the instance. Each file is accompanied by a semantic description of its contents, enabling explicit and unambiguous data binding across language, mathematics, and code.

- `concrete_problem` (optional): a numerically instantiated natural-language description intended for small-scale or didactic cases where explicit values do not exceed context limits.

- `mathematical_formulation`: a pointer to a canonical, index-explicit formulation file (e.g., `model.md`) that defines sets, variables, objective, and constraint families using symbolic notation. This artifact preserves loop structure and constraint semantics and serves as the structural reference point for the instance.

- `solver_code`: executable code that instantiates indexed variables and constraints from external data and invokes an industrial-grade solver for execution and verification.

- `generator_code`: the deterministic generator used to produce the instance, providing traceability from abstract structure to concrete realization and enabling controlled augmentation.

- `optimal_value`: the solver-verified objective value when available.

- `verification`: structured solver metadata, including solution status, runtime, optimality gap, and paths to solver logs when available.

- `metadata`: auxiliary information such as language, difficulty tag, size summaries, and provenance when available.

An illustrative `instance.json` example is shown in Figure 13, which demonstrates how these fields are populated and linked in practice. The example is provided solely to clarify schema structure and field semantics.

### B.4. Canonical Mathematical Formulation (`model.md`)

The file `model.md` provides a canonical, index-explicit mathematical specification of the optimization problem. Unlike solver-oriented LP/MPS files, `model.md` is designed to be human-readable and structurally transparent. The formulation explicitly declares: (i) index sets and parameters, (ii) decision variables with their domains, (iii) objective functions, and (iv) constraint families expressed using symbolic notation. Constraint families are written at the loop level (e.g., $\forall i \in \mathcal{I}, \forall t \in \mathcal{T}$), preserving the structural information that is typically flattened away in low-level solver formats. As such, `model.md` serves as the *structural reference point* of an instance, aligning natural-language descriptions, external data tables, and solver implementations. Optional extensions or controlled variants may also be documented here to clarify the scope of the modeled problem.

### B.5. Generator Templates (`generator.py`) and Controlled Instance Synthesis

In addition to storing solver-verified instances, MIPLIB-NL releases *expert-defined generator templates* for each problem instance (`generator.py`). A generator template is a deterministic program that produces the instance directory described above, including external data tables (`data/`), the unified schema file (`instance.json`), and (when applicable) the corresponding formulation and solver artifacts. This design supports reproducible instance creation and enables controlled augmentation (e.g., scaling index sizes or rebalancing scenario coverage).

**Template contract.** Each generator template implements a consistent contract: given a set of *structural parameters* (e.g., number of nodes, commodities, time periods, density), it produces (i) tabular data files with well-defined schemas and (ii) a JSON record that binds these data to the NL description, math specification, and solver code. Concretely, the generator output must satisfy:

- *Determinism*: generation is reproducible given a random seed.

- *Schema compliance*: all produced files match the expected column names and formats referenced by `instance.json`.

- *Structural alignment*: generated indices and parameters are consistent across `abstract_problem`, `model.md`, and `solve.py`.

**Example: MCND data generator.** As a representative example, the Multi-Commodity Capacitated Network Design (MCND) generator produces five CSV files: `nodes.csv`, `arcs.csv`, `commodities.csv`, `arc_costs.csv`, and `parameters.csv`. The generator exposes explicit knobs controlling both scale and structure, including the number of nodes ($|\mathcal{N}|$), number of commodities ($|\mathcal{K}|$), and graph density. These parameters induce predictable changes in loop-based constraint families:

- *Commodity replication* ($k \in \mathcal{K}$) produces stacked copies of flow-conservation families.

- *Graph density* controls the number of arcs and therefore the size of arc-indexed capacity and cost constraints.

- *Node count* affects the dimensionality of node-wise balance families.

The output `parameters.csv` records the realized structural parameters (e.g., `n_nodes`, `n_commodities`, `n_arcs`, `density`), ensuring that downstream components can bind these values consistently.

**Separation of data generation vs. language realization.** Generator templates are responsible for creating *structured data* and (optionally) a raw, schema-grounded `abstract_problem` draft that references the produced files and parameters. When we apply optional LLM-based language refinement, it is strictly constrained to be *non-semantic*: the refinement is permitted to rewrite for fluency and readability but is not allowed to introduce, remove, or alter any modeling content. In particular, refinement must preserve:

- the set of decision variables and their indices,

- the objective intent and terms (up to paraphrase),

- the set of constraint families and their loop scopes,

- all file bindings (`files.*.path`) and parameter references.

This separation ensures that correctness and verifiability are guaranteed by template logic and solver-grounded checking, while still allowing natural language to be presented in a human-readable style.

### B.6. Relation to Repository Organization

At the dataset level, instances are organized by problem type, following a taxonomy of nine major categories and 54 subcategories from Lu et al. (2025), which is extended to Table 4. Each leaf directory contains multiple instances of the same subclass, each conforming to the unified schema described above. This organization enables systematic analysis, stratified evaluation, and controlled instance generation across both semantic and structural dimensions.

## C. Compression Ratio Analysis

### C.1. Compression from the Perspective of Information Theory

To rigorously analyze the transformation from MPS to our natural language (NL) and CSV representation, we model the optimization problem as a random variable $X$ defined on a probability space $(\Omega, \mathcal{F}, P)$. We decompose the random variable into a tuple $X = (M, D, S)$, where:

- $M$ represents the mathematical structure (e.g., constraint types, variable dependencies, graph topology);

- $D$ represents the metric data (e.g., coefficient matrices, bounds, rhs values);

- $S$ represents the semantic context (e.g., physical meaning, domain logic, variable naming conventions).

The traditional MPS format can be viewed as an encoding function $Y = E_{\text{MPS}}(X) \approx (M, D, \epsilon)$. This process is mathematically lossless regarding $M$ and $D$, ensuring solver feasibility. However, from a semantic perspective, it acts as a *lossy compression* where the conditional entropy $H(S|Y)$ is high, effectively discarding the semantic context $S$. In the pre-LLM era, this loss was acceptable as the utility function focused solely on numerical solvability.

In the context of Large Language Models (LLMs), the utility function shifts towards semantic understanding, requiring the recovery of $X$ from $Y$. Since the direct inverse is ill-posed, our work functions as a reconstruction estimator $\hat{X} = D_{\text{New}}(Y, W) = (\hat{M}, \hat{D}, \hat{S})$. Here, $W$ represents *side information* introduced through structural pattern recognition and heuristic inference. Our transformation ensures strict mathematical equivalence ($\hat{M} = M, \hat{D} = D$) while generating a compatible semantic description $\hat{S}$ that maximizes the probability $P(\hat{S}|M, D)$. We serialize $\hat{X}$ into a dual-channel format $Z = (Z_{\text{NL}}, Z_{\text{CSV}})$, separating *structural entropy* (NL) from *metric entropy* (CSV) to minimize the Kullback-Leibler divergence $D_{\text{KL}}(P_Z||Q_{\text{dec}})$ with the decoder's prior.

*Table 4.* Augmented application taxonomy of optimization problem types. We keep the original nine main classes from Lu et al. (2025) unchanged and add additional six main classes (and subclasses) to accommodate MIPLIB-derived instances.

| Main Class | Problem Class | Instances |
|---|---|---|
| Assignment and Resource Allocation Optimization | Car Selection Problem | – |
| | Contract Allocation Problem | – |
| | Assignment Problem | `assign1-5-8`, `netasgn_parsed`, `bnatt400` `bnatt500`, `rmatr100-p5`, `rmatr100-p10` `rmatr200-p5`, `rmatr200-p10`, `rmatr200-p20` `f2gap40400`, `f2gap201600`, `f2gap801600` `f2gap401600` |
| | Structure-Based Assignment Problem | – |
| | Team Formation Problem | – |
| | Military Personnel Deployment Problem | – |
| Combinatorial Optimization | Knapsack Problem | `mik-250-20-75-1`, `mik-250-20-75-2`, `mik-250-20-75-3` `mik-250-20-75-4`, `mik-250-20-75-5` |
| | Market Share Optimization Problem | `marketshare4`, `markshare_5_0` |
| | Set Multi-Cover Problem | `iis-hc-cov`, `iis-glass-cov`, `beasleyC3` |
| | Set Cover Problem | `stein9inf`, `stein15inf`, `stein45inf` `seymour1` |
| Cutting and Packing Optimization | Bin Packing Problem | `bppc4-08`, `bppc6-02`, `bppc6-06` `bppc8-02`, `bppc8-09` |
| | Blending Problem | – |
| | Cutting Stock Problem | `glass-sc` |
| Domain-Specific Optimization | Diet Problem | – |
| | Unit Commitment Problem | `thor50dday` |
| | Farm Planning Problem | `gmu-35-40`, `gmu35-50` |
| Facility Location Optimization | Facility Location Problem | – |
| | Capacitated Facility Location Problem | – |
| | Transportation Problem (Airline Resource Allocation) | `air03`, `air04`, `air05` |
| | Facility Dispersion Problem | – |
| Financial and Revenue Optimization | Portfolio Optimization Problem | – |
| | Profit Maximization Problem | – |
| | Revenue Management Problem | – |
| | Revenue Maximization Problem | – |
| Network Flow Optimization | Multi-Commodity Capacitated Network Design Problem | `MCND` |
| | Multi-Commodity Transportation Problem | – |
| | Minimum Cost Flow Problem | `network_flow` |
| | Multi-Commodity Network Flow Problem | `MCND` |
| | Network Flow Problem | `neos-1425699`, `neos5` |
| | Static Line Planning Problem | `StaticLinePlanning` |
| | Supply Chain Optimization | `SupplyChain` |
| | Network Optimization | – |
| Production Planning and Scheduling Optimization | Capacitated Lot-Sizing Problem | `prod1`, `prod2` |
| | Factory Planning Problem | `decomp1`, `decomp2` |
| | Flow Shop Scheduling Problem | – |
| | Job Shop Scheduling Problem | `30n20b8` |
| | Discrete Lot-Sizing and Scheduling Problem | – |
| | Production Planning Problem | `exp-1-500-5-5` |
| | Lot-Sizing Problem | `exp-1-500-5-5` |
| Transportation and Routing Optimization | Aircraft Assignment Problem | `air03`, `air04`, `air05` |
| | Aircraft Landing Problem | `flugpl`, `flugplinf` |
| | Transportation Problem | `rail507`, `fast0507` |
| | Traveling Salesman Problem | `TSP`, `eil33-2`, `eilA101-2` `eilC76-2` |
| | Operations Optimization | `wachplan`, `timtab1`, `supportcase33` |
| | Capacitated Vehicle Routing Problem with Time Windows | – |
| Graph-Structured Optimization (new) | Graph Coloring / Chromatic Index | `chromaticindex32-8`, `chromaticindex128-5`, `chromaticindex256-8` `chromaticindex512-7`, `chromaticindex1024-7` |
| | Clique / Conflict Packing | `cvs08r139-94`, `cvs16r106-72`, `cvs16r128-89` `cvs16r70-62`, `cvs16r89-60` |
| | Graph Covering / Dominating Set | `v150d30-2hopcds` |
| Disjunctive and Pairwise Scheduling (new) | Pairwise Disjunctive Scheduling | `supportcase21i`, `supportcase26`, `supportcase33` |
| | Compatibility / Separation Scheduling | – |
| Random-Graph Benchmark MILPs (new) | Random Graph Cover / Pack (structural) | `graph20-20-1rand`, `graph20-80-1rand`, `graph40-20-1rand` `graph40-40-1rand`, `graph40-80-1rand` |
| | Generic Graph-Indexed MILPs | `pg`, `pk1`, `p2m2p1m1p0n100` |
| Synthetic / Prototype MILPs (new) | Synthetic Examples / Prototypes | `ex9`, `ex10`, `prototype` |
| | Large Synthetic MILPs | `ex1010-pi` |
| Dense Algebraic / Stress-test MILPs (new) | Dense Constraint Matrices | `h50x2450`, `h80x6320d`, `ramos3` |
| | Misc. Stress-test Instances | `cost266-UUE` |
| Misc. Application Stubs (new) | Sports / Tournament Scheduling | `b-ball` |
| | Small Misc. MILPs | `50v-10`, `app1-1` |
| **Others** | | – |

Finally, the substantial reduction in file size ($|Z| \ll |Y|$) can be explained by *Kolmogorov Complexity* $K(X)$. The MPS format relies on an *extensional definition*, explicitly enumerating every scalar constraint and variable, leading to significant syntactic redundancy where the file size scales linearly with the problem dimension $O(N)$. In contrast, our NL+CSV format serves as an *intensional definition*, capturing the generative logic (e.g., loop structures) in the NL component. This allows the representation to approach the theoretical lower bound $K(X)$, where the structural description size is nearly invariant to $N$, and the CSV compactly stores only the non-redundant sparse data.

### C.2. Examples of Compression

We present two examples to illustrate the mechanics of file size compression observed in our study. While our actual methodology involves extracting natural language descriptions and data structures from MPS files, for the sake of clarity, we present the reverse-engineered problem formulation and data format first, followed by their corresponding representations within the MPS files.

**Example 1: `ex9`.** The first example, problem `ex9`, is equivalent to the following integer programming problem: a tiling puzzle on a $9 \times 9$ grid. Each puzzle piece possesses four edges (top, right, bottom, left). Each edge is characterized by an attribute: either a boundary edge marked as 0 (simulating the straight edge of a physical puzzle piece) or an internal edge marked with a specific positive integer (slightly different from the physical interlocking tabs of real jigsaw puzzles). The assembly rules dictate that the contacting edges of adjacent puzzle pieces must share the same positive integer mark. Based on geometric features, the pieces are categorized into three types: 4 corner pieces (possessing two adjacent boundary edges), 28 edge pieces (possessing one boundary edge), and 49 internal pieces (possessing no boundary edges). The file `numbers` provides the edge markings for all pieces. The objective is to determine a valid layout that satisfies these rules, with the objective function set to maximize the number of successfully placed pieces. Consequently, the data file need only store the four edge values for each piece. Although we include an additional ID field for convenience, the final data file stores only $81 \times 5 = 405$ integers. The combined storage for the problem description and data is merely 3.29 KB.

In contrast, the original MPS file employs binary integer variables in the form $b_{x\_ijr}$ ($0 \le x \le 80$, $0 \le i, j \le 8$, $0 \le r \le 4$) to indicate that piece $x$ is placed at position $(i, j)$ after being rotated counter-clockwise $r$ times. For instance, the variable `b64_713` signifies that piece 64 is placed at position $(7, 1)$ after 3 counter-clockwise rotations. `NOD` constraints ensure that each piece is placed in at most one location, while `NODiXj` constraints ensure that position $(i, j)$ holds only one piece. The matching constraints (`EQ`) are significantly more complex. For example, the constraint `EQ1`:

$$\texttt{b0\_110} - \texttt{b0\_212} - \texttt{b13\_212} - \texttt{b15\_211} - \cdots \le -0.0$$

implies that if piece 0 is placed at $(1, 1)$ without rotation, specific compatible pieces (potentially rotated) must be placed at $(2, 1)$. This is because the bottom edge of piece 0 matches the values of the bottom edge of piece 0, the bottom of piece 13, the right of piece 15, and so forth. Due to these complex constraints, the repetitive variable names in the MPS file, and significant whitespace, the number of non-zero elements reaches 517,112. The file size balloons to 15,917 KB, which is thousands of times larger than the compressed representation.

**Example 2: `wachplan`.** Another illustrative example is `wachplan`. In this specific instance, a sailing training ship with 8 cadets requires a cyclic watch schedule covering 28 time periods (indexed 1 to 28, where period 1 follows period 28). The rules are as follows:

- Each time period must be staffed by 2 to 3 cadets.

- No cadet is permitted to serve consecutive watches (i.e., working in adjacent time periods is prohibited).

- In any sequence of 6 consecutive time periods, a cadet's cumulative watch count must not exceed 2.

- Any pair of cadets on the ship must have served together during the same time period at least once.

- The total number of watches served over the entire cycle must be exactly equal for every cadet.

- Fixed assignments: In period 1, cadets 1, 2, and 3 must watch; in period 2, cadets 4 and 5 must watch (cadet IDs are 1 to 8).

The objective is to determine the maximum total number of watches per cadet within this cycle.

This problem can be clearly and fully described using natural language without the need for an auxiliary data file. However, to provide an equivalent description in the MPS file, boolean variables are used, such as $x\#i\#t$ (whether cadet $i$ is on watch at time $t$), $ythree\#i\#j\#k\#t$ (whether cadets $i, j, k$ are scheduled together at time $t$), $ytwo\#i\#j\#t$ (whether cadets $i, j$ are scheduled), and $z\#i\#j\#t$ (whether cadet $i$ and $j$ have served together by time $t$). These variables are linked via a series of complex constraints. Ultimately, the MPS file contains 89,361 non-zero elements and occupies 5,861 KB, whereas the natural language description requires only 2.77 KB.

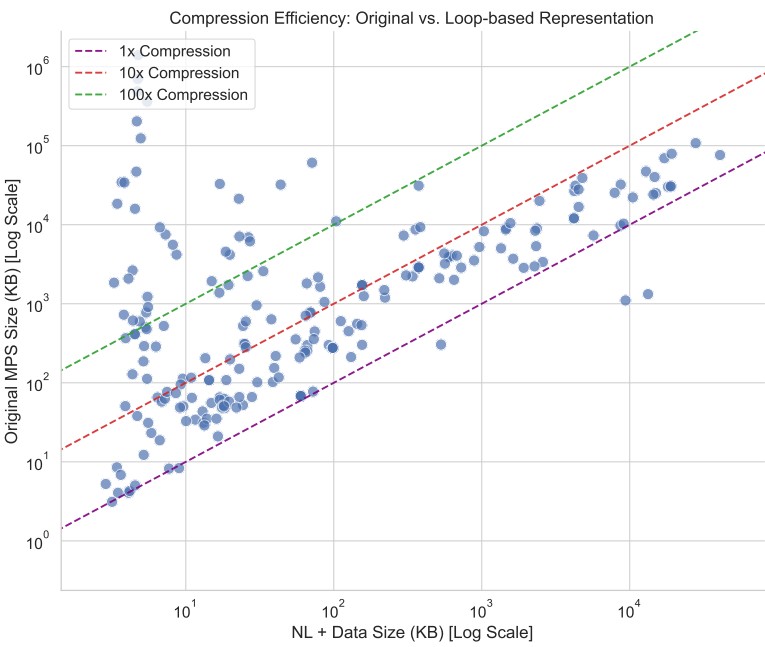

*Figure 14.* Comparison of original MPS file size versus the compressed loop-based representation size (NL + Data, log-log scale). Dashed lines indicate $1\times$, $10\times$, and $100\times$ compression ratios.

## C.3. Compression Ratio Statistics

Across MIPLIB-NL, loop-based abstraction yields substantial compression from expanded LP/MPS representations to a small set of loop templates. In the current build, we observe the following indicative statistics:

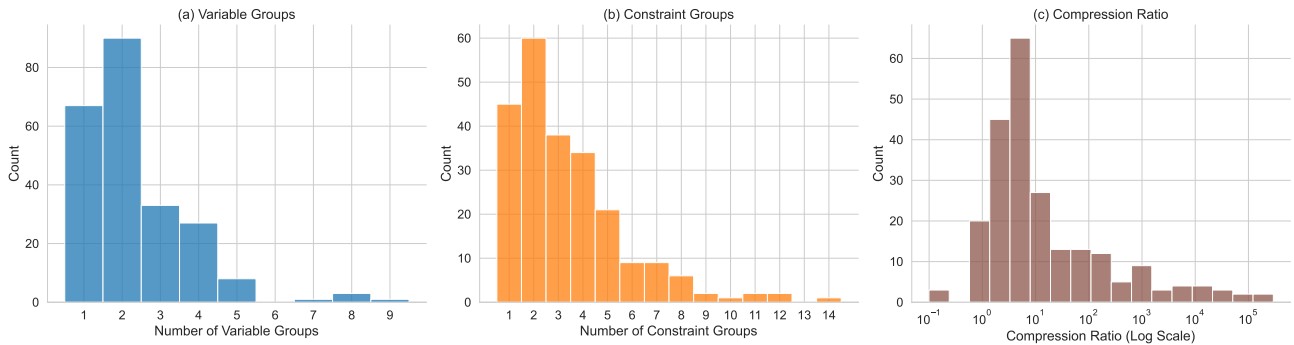

*Figure 15.* Distribution statistics across MIPLIB-NL instances: (a) Number of variable groups; (b) Number of constraint groups; (c) Compression ratio (log scale).

- **Original MPS Files for MIPLIB-NL:** Median file size 1,194 KB (max 1,398,509 KB); median variables 2,183 (max 283,648); median constraints 1,388 (max 1,050,112).
- **MIPLIB-NL:** Median data file size 36.3 KB (max 40,779 KB); median variable groups 2 (max 9); median constraint groups 3 (max 14).

- **Compression Ratio:** Defined as the ratio of the original MPS file size to the combined NL and Data file sizes, the median is 6.5, with a maximum reaching 293,277.

Figure 14 visualizes the relationship between the original MPS file sizes and the compressed representation sizes. As shown, the majority of instances fall comfortably below the $10\times$ compression line, with many exceeding $100\times$.

Furthermore, Figure 15 illustrates the distribution of structural complexity and compression efficiency across the dataset. Most problems can be described using a very small number of abstract variable and constraint groups (panels a and b), highlighting the structural regularity captured by our approach. Panel (c) confirms the significant space efficiency gains, showing the overall distribution of compression ratios.

These trends highlight loop-based abstraction as a practical and scalable mechanism for Opt-to-NL generation and for evaluating NL-to-Opt systems under industrial-scale structural complexity.

## D. Constraint Types under Loop-Based Structural Scaffolds

This appendix provides a structure-aware analysis of constraint types in optimization models, with a particular focus on how constraints are instantiated as large indexed families through loop-based aggregation patterns. Our analysis builds on existing NL4Opt-style constraint classifications (Ramamonjison et al., 2022; Li et al., 2023), which organize constraints according to their *semantic forms* (e.g., bounds, sums, ratios, and logical relations), and augments them with an explicit *structural* dimension that captures how such semantic forms scale and interact in practical mixed-integer linear programs (MILPs).

### D.1. Semantic Constraint Types from Prior Work

Prior work on NL-to-Opt benchmarks (Ramamonjison et al., 2022; Li et al., 2023) has shown that a relatively small set of semantic constraint templates is sufficient to describe the surface algebraic forms of many optimization constraints in MILP problems. Representative categories include variable bounds, linear sum constraints, weighted capacity constraints, proportion constraints, comparison relations, and simple logical constraints. These semantic types provide a useful vocabulary for mapping NL descriptions to algebraic expressions, particularly in small or textbook-scale problem instances.

*Table 5.* Semantic constraint types adapted from prior NL4Opt-style classifications (Ramamonjison et al., 2022; Li et al., 2023), augmented with additional atomic constraint forms observed in MIPLIB-derived instances. Semantic types describe atomic algebraic relations, while loop-based scaffold annotations indicate how these atoms are instantiated as indexed constraint families in practical MILP models.

| Type | Semantic constraint type | Mathematical form | Typical loop scaffold | Structural interpretation |
|---|---|---|---|---|
| 1 | Upper bound (single var.) | $x_i \leq b$ | Global / single-loop | Scalar bound replicated over indices |
| 2 | Upper bound (sum) | $\sum_i x_i \leq b$ | Global / single-loop / subset-loop | Aggregation over indexed sets |
| 3 | Upper bound (weighted sum) | $\sum_i a_i x_i \leq b$ | Single-loop / nested-loop | Indexed capacity or budget families |
| 4 | Upper bound (proportion) | $x_j \leq c \sum_i x_i$ | Single-loop / coupled-loop | Ratio over aggregated quantities |
| 5 | Lower bound (single var.) | $x_i \geq b$ | Global / single-loop | Minimum activity or demand |
| 6 | Lower bound (sum) | $\sum_i x_i \geq b$ | Single-loop / subset-loop | Covering or service requirements |
| 7 | Lower bound (weighted sum) | $\sum_i a_i x_i \geq b$ | Single-loop / nested-loop | Resource fulfillment over indices |
| 8 | Lower bound (proportion) | $x_j \geq c \sum_i x_i$ | Coupled-loop | Proportional allocation constraints |
| 9 | Comparison constraint | $d x_i \leq x_j$ | Single-loop / paired-loop | Indexed balance or dominance |
| 10 | Logical implication | $y_A \leq y_B$ | Single-loop / coupled-loop | Conditional activation (often via Big-$M$) |
| 11 | Exactly-one constraint | $\sum_{k \in S} y_k = 1$ | Single-loop / subset-loop | Discrete choice / partitioning |
| 12 | At-least-one constraint | $\sum_{k \in S} y_k \geq 1$ | Single-loop / subset-loop | Logical covering constraints |
| 13 | At-most-one constraint | $\sum_{k \in S} y_k \leq 1$ | Single-loop / subset-loop | Logical packing or conflicts |
| *Additional atomic constraint types observed in practice* | | | | |
| 14* | Equality constraint | $x_i = b$ or $\sum_i a_i x_i = b$ | Single-loop / nested-loop | Conservation, balance, or exact fulfillment |
| 15* | Aggregation definition | $H_t = \sum_i a_i x_{i,t}$ | Single-loop / coupled-loop | Auxiliary variable definition for abstraction |
| 16* | Indicator constraint | $y = 1 \Rightarrow a^\top x \leq b$ | Paired-loop / coupled-loop | Logic-gated constraint activation |
| 17* | Domain / integrality constraint | $x_i \in \{0,1\}$ or $x_i \in \mathbb{Z}$ | Single-loop | Discrete decision enforcement |

Table 5 reproduces these semantic constraint types following existing formulations, while extending them with annotations that indicate the *loop-based structural scaffolds* under which these semantic forms most commonly arise. Importantly, our goal is not to redefine or replace the constraint taxonomies proposed in prior work, but rather to make explicit a structural modeling dimension that is typically implicit or flattened away in existing NL-to-Opt benchmarks.

**Semantic–Structural Orthogonality.** A central modeling principle underlying this appendix is that a *constraint family* in

a practical MILP model is not a primitive object, but the outcome of a *loop-based generation mechanism*. In particular, constraint families are produced when an atomic semantic constraint template is instantiated through a specific looping rule over one or more index domains.

In this formulation, semantic constraint types specify the atomic algebraic relations to be enforced (e.g., bounds, sums, comparisons, or logical implications), while index sets provide the static domains over which such relations may be instantiated. A scaffold specifies *how* index domains are traversed, how aggregation ranges are formed, and how constraints are replicated or coupled (e.g., via nesting, subset selection, sliding windows, pairwise instantiation, or temporal recursion).

The expanded collection of constraints generated by applying a loop-based scaffold to a semantic constraint template over given index domains—often numbering in the thousands at the solver level—is what we refer to as a *constraint family*.

From an analytical perspective, it is therefore useful to characterize each constraint family using two orthogonal descriptors:

$$\text{Constraint Family} \approx \langle \text{Semantic Constraint Type, Loop-Based Structural Scaffold} \rangle.$$

This representation should be understood as an analytical characterization rather than a generative equation. The semantic constraint type determines *what* algebraic relation is enforced, while the loop-based structural scaffold determines *how* that relation is generated and scaled across index domains.

Importantly, two models may share identical index sets yet induce fundamentally different constraint families due to differences in loop-based scaffolds. Conversely, the same scaffold may host multiple semantic constraint types. Although a single constraint family may involve multiple semantic atoms, treating semantic constraint types and loop-based structural scaffolds as orthogonal analytical dimensions provides a principled framework for understanding how constraint systems scale from isolated textbook formulations to large, industrial-scale MILP models.

### D.2. Loop-Based Structural Scaffolds

In industrial-scale optimization models, constraints rarely appear as isolated expressions. Instead, they are generated as *constraint families* through loops over index sets such as products, locations, time periods, network nodes, or scenarios. At the solver level, these families may expand into thousands or millions of algebraic constraints, while at the modeling level they typically correspond to a small number of conceptual rules.

While many existing benchmarks implicitly assume simple loop patterns (e.g., single-index repetition), our MIPLIB-derived instances exhibit a much richer set of loop-based structural scaffolds. Crucially, these structural scaffolds are largely independent of the underlying semantic constraint types. The same semantic atom—such as a capacity constraint or a logical implication— may give rise to dramatically different constraint families depending on how it is instantiated through indexing and aggregation. As a result, semantic constraint coverage alone substantially underestimates the structural richness and modeling difficulty of real-world MILP instances. Below we summarize the major loop-based structural scaffolds together with canonical mathematical forms and at least one concrete example from our instances.

**(i) Global (non-indexed) constraints.** These constraints do not explicitly quantify over an index set:

$$a^\top x \leq b.$$

They typically represent global budgets or single-shot coupling rules.

**(ii) Single-loop constraint families with inner aggregation.** Constraint families in this category are generated by a single loop over an index set $I$, with one constraint instantiated for each $i \in I$. Within each constraint, an inner aggregation over a secondary index set is used to express resource usage, coverage, or assignment relations:

$$\forall i \in I: \quad \sum_{j \in J(i)} a_{ij} x_{ij} \leq b_i.$$

For example, in `v150d30-2hopcds`, coverage is enforced independently for each demand point $i$ by requiring at least one selected facility to cover it:

$$\sum_{j=1}^{N} a_{ij} x_j \geq 1 \quad \forall i.$$

**(iii) Nested-loop (multi-index) constraint families.** Constraint families in this category are generated by multiple nested loops over two or more index sets, producing a multi-dimensional block of constraints. A canonical form is:

$$\forall (i,t) \in I \times T: \quad \sum_{j \in J} a_{ijt} x_{ijt} \leq b_{it}.$$

A representative example appears in `30n20b8`, where execution decisions are indexed by job, mode, and start time, and each job must be executed exactly once:

$$\sum_{m \in M} \sum_{t=0}^{TL - D_{j,m}} x_{j,m,t} = 1, \quad \forall j \in J.$$

Such nested-loop structures form the structural backbone for many high-dimensional models, and serve as the basis upon which more specialized aggregation patterns (e.g., sliding-window or convolution-style sums, which will be introduced later) are constructed.

**(iv) Subset-indexed (implicit-set) loops.** In this scaffold, constraint families are generated by looping over a collection of *predefined subsets* rather than over a regular index set. These subsets typically correspond to semantic groupings such as conflict groups, cliques, neighborhoods, or compatibility sets. Each constraint aggregates variables belonging to a specific subset, and the subsets themselves may be irregular and overlapping:

$$\forall k \in \mathcal{K}: \quad \sum_{j \in S_k} x_j \leq 1.$$

Such constraints are common in conflict, clique, or compatibility modeling, where each $S_k$ represents a group of mutually incompatible decisions. For instance, in `gmu-35-40`, mutually exclusive activity groups are encoded by defining conflict sets $S_k$ and enforcing:

$$\sum_{(i,j) \in S_k} x_{i,j} \leq 1 \quad \forall k.$$

**(v) Coupled or recursive (temporal) loops.** In this scaffold, aggregate quantities are first defined independently at each time period and are then linked across periods through recursive constraints. Such coupling introduces dependencies between constraint families indexed by different time steps. A canonical pattern is:

$$\forall t \in T: \quad H_t = \sum_i a_i x_{i,t}, \qquad \forall t \geq 2: \quad H_t \geq \alpha H_{t-1}.$$

Here, the first loop defines an auxiliary aggregate variable $H_t$ at each time step, while the second loop enforces temporal consistency or smoothing across adjacent periods.

A representative example appears in `gmu-35-40`, where harvest volume is defined for each period and then constrained to evolve smoothly over time:

$$H_j = \sum_{i=0}^{M-1} V_{i,j} x_{i,j}, \qquad H_j \geq N_1 \cdot H_{j-1}, \ \ H_j \leq N_2 \cdot H_{j-1}, \quad \forall j \in \{2, \ldots, T-1\}.$$

**(vi) Cyclic or modular-index loops (wrap-around and periodic constraints).** In this scaffold, the index structure itself is defined modulo a fixed period, leading to cyclic adjacency relations or periodic feasibility patterns. Unlike standard linear indexing, constraints may "wrap around" index boundaries or selectively activate variables based on modular conditions.

A representative example appears in `wachplan`, where the planning horizon is cyclic and the successor of the final time slot is the first slot. Adjacency constraints are therefore enforced using modular indexing:

$$x_{i,t} + x_{i,(t \bmod S)+1} \leq 1, \quad \forall i \in I, \ \forall t \in T.$$

A different modular pattern appears in `30n20b8`, where feasibility of decision variables depends on periodic time blocks. Here, the time axis is partitioned into cycles of fixed length, and variables are disabled outside admissible windows within each cycle:

$$x_{j,m,t} = 0, \quad \forall j \in J, \ \forall m \in M, \ \forall t \text{ s.t. } (t \bmod 25) \geq L_{j,m}.$$

Both examples illustrate how modular arithmetic on indices introduces structural constraints that cannot be captured by linear time indexing alone.

**(vii) Sliding-window / convolution-style aggregation loops.** In this scaffold, each constraint aggregates variables over a *window* whose range depends on the outer index. Unlike standard nested-loop aggregation with fixed summation ranges, the summation interval here moves with the index, resulting in a convolution-style pattern. A canonical form is:

$$\forall t \in T: \quad \sum_{s=t}^{t+W-1} x_s \leq K.$$

A representative example appears in `wachplan`, where workload is limited over any consecutive block of $W$ time slots. Because the planning horizon is cyclic, modular indexing is used to implement the sliding window:

$$\sum_{k=0}^{W-1} x_{i,((t+k-1) \bmod S)+1} \leq K, \quad \forall i \in I, \, \forall t \in T.$$

**(viii) Pairwise or all-pairs loop scaffolds (complete-graph indexing).** In this scaffold, constraint families are instantiated over *pairs* (or, more generally, tuples) of entities rather than over individual indices. While the underlying semantic constraint type often corresponds to conditional constraints implemented via binary variables and Big-$M$ formulations (as in conditional aggregation), instantiating such constraints over all relevant pairs induces a complete-graph–style indexing structure and a quadratic number of constraints.

Such pairwise scaffolds are common in scheduling, separation, and compatibility models. For example, in `supportcase33`, temporal separation between jobs belonging to different groups is enforced via Big-$M$ disjunctive constraints applied to each incompatible job pair $(i, j)$:

$$t_j \geq t_i + S - M(1 - y_{ij}), \qquad t_i \geq t_j + S - My_{ij},$$

where $y_{ij} \in \{0, 1\}$ selects the enforced ordering. Although these constraints fall under the same semantic category as conditional aggregation, their pairwise instantiation constitutes a distinct loop-based structural scaffold.

**(ix) Extra-dimension loop scaffolds (commodity or scenario replication).** In this scaffold, an entire constraint family is replicated over an additional index dimension, such as commodity, scenario, or skill type. Unlike nested-loop aggregation, the extra dimension does not alter the structure of individual constraints; instead, it produces multiple identical copies of the same constraint family, one for each value of the added index.

A canonical example is multi-commodity flow conservation `MCND`, where standard flow balance constraints are instantiated separately for each commodity. For each commodity $k \in \mathcal{K}$ and node $i \in \mathcal{N}$, flow conservation is enforced as:

$$\sum_{j:(i,j)\in\mathcal{A}} x_{ij}^k - \sum_{j:(j,i)\in\mathcal{A}} x_{ji}^k = b_i^k, \quad \forall k \in \mathcal{K}, \, \forall i \in \mathcal{N}.$$

Here, the commodity index $k$ serves as a replication layer rather than an aggregation index, resulting in a stacked family of otherwise identical constraints.

In summary, our analysis shows that loop structure in practical MILP models extends far beyond the distinction between single and nested iteration. While many constraints share simple semantic forms, their instantiation through cyclic or modular indexing, sliding-window aggregation, pairwise (complete-graph) generation, and extra-dimension replication fundamentally changes how constraint families scale and interact. These structural scaffolds therefore constitute a critical dimension of modeling complexity that is not captured by semantic constraint types alone. We summarize the interaction between semantic constraint families and loop-based structural scaffolds in Table 6.

### D.3. Empirical Observations from MIPLIB-NL

Analyzing over 200 reverse-generated instances derived from MIPLIB 2017, we observe several consistent structural patterns that are not captured by semantic constraint types alone.

*Table 6.* Loop-based structural scaffolds observed in MIPLIB-derived instances. Each row corresponds to one loop-based scaffold type (i)–(ix), which characterizes how constraint families are instantiated, replicated, and coupled through indexing and aggregation patterns in practical MILP models. The column "Typical constraint families" refers to atomic/semantic constraint categories (as defined in Table 5), while "Typical applications and dataset instances" describe problem-level modeling contexts empirically observed in our collected instances.

| Scaffold type | Typical constraint families | Canonical form | Structural role | Typical applications and dataset instances |
|---|---|---|---|---|
| (i) Global (non-indexed) | global budget, global coupling | $a^\top x \le b$ | one-shot coupling, no indexed repetition | single-period budget-constrained optimization, global feasibility linking across decisions ▷ **Instances:** `mik-250-20-75-1, mik-250-20-75-2, mik-250-20-75-3, mik-250-20-75-4, mik-250-20-75-5` |
| (ii) Single-loop with inner aggregation | assignment, covering, node-wise balance | $\forall i : \sum_{j \in J(i)} a_{ij} x_{ij} \ (\le, \ge, =) \ b_i$ | core repeated rule, indexed by $i$, inner aggregation over $j$ | bipartite and single-index assignment problems, facility coverage and service placement models, node-based constraints in network formulations ▷ **Instances:** `rmatr100-p5, rmatr200-p10, rmatr200-p20, bnatt400, bnatt500, assign1-5-8, netasgn_parsed, air03, air04, air05, v150d30-2hopcds` |
| (iii) Nested-loop (multi-index) | multi-index capacity, multi-period balance, time-indexed constraints | $\forall (i,t) : \sum_j a_{ijt} x_{ijt} \le b_{it}$ | high-dimensional constraint blocks, coupling across multiple indices | multi-period production and inventory planning, job–mode–time scheduling models, spatio-temporal resource allocation ▷ **Instances:** `30n20b8, exp-1-500-5-5, prod1, prod2, decomp1, decomp2` |
| (iv) Subset-indexed (implicit-set) loops | set cover / multi-cover, clique packing, conflict constraints | $\forall k \in \mathcal{K} : \sum_{j \in S_k} x_j \ (\le, \ge) \ b_k$ | irregular and overlapping index sets, constraints defined over subsets | set covering and hitting-set formulations, conflict graph and compatibility selection models, neighborhood-based combinatorial optimization ▷ **Instances:** `stein9inf, stein15inf, seymour1, iis-hc-cov, iis-glass-cov, cvs08r139-94, cvs16r106-72, cvs16r128-89, cvs16r70-62, cvs16r89-60` |
| (v) Coupled / recursive temporal loops | aggregation definition, smoothing / ramping | $\forall t : \ H_t = \sum_i a_i x_{i,t}; \ \forall \ t \ge 2 : H_t \ge \alpha H_{t-1}$ | inter-period dependency, explicit temporal coupling | multi-period planning with temporal coupling, inventory and harvest smoothing models ▷ **Instances:** `gmu-35-40, gmu35-50, exp-1-500-5-5, flugpl, flugplinf` |
| (vi) Cyclic / modular-index loops | wrap-around adjacency, periodic feasibility masking | $\forall t : \ x_t + x_{(t \bmod S)+1} \le 1,$ or $\forall t : \ x_t = 0$ if $(t \bmod P) \ge L$ | cyclic adjacency relations, periodic activation rules | cyclic workforce and roster planning, periodic scheduling with wrap-around constraints ▷ **Instances:** `wachplan, timtab1, 30n20b8` |
| (vii) Sliding-window / convolution-style aggregation | rolling-horizon capacity, windowed workload limits | $\forall t : \sum_{k=0}^{W-1} x_{t+k} \le K$ | moving aggregation window, index-dependent summation range | rolling-horizon workload regulation, consecutive-activity feasibility constraints ▷ **Instances:** `wachplan, 30n20b8` |
| (viii) Pairwise / all-pairs (complete-graph) loops | disjunctive ordering, separation constraints | $\forall (i,j) : t_j \ge t_i + S - M(1 - y_{ij})$ | quadratic constraint families, pairwise coupling dominates size | disjunctive and separation-based scheduling, pairwise compatibility and ordering models ▷ **Instances:** `supportcase21, supportcase21i, supportcase26, supportcase33` |
| (ix) Extra-dimension replication | multi-commodity flow, replicated constraint families | $\forall k, i : \sum_{(i,j)} x_{ij}^k - \sum_{(j,i)} x_{ji}^k = b_i^k$ | stacked copies of identical constraints, replication over extra dimensions | multi-commodity network design, commodity-indexed planning formulations ▷ **Instances:** `MCND, SupplyChain, StaticLinePlanning` |

**Summation is predominantly instantiated through indexed constraint families.** Across the dataset, constraints involving summation rarely appear as isolated, global aggregates. Instead, summation almost always occurs within loop-based constraint families, where a single semantic rule is instantiated repeatedly over an index set. Typical examples include assignment constraints indexed by jobs or agents, capacity constraints indexed by resources or subsets, and flow conservation constraints indexed by nodes. This observation indicates that the modeling challenge lies not in recognizing the presence of a sum, but in correctly identifying the index structure that governs its repetition.

**Nested and subset-indexed loops are pervasive, even in compact models.** We further observe that many instances exhibit nested-loop or subset-indexed constraint families, even when the high-level problem description appears concise. For example, scheduling and production planning models often combine job, mode, and time indices, leading to multi-dimensional constraint blocks, while graph-based models generate constraints over implicitly defined subsets such as conflict cliques or neighborhoods. These patterns demonstrate that structural complexity arises from index interactions rather than from the sheer number of semantic constraint types.

**Aggregation frequently introduces intermediate variables with cross-index coupling.** In a large fraction of instances, aggregation constraints are used to define intermediate quantities—such as inventory levels, workload measures, or harvest volumes—that are subsequently coupled across indices or time periods. These couplings induce recursive or temporal dependency structures, as seen in multi-period planning, smoothing, and rolling-horizon formulations. Such dependencies fundamentally differ from the flat constraint structures found in toy benchmarks and require models to reason about the

interaction between multiple constraint families.

Taken together, these observations highlight a systematic gap between semantic constraint classifications and the structural realities of industrial MILP models. While semantic types describe the algebraic form of individual constraints, they substantially underestimate the role of indexed repetition, aggregation, and inter-family coupling. This gap motivates our explicit treatment of loop-based structural scaffolds and their integration with application-level taxonomy in the subsequent analysis.

### D.4. Augmented Application Taxonomy

While the preceding subsection highlights the importance of loop-based structural scaffolds for dataset design and evaluation, an additional challenge emerges at the level of *application taxonomy*. Most existing NL4Opt-style benchmarks organize instances using a small set of high-level, application-driven categories (e.g., assignment, network flow, and scheduling). Although effective for textbook-scale problems, these taxonomies become insufficient when applied to heterogeneous, benchmark-derived MILP collections.

In constructing our MIPLIB-derived dataset, we therefore preserve the original nine application classes proposed by Lu et al. (2025) as a stable semantic backbone, but observe that a non-trivial subset of instances cannot be faithfully categorized within these classes without substantial loss of interpretability. Importantly, these instances are not anomalous; rather, they correspond to well-established optimization problem families that are underrepresented or entirely absent in existing NL-to-Opt benchmarks.

To address this gap, we introduce an *augmented application taxonomy* (Table 4) that extends the original nine classes with several additional main categories. These new categories are introduced conservatively and only when a recurring structural and semantic pattern cannot be adequately captured by the original taxonomy.

**Graph-Structured Optimization.** This class captures instances whose defining structure is induced by an explicit graph, including graph coloring, clique or conflict packing, and dominating set formulations. While such problems can often be expressed using set packing or covering constraints, their constraint families are generated by graph adjacencies or conflict cliques rather than by externally specified resources. Consequently, these instances are dominated by subset-indexed and pairwise loop-based scaffolds, which differ structurally from the capacity-driven constraints typical of classical combinatorial optimization problems.

**Disjunctive and Pairwise Scheduling.** Several instances enforce feasibility through large collections of pairwise disjunctive or separation constraints, commonly encoded using Big-$M$ formulations. Unlike canonical job shop or flow shop models, these problems are characterized by complete-graph or near-complete-graph indexing over task pairs, leading to $\mathcal{O}(n^2)$ constraint families. We therefore distinguish them from standard production and scheduling models to explicitly highlight their dominant all-pairs loop structure.

**Random-Graph Benchmark MILPs.** This category groups instances generated from random graph constructions that serve primarily as structural benchmarks. Such models typically involve covering, packing, or feasibility constraints defined over randomly generated adjacency sets. Although they lack a clear real-world application narrative, their highly irregular and implicitly defined index sets make them valuable for analyzing how systems handle large, unstructured loop scaffolds. We separate these instances to distinguish benchmark-driven graph structure from application-driven graph models.

**Synthetic / Prototype MILPs.** This class contains synthetic or prototype instances that are manually designed or algorithmically generated to probe modeling or solver behavior. These instances are common in benchmark collections and often serve as minimal or illustrative examples rather than representations of real-world decision problems. By explicitly identifying them as synthetic, we avoid conflating benchmark artifacts with application-driven optimization tasks.

**Dense Algebraic / Stress-Test MILPs.** Dense algebraic and stress-test instances are characterized by large, dense constraint matrices with limited semantic interpretability. Their primary purpose is to stress-test optimization algorithms under extreme numerical or combinatorial conditions. From a structural perspective, these models exhibit highly repetitive linear constraint families without a clear application-level decomposition, motivating their separation into a distinct application class.

**Miscellaneous Application Stubs.** Finally, a small number of instances correspond to narrow or idiosyncratic application settings, such as sports or tournament scheduling, that do not naturally align with the dominant application classes. We group these instances into a lightweight miscellaneous category to minimize the use of an undifferentiated *Others* class while avoiding unnecessary fragmentation of the taxonomy.

Overall, the augmented application taxonomy provides a faithful organizational scheme for a diverse, MIPLIB-derived dataset without collapsing structurally distinct problems into overly broad categories. When combined with the loop-based scaffold analysis, it enables a two-dimensional view of dataset coverage in which each instance is characterized jointly by its application class and its dominant loop-based structural patterns.

### D.5. Implications for Dataset Analysis

The prevalence of loop-based structural scaffolds has direct implications for the design and evaluation of NL-to-Opt benchmarks. Datasets that emphasize semantic constraint types while flattening or omitting indexed and aggregated constraint families allow models to succeed through surface-level pattern matching, without requiring robust reasoning over repetition, indexing, or inter-constraint dependencies.

Our analysis shows that, in realistic MILP models, structural complexity arises primarily from how constraints are instantiated and coupled through loops, rather than from the diversity of algebraic forms alone. By explicitly preserving loop-based scaffolds and organizing instances using an augmented application taxonomy, our dataset enables structure-aware analysis that more faithfully reflects real-world modeling practice.

We therefore advocate treating loop-based structural scaffolds as a first-class analytical dimension—complementary to semantic constraint types—when designing, curating, and evaluating future NL-to-Optimization benchmarks.

### D.6. Loop Abstraction Examples

This subsection provides illustrative examples that concretely demonstrate how the abstract notions of *constraint families* and *loop-based structural scaffolds* introduced above manifest in large, expanded LP/MPS formulations. Rather than introducing new benchmark instances, the goal of these examples is to show how complex collections of algebraic constraints can be systematically compressed into a small number of loop-based structural templates that preserve the essential modeling logic.

**Example 1: Multiperiod Production Planning**

A multiperiod production planning model with products $i \in \mathcal{I}$, facilities $j \in \mathcal{J}$, and time periods $t \in \mathcal{T}$ typically expands to thousands of algebraic constraints when represented in LP or MPS form. Structure extraction from such formulations commonly reveals a small number of recurring loop-based constraint families, including:

- Inventory balance loops indexed by $(i, t)$, corresponding to flow or balance-type constraint families.
- Capacity constraint loops indexed by $(j, t)$, capturing resource limitations over facilities and time.
- Demand satisfaction loops indexed by $(i, t)$, enforcing service or coverage requirements.

At the modeling level, these constraint families can be expressed compactly in natural language (e.g., "for each product and each time period, inventory evolves according to production minus demand"), without enumerating the expanded rows that appear in the LP/MPS representation.

**Example 2: Capacitated Facility Location**

A capacitated facility location instance with candidate facilities $j \in \mathcal{J}$ and customers $i \in \mathcal{I}$ is naturally structured around a small number of loop-based scaffolds:

- Variable groups corresponding to facility opening decisions $y_j$ and customer assignment decisions $x_{ij}$.
- Assignment constraint families indexed by $i \in \mathcal{I}$, ensuring that each customer is served.
- Capacity constraint families indexed by $j \in \mathcal{J}$, limiting the total demand assigned to each open facility.

Even when the Cartesian product $|\mathcal{I}| \times |\mathcal{J}|$ is large, the underlying loop-based structural scaffold remains compact and serves as the conceptual backbone for both model construction and natural-language descriptions.

## E. Summary of human-LLM interaction

This appendix provides further details regarding the expert-driven human–LLM interaction methodology outlined in Section 3.4. We present specific techniques and illustrative examples concerning interactive observation and interactive

*Table 7.* Augmented application taxonomy with dominant semantic constraint types and loop-based structural scaffolds. We keep the original nine main classes unchanged and add additional classes to accommodate MIPLIB-derived instances.

| Main Class | Problem Class | Instances | Dominant Semantic Constraint Types | Dominant Loop-Based Structures |
|---|---|---|---|---|
| Assignment and Resource Allocation Optimization | Car Selection Problem | – | Upper bound (weighted sum) | Single-loop; subset-indexed |
| | Contract Allocation Problem | – | Upper bound (weighted sum) | Single-loop; subset-indexed |
| | Assignment Problem | `assign1-5-8, netasgn_parsed, bnatt400` `bnatt500, rmatr100-p5, rmatr100-p10` `rmatr200-p5, rmatr200-p10, rmatr200-p20` `f2gap40400, f2gap201600, f2gap801600` `f2gap401600` | Exactly-one; At-most-one | Single-loop + inner sum |
| | Structure-Based Assignment Problem | – | Exactly-one; implication (Big-$M$) | Single-/subset-indexed |
| | Team Formation Problem | – | Lower bound (sum); At-most-one | Subset-indexed |
| | Military Personnel Deployment Problem | – | Upper bound (sum) | Nested-loop |
| Combinatorial Optimization | Knapsack Problem | `mik-250-20-75-1, mik-250-20-75-2, mik-250-20-75-3` `mik-250-20-75-4, mik-250-20-75-5` | Upper bound (weighted sum) | Single-loop (items) |
| | Market Share Optimization Problem | `markethare4, markshare_5_0` | Proportion constraints | Coupled-loop (ratio) |
| | Set Multi-Cover Problem | `iis-hc-cov, iis-glass-cov, beasleyC3` | Lower bound (sum) | Subset-indexed |
| | Set Cover Problem | `stein9inf, stein15inf, stein45inf` `seymour1` | Lower bound (sum) | Subset-indexed |
| Cutting and Packing Optimization | Bin Packing Problem | `bppc4-08, bppc6-02, bppc6-06` `bppc8-02, bppc8-09` | Upper bound (weighted sum) | Single-loop; nested variants |
| | Blending Problem | – | Equality + bounds | Single-loop |
| | Cutting Stock Problem | `glass-sc` | Upper bound (weighted sum) | Subset-indexed |
| Domain-Specific Optimization | Diet Problem | – | Upper/lower bounds (weighted sums) | Single-loop |
| | Unit Commitment Problem | `thor50dday` | Implication (Big-$M$); bounds | Coupled temporal loops |
| | Farm Planning Problem | `gmu-35-40, gmu35-50` | Aggregation definition; bounds | Coupled / recursive loops |
| Facility Location Optimization | Facility Location Problem | – | Implication (open/assign); bounds | Single-loop + inner sum |
| | Capacitated Facility Location Problem | – | Upper bound (weighted sum) | Nested-loop (facility×demand) |
| | Transportation Problem (Airline Resource Allocation) | `air03, air04, air05` | Flow/assignment constraints | Network-indexed; single-loop |
| | Facility Dispersion Problem | – | At-most-one | Subset-indexed |
| Financial and Revenue Optimization | Portfolio Optimization Problem | – | Proportion; weighted sums | Coupled-loop |
| | Profit Maximization Problem | – | Upper/lower bounds | Single-loop |
| | Revenue Management Problem | – | Capacity constraints | Nested-loop |
| | Revenue Maximization Problem | – | Proportion constraints | Coupled-loop |
| Network Flow Optimization | Multi-Commodity Capacitated Network Design Problem | MCND | Flow conservation; capacity | Extra-dimension (commodity) + network loops |
| | Multi-Commodity Transportation Problem | – | Flow conservation | Extra-dimension loop |
| | Minimum Cost Flow Problem | `network_flow` | Flow conservation | Single-loop (nodes) |
| | Multi-Commodity Network Flow Problem | MCND | Flow conservation | Extra-dimension loop |
| | Network Flow Problem | `neos-1425699, neos5` | Flow conservation | Single-loop; dual sums |
| | Static Line Planning Problem | `StaticLinePlanning` | Balance / coupling | Coupled-loop |
| | Supply Chain Optimization | `SupplyChain` | Flow + capacity + coupling | Nested + coupled loops |
| | Network Optimization | – | Mixed flow/capacity | Nested-loop |
| Production Planning and Scheduling Optimization | Capacitated Lot-Sizing Problem | `prod1, prod2` | Capacity (weighted sum); bounds | Nested (item×time) |
| | Factory Planning Problem | `decomp1, decomp2` | Aggregation definition; bounds | Coupled loops |
| | Flow Shop Scheduling Problem | – | Ordering / separation | Pairwise loops |
| | Job Shop Scheduling Problem | `30n20b8` | Exactly-one; capacity bounds | Nested + sliding-window-like |
| | Discrete Lot-Sizing and Scheduling Problem | – | Coupling + capacity | Nested + coupled |
| | Production Planning Problem | `exp-1-500-5-5` | Aggregation + coupling | Temporal coupled loops |
| | Lot-Sizing Problem | `exp-1-500-5-5` | Aggregation + coupling | Temporal coupled loops |
| Transportation and Routing Optimization | Aircraft Assignment Problem | `air03, air04, air05` | Assignment / flow | Network-indexed |
| | Aircraft Landing Problem | `flugpl, flugplinf` | Pairwise separation; bounds | All-pairs loop |
| | Transportation Problem | `rai1507, fast0507` | Flow balance | Network-indexed |
| | Traveling Salesman Problem | `TSP, ei133-2, ei1A101-2` `ei1C76-2` | Degree constraints; subtour cuts | Single-loop + subset-indexed |
| | Operations Optimization | `wachplan, timtab1, supportcase33` | Scheduling bounds; disjunction | Cyclic / sliding-window; pairwise |
| | Capacitated Vehicle Routing Problem with Time Windows | – | Time-window bounds; capacity | Nested temporal loops |
| Graph-Structured Optimization (new) | Graph Coloring / Chromatic Index | `chromaticindex32-8, chromaticindex128-5, chromaticindex256-8` `chromaticindex512-7, chromaticindex1024-7` | At-most-one; equality | Pairwise / subset-indexed |
| | Clique / Conflict Packing | `cvs08r139-94, cvs16r106-72, cvs16r128-89` `cvs16r70-62, cvs16r89-60` | At-most-one | Subset-indexed |
| | Graph Covering / Dominating Set | `v150d30-2hopcds` | Lower bound (sum) | Subset-indexed |
| Disjunctive and Pairwise Scheduling (new) | Pairwise Disjunctive Scheduling | `supportcase21i, supportcase26, supportcase33` | Implication (Big-$M$) | All-pairs loop |
| | Compatibility / Separation Scheduling | – | At-most-one; implication | Pairwise loop |
| Random-Graph Benchmark MILPs (new) | Random Graph Cover / Pack (structural) | `graph20-20-1rand, graph20-80-1rand, graph40-20-1rand` `graph40-40-1rand, graph40-80-1rand` | Covering / packing | Subset-indexed |
| | Generic Graph-Indexed MILPs | `pg, pk1, p2m2p1m1p0n100` | Mixed linear constraints | Irregular nested loops |
| Synthetic / Prototype MILPs (new) | Synthetic Examples / Prototypes | `ex9, ex10, prototype` | Mixed bounds; linear sums | Global / shallow loops |
| | Large Synthetic MILPs | `ex1010-pi` | Dense linear constraints | Nested-loop |
| Dense Algebraic / Stress-test MILPs (new) | Dense Constraint Matrices | `h50x2450, h80x6320d, ramos3` | Linear bounds; dense sums | Large single-loop |
| | Misc. Stress-test Instances | `cost266-UUE` | Mixed constraints | Irregular loops |
| Misc. Application Stubs (new) | Sports / Tournament Scheduling | `b-ball` | Scheduling bounds; disjunction | Pairwise loops |
| | Small Misc. MILPs | `50v-10, app1-1` | Mixed constraints | Shallow loops |
| Others | | – | – | – |

inspection, demonstrating how human intuition synergizes with the computational capabilities of Large Language Models (LLMs) to resolve complex semantic ambiguities.

### E.1. Interactive Observation

Interactive observation focuses on identifying latent data patterns through human–machine collaboration. While current LLM agents excel at routine statistical analysis and code-based visualization, human experts retain a distinct advantage in intuitive pattern recognition within complex, non-standard scenarios. In this workflow, rather than relying on the LLM to solve the problem autonomously, the human expert directs the investigation. The expert utilizes the LLM to rapidly generate scripts, check data ranges, verify symmetries, validate conjectures, and distill key information. This allows the expert to bypass implementation details and concentrate their cognitive effort on the core structural challenges of the problem.

A compelling example of this synergy is found in the analysis of problem `sorrell3`. The mathematical form of the problem is straightforward: it involves binary variables $x_i$ with $1 \leq i \leq 1024$. The objective is to maximize the sum of these variables, subject to a set of pairwise mutual exclusion constraints in the form $x_i + x_j \leq 1$. The semantic challenge lies entirely in identifying the rule governing which $(i, j)$ pairs are mutually exclusive. Understanding this rule is equivalent

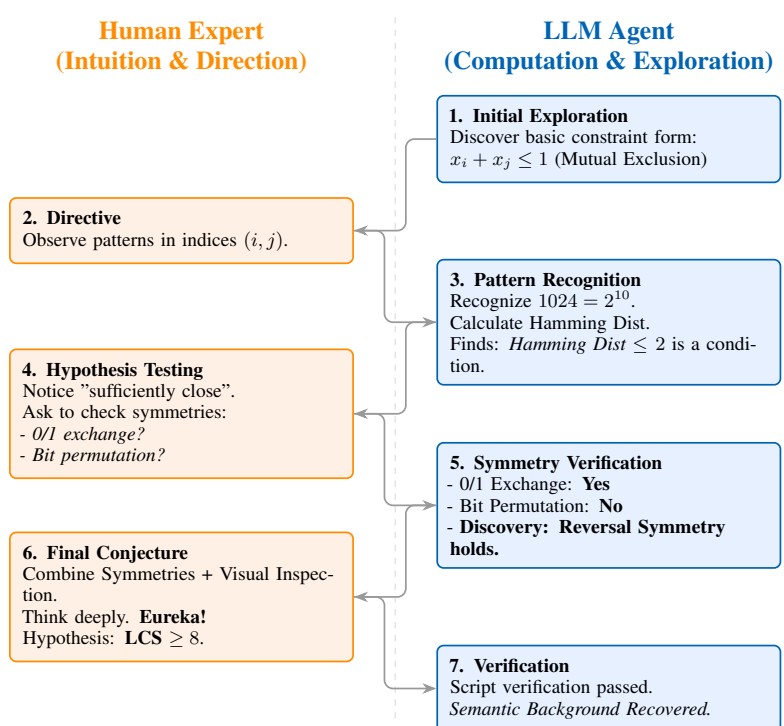

*Figure 16.* Interaction timeline for problem `sorrell3`. The process illustrates the synergy where the Human provides structural hypotheses (Symmetries, LCS) while the LLM performs verification and discovers hidden invariants (Reversal Symmetry).

to recovering the problem's background.

Initially, the LLM was allowed to explore the data freely. It quickly noted that $1024 = 2^{10}$ and hypothesized that the indices related to binary encoding. It then calculated the Hamming distances between constrained indices. The LLM independently discovered that constraints existed for all pairs with a Hamming distance of 1 or 2. This implied that any feasible solution (an independent set in the conflict graph) constituted an Error-Correcting Code with a minimum Hamming distance of at least 3. However, for pairs with Hamming distances greater than 2, the LLM failed to identify a clear pattern, producing only generic statistical summaries and visualizations.

At this stage, the human expert intervened. Realizing that the constrained $(i, j)$ pairs likely represented variables that were "similar" in some sequence-based metric, the expert instructed the LLM to write a script exporting the binary representations of these specific pairs to a CSV file for direct inspection. To narrow down the search space, the expert then guided the LLM to test various symmetry hypotheses: whether the structure remained isomorphic under 0/1 exchange (verified as true) and whether it remained isomorphic under bit permutations. The LLM reported that swapping arbitrary or even symmetric bit positions destroyed the isomorphism, but—through its own active exploration—discovered that reversing the sequence order preserved isomorphism. Based on these invariants (0/1 symmetry and reversal symmetry) and a visual inspection of the CSV data, the expert hypothesized that the constraints corresponded to pairs sharing a Longest Common Subsequence (LCS) exceeding a certain threshold. The expert then tasked the LLM with verifying this specific LCS hypothesis, which confirmed the pattern and successfully recovered the problem's semantic background.

### E.2. Interactive Inspection

As described in the main text, the primary validation loop involves the LLM generating solver code based on the inferred problem description. If the solver's result matches the ground truth (the optimal value from the original MPS file), the instance is validated. However, if the results diverge, the discrepancy may stem from an incorrect problem derivation or a bug in the solver code. In these cases, we employ interactive inspection to locate and correct the error. The core strategy relies on using contradictions as the breakthrough point rather than simply informing the LLM that an error exists.

For minimization problems (the logic is inverted for maximization), two distinct error modes exist. First, if the solver

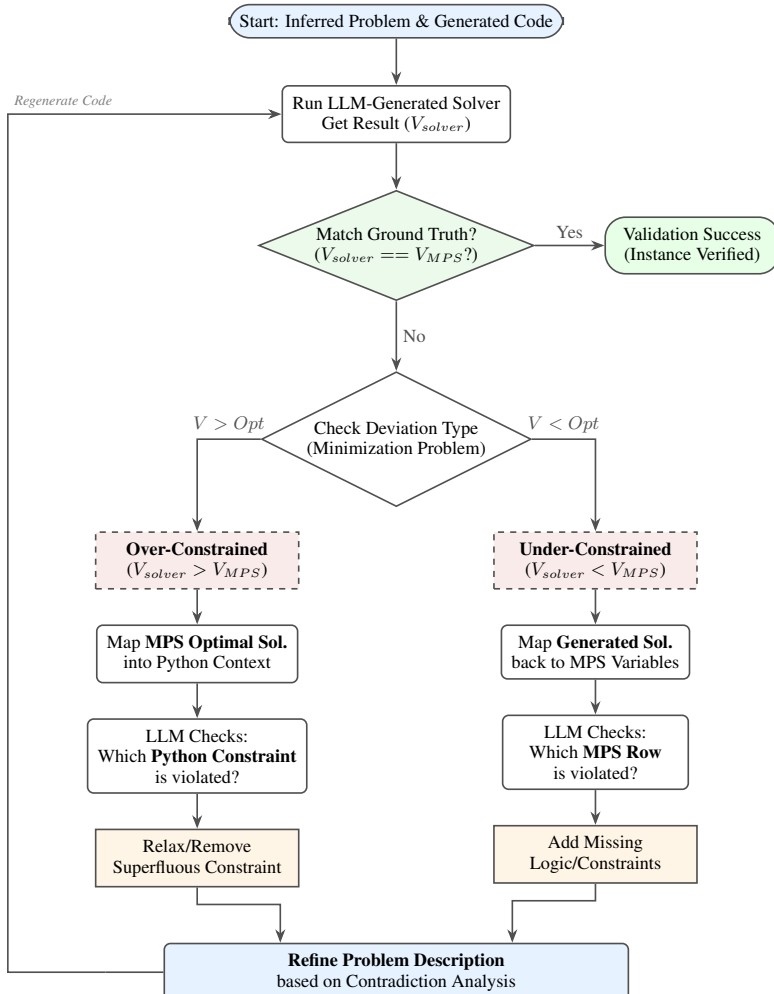

*Figure 17.* Workflow of the interactive inspection mechanism for minimization problems.

provides a solution strictly greater than the MPS optimal value, the constructed model is likely over-constrained (or the objective function is misaligned, though this is rare with state-of-the-art agents). This implies that the true optimal solution from the MPS file satisfies the physical constraints but violates the inferred constraints. In this scenario, we map the MPS solution into the context of the generated problem and use the LLM to check which specific constraint in the Python code is violated. This pinpoints exactly where the inferred logic is too strict. Second, if the solver provides a solution strictly less than the MPS optimal value, the constructed model is under-constrained. Here, the solution derived from the generated model is invalid in the original MPS context. We map the generated solution back to the MPS variables and use the LLM to identify which original MPS rows are violated, thereby locating the missing logic.

This technique was effectively applied to problem `30n20b8`. After deriving an initial problem description (minimizing cost) and generating the corresponding code, the solver produced a value lower than the MPS optimum. This indicated missing constraints. Upon checking the violated MPS rows, it was discovered that the initial abstraction ignored time periodicity; specifically, tasks could only be completed during specific windows within a cycle. The problem description was corrected accordingly. However, the subsequent run yielded a value higher than the MPS optimum, indicating that the new model was now over-constrained. We then "challenged" the LLM with the ground truth, effectively asking: "The ground truth solution achieves a lower cost; specifically which of your constraints does it fail to satisfy?" Assisted by the LLM's analysis of the specific violation, the expert identified that a misunderstanding of the problem context had led to the inclusion of a superfluous constraint. Once this extra restriction was removed, the solver output matched the ground truth, confirming the semantic correctness of the reverse-engineered instance.

### E.3. Handling infeasible and open instances.

A small subset of the constructed instances corresponds to MIPLIB models that are *infeasible* or remain *open*, for which no certified optimal objective value is available (8 infeasible and 15 open instances). For these cases, solver-level outcome matching is not applicable as a validation signal.

For the majority of these cases, solvable counterparts exist within the same problem family. Typically, instances with similar names share an identical underlying structure. The infeasible or open instances used for reconstruction were not arbitrarily selected; rather, they were processed using the same generation scripts applied to "easy" instances of the same class, but only after the easy instances were solved and verified to confirm structural equivalence. Consequently, the quality and reliability of these infeasible and open instances remain generally controllable.

Furthermore, regarding infeasible instances, the initial problem description aligns with that of a corresponding easy instance. Although subsequent polishing may introduce variations in the final text, this process is monitored and verified. Even in the unlikely worst-case scenario where subtle structural discrepancies prevent full equivalence, the generated problem remains a valid benchmark instance, resulting, at most, in a minor loss of dataset diversity rather than incorrectness. Regarding open instances, we observe that for many cases, the solver is able to locate the reported best-known solution within a 4-hour time limit with a minimal optimality gap. This serves as a crucial reference for validating the correctness of the problem reconstruction and the subsequent modeling process.

For instances that cannot be validated via the approaches described above, OR experts perform thorough manual validation to confirm that the instance specification is semantically complete and faithful to the original model. This includes verifying the correctness of the recovered structure, variable roles, constraint families, data bindings, and objective intent, ensuring that the instance is well-posed in principle even though optimality cannot be numerically certified.

These instances are retained in the released benchmark to reflect the realistic composition of industrial optimization repositories. However, because standard quantitative metrics rely on comparing solver outcomes, we exclude infeasible and open instances from the experimental evaluation reported in Section 4.

## F. Experimental Setup Details

This appendix provides the full experimental setup corresponding to the results reported in Section 4. The goal of the experiments is not to propose new modeling algorithms, but to systematically stress-test existing NL-to-Opt systems under realistic data conditions enabled by MIPLIB-NL.

### F.1. Evaluated Datasets

We evaluate all methods on a mix of existing benchmarks and our newly introduced datasets, covering a wide spectrum of problem scales and structures.

**Existing benchmarks.** We include the following representative datasets: (i) the *LLM4OR Collection* (Xiao et al., 2025), which aggregates and cleans several widely used benchmarks (NL4Opt, IndustryOR, MAMO-EasyLP, MAMO-ComplexLP, NLP4LP, ReSocratic) and largely reflects the current toy- and small-scale evaluation regime; (ii) *OptMath* (Lu et al., 2025), consisting of 166 curated NL-to-Opt instances; (iii) Bench4Opt (Wang et al., 2025), which contains 394 problems (roughly half in unstructured natural language) and adopts a model–data separation format. As many instances in this dataset were originally formulated as infeasible, we omitted them to focus our benchmarking on solvable cases; and (iv) LogiOR (Yang et al., 2025a), a logistics-focused benchmark with 92 instances that is unlikely to appear in the training data of existing fine-tuned models.

**A new benchmark.** We evaluate on our proposed dataset MIPLIB-NL, consisting of 223 instances faithfully reverse-generated from MIPLIB 2017. This dataset enables evaluation from toy-scale problems to genuinely industrial-scale optimization models with up to $10^7$ variables and constraints.

### F.2. Systems and Baselines

We evaluate two broad classes of systems.

**Fine-tuned models.** We benchmark leading fine-tuned models for optimization modeling, including OptMath-Qwen2.5-7B,

OptMath-Qwen2.5-32B (Lu et al., 2025), ORLM-LlaMa3-8B (Huang et al., 2025a), SIRL-Qwen2.5-7B and SIRL-Qwen2.5-32B (Chen et al., 2025). These models represent the strongest fine-tuned approaches.

**Bare LLMs.** To isolate base model capability from algorithmic scaffolding, we evaluate bare LLMs prompted directly to generate mathematical formulations and solver code, without multi-agent workflows or external tools. We include both open-source and closed-source models under a unified protocol: DeepSeek-V3.2, DeepSeek-V3.2-Think, Qwen3-Max (No-Thinking), Qwen3-Max-Preview (Thinking), Gemini-3-Pro-Preview, Claude-Sonnet-4.5 (No-thinking), Claude-Sonnet-4.5 (Thinking), GPT-5.1, and GPT-5.1 CodeX.

### F.3. Evaluation Metrics

We report two complementary metrics.

**Pass@N accuracy.** An instance is considered solved if the solver-executed objective value of the generated model matches the ground-truth optimum within a relative error tolerance of $1 \times 10^{-6}$ in at least one of $N$ (we consider $N = 1$ and $N = 8$) independent trials. For the generation of solutions, we set the LLM temperature to 0.6. For large-scale problems with high solver cost, we apply a fixed solver time budget and record solver status accordingly.

**Solver code executability.** We report the fraction of instances for which the generated solver code executes successfully and produces a feasible solution, providing a coarse but robust signal of end-to-end usability.

## G. Error Mode Analysis

We analyze error modes observed in the MIPLIB-NL dataset. The errors are categorized into three main types: Compile Error, Modeling Error, and Time Limited Error.

**Fine-grained subtype analysis.** To complement the coarse error categories in Figure 7, we further decompose failures into operational diagnostic subtypes. Modeling errors are divided into *Variable Domain / Type* errors, *Constraint / Global Logic* errors, *Data / Index Coupling* errors, *Objective / Bound / Coefficient* errors, and *Template / Incomplete Model* errors. Execution errors are divided into *Python / API* errors, *Data Binding* errors, *Solver Status* failures, *Environment / Resource* errors, and *Other Runtime* errors. These subtypes are defined from the prompt, generated code, execution outcome, and diagnostic traces, and are used as comparative categories for understanding where generated optimization models fail.

The across-benchmark comparison in Figure 18 shows that MIPLIB-NL is not simply a harder version of existing datasets. Prior benchmarks are often dominated by relatively local failures, such as wrong variable domains or missing global constraints. In contrast, failures on MIPLIB-NL are dominated by the need to preserve heterogeneous data relations, index mappings, and cross-entity coupling in a full solver-executable model. This pattern is exactly what we expect from industrial-scale optimization instances whose difficulty lies in recovering structured model–data bindings rather than isolated constraint templates.

The relative-scale analysis in Figure 19 provides a complementary within-MIPLIB-NL view. Larger MIPLIB-NL instances do not merely increase the overall failure rate; they change the composition of failures. For execution errors, small relative-scale instances are more affected by data-binding failures, whereas medium to very large instances are increasingly dominated by environment/resource failures. For modeling errors, variable-domain/type mistakes remain common, but their share decreases at the largest scale, while data/index coupling and incomplete-model failures become more prominent. The relative-scale subtype shifts are statistically significant, indicating that scale changes the nature of the modeling and execution bottlenecks exposed by MIPLIB-NL.

---

**Compile Error: Steiner Tree Network Design (MIPLIB-NL-thor50dday)**

**Problem**: Find the minimum cost network to connect 50 priority hubs within a set of 231 cities.
**Code Snippet**:

```
# Flow conservation constraint construction
flow_balance = gp.QuickSum(f[i, j] for j in adj[i]) - gp.QuickSum(f[j, i]
for j in adj[i])
```

---

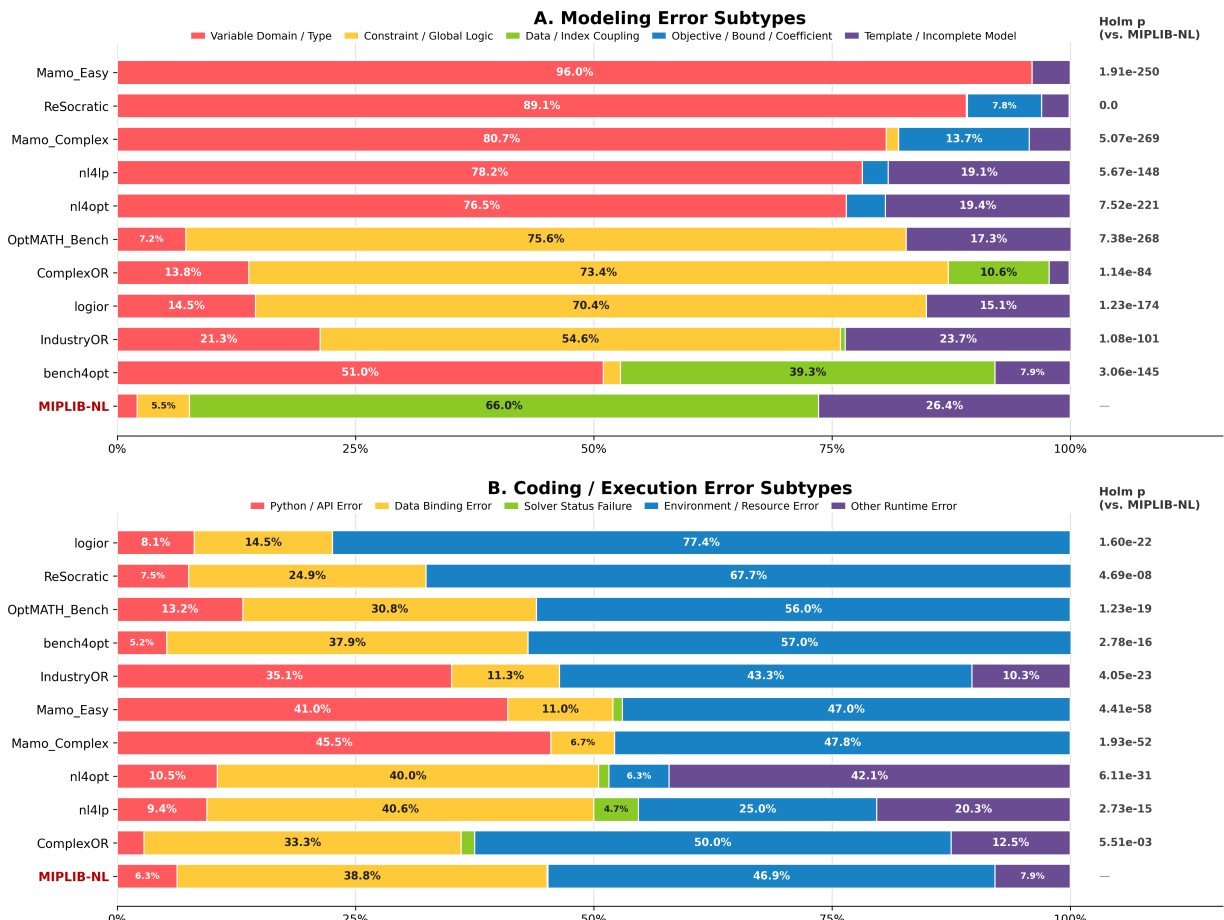

*Figure 18.* Fine-grained error subtype distributions across benchmarks. The top panel compares modeling-error subtypes, and the bottom panel compares coding/execution-error subtypes. Compared with prior benchmarks, MIPLIB-NL exhibits a markedly different modeling-error profile: errors concentrate much more strongly in data/index coupling and incomplete-model recovery, rather than in basic variable-domain or global-constraint mistakes. Pairwise distributional tests between MIPLIB-NL and each prior benchmark are statistically significant for both modeling and execution subtype distributions, with the strongest contrast on modeling errors.

```
if i == root:
    model.addConstr(flow_balance == M, name=f"flow_bal_root_{i}")
```
**Error**: AttributeError: module 'gurobipy' has no attribute 'QuickSum'. Did you mean: 'quicksum'?
**Analysis**: The LLM incorrectly capitalized the Gurobi summation function. The correct method in the `gurobipy` library is `quicksum` (lowercase), whereas the generated code used `QuickSum`. This demonstrates an API hallucination where the model applies CamelCase naming conventions to a library that uses lowercase conventions.

**Modeling Error: Robotic Arm Scheduling (MIPLIB-NL-supportcase33)**

**Problem**: Schedule 2 robotic arms to process a set of parts within specific time windows to maximize total profit. Constraints include fixed processing times per arm, sequence-dependent setup times, mutual exclusions between specific parts, and a strict maximum time gap between consecutive operations on the same part.
**Result**: Model Objective: 310.0 (Optimal: 345.0).
**Analysis**: The model produced a suboptimal solution (lower profit than optimal). The error lies in the variable

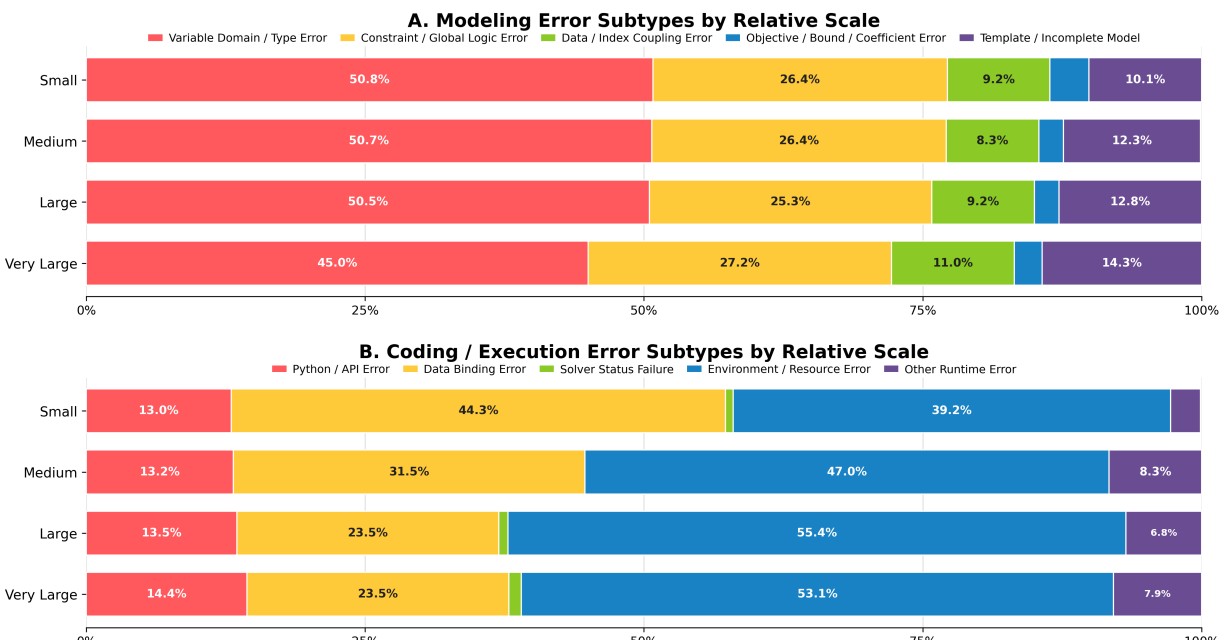

*Figure 19.* Fine-grained error subtype distributions across relative scale buckets within MIPLIB-NL. Scale buckets are computed only over MIPLIB-NL instances, so *Very Large* means large relative to other MIPLIB-NL instances rather than globally largest across all benchmarks. The top panel shows modeling-error subtypes and the bottom panel shows coding/execution-error subtypes. As relative scale increases, execution failures shift from data-binding issues toward environment/resource bottlenecks, while modeling failures increasingly involve data/index coupling and incomplete model recovery.

---

initialization phase where the model applied an incorrect graph pruning heuristic. Specifically, it explicitly forbade flow variables (arcs) between operation nodes of the same part unless they were immediately consecutive indices (allowing $k \to k+1$ but blocking $k \to k+2$). This prevents valid schedules where a single arm performs non-consecutive operations on the same part (e.g., Arm A doing pass 1 and 3, while Arm B handles pass 2), thereby restricting the search space and missing the optimal configuration.

---

**Time Limited Error: Researcher Assignment (MIPLIB-NL-cvs16r128-89)**

**Problem**: Assign 128 researchers from 89 project groups to 16 secure labs (capacity 9) such that members of the same group are not split across different labs.
**Result**: Timeout (Time Limit Exceeded).
**Analysis**: The problem involves a complex assignment with group consistency constraints ("all members of a group must be in the same lab or public hall"). The generated MIP formulation introduced a large number of binary variables ($128 \times 16$ assignment variables plus auxiliary variables) and constraints. The solver failed to prove optimality within the time limit, likely due to the weak LP relaxation or the symmetry in the problem formulation (identical labs). This illustrates the difficulty of generating efficient formulations for combinatorial problems with complex logical grouping constraints.

## H. Full Experimental Results and Analysis

Tables 1, 8, and 9 report the full experimental results for Pass@1 accuracy, Pass@8 accuracy, and execution-level failure rates ($N$=1), respectively. Several consistent trends emerge. We organize the analysis into five points: transfer from prior benchmarks, the effect of scaling and specialization, execution-level failures, Pass@8 recoverability, and an additional held-out fine-tuning check.

*Table 8.* Pass@8 Accuracy across datasets (%).

| | Method | NL4Opt | MAMO-E | MAMO-C | ComplexOR | IndustryOR | NLP4LP | ReSocratic | OptMath | Bench4Opt | LogiOR | MIPLIB-NL(Ours) |
|---|---|---|---|---|---|---|---|---|---|---|---|---|
| **Fine-Tuned Models** | OptMATH-Qwen2.5-7B | 85.51 | 98.72 | 57.66 | 50.00 | 38.10 | 91.01 | 80.65 | 31.33 | 67.05 | 16.30 | 3.00 |
| | OptMATH-Qwen2.5-32B | 32.24 | 99.08 | 54.05 | 50.00 | 26.19 | 96.63 | 87.10 | 57.83 | 78.41 | 48.91 | 12.00 |
| | SIRL-Qwen2.5-7B | 87.38 | 98.90 | 88.29 | 44.44 | 57.14 | 97.75 | 90.57 | 37.95 | 73.30 | 31.52 | 0.48 |
| | SIRL-Qwen2.5-32B | 87.85 | 99.08 | 92.79 | 44.44 | 73.81 | 98.31 | 96.53 | 47.59 | 79.55 | 55.43 | 11.90 |
| | ORLM-LlaMa3-8B | 94.86 | 95.23 | 66.67 | 0.00 | 73.81 | 97.75 | 82.63 | 19.88 | 68.18 | 25.00 | 1.90 |
| **Bare LLMs (Direct Prompting)** | DeepSeek-V3.2-685B | 87.38 | 97.80 | 84.68 | 61.11 | 83.33 | 95.51 | 93.30 | 63.86 | 84.66 | 76.09 | 23.81 |
| | DeepSeek-V3.2-Think-685B | 86.45 | 97.43 | 76.58 | 55.56 | 88.10 | 96.07 | 93.30 | 60.84 | 80.11 | 75.00 | 33.33 |
| | Qwen3-Max (No-Thinking) | 84.58 | 96.88 | 67.57 | 61.11 | 76.19 | 94.94 | 91.81 | 56.02 | 77.84 | 75.00 | 23.33 |
| | Qwen3-Max-Preview (Thinking) | 84.11 | 97.61 | 62.16 | 61.11 | 76.19 | 96.07 | 91.32 | 62.05 | 82.37 | 70.65 | 27.62 |
| | Claude-Sonnet-4.5 (No-Thinking) | 83.17 | 98.53 | 59.46 | 55.56 | 76.19 | 93.82 | 84.86 | 49.40 | 71.02 | 76.09 | 25.71 |
| | Claude-Sonnet-4.5 (Thinking) | 82.71 | 98.90 | 63.06 | 55.56 | 76.19 | 94.38 | 84.86 | 50.60 | 73.30 | 72.83 | 32.38 |
| | Gemini-3-Pro-Preview | 87.38 | 89.91 | 73.87 | 61.11 | 66.67 | 92.13 | 83.62 | 54.21 | 75.57 | 75.00 | 39.52 |
| | GPT-5.1 | 85.98 | 97.61 | 77.48 | 66.67 | 85.71 | 95.51 | 93.05 | 53.01 | 83.52 | 75.00 | 42.38 |
| | GPT-5.1-CodeX | 81.31 | 97.80 | 68.47 | 66.67 | 83.33 | 95.51 | 91.81 | 52.41 | 84.09 | 76.09 | 28.75 |
| | Avg | 82.21 | 97.39 | 70.91 | 52.38 | 70.07 | 95.38 | 88.96 | 49.78 | 77.07 | 60.64 | 21.87 |

*Table 9.* Overall error rate across datasets for $N = 1$ (%).

| | Method | NL4Opt | MAMO-E | MAMO-C | ComplexOR | IndustryOR | NLP4LP | ReSocratic | OptMath | Bench4Opt | LogiOR | MIPLIB-NL(Ours) |
|---|---|---|---|---|---|---|---|---|---|---|---|---|
| **Fine-Tuned Models** | OptMATH-Qwen2.5-7B | 4.67 | 0.73 | 3.60 | 22.22 | 42.86 | 3.37 | 8.44 | 34.94 | 32.74 | 36.96 | 90.48 |
| | OptMATH-Qwen2.5-32B | 3.27 | 1.47 | 1.80 | 50.00 | 11.90 | 2.81 | 4.47 | 18.67 | 12.44 | 20.65 | 44.76 |
| | SIRL-Qwen2.5-7B | 0.00 | 0.00 | 4.50 | 55.56 | 14.29 | 0.00 | 1.74 | 25.90 | 20.05 | 34.78 | 96.97 |
| | SIRL-Qwen2.5-32B | 0.00 | 0.00 | 0.90 | 38.89 | 4.76 | 0.56 | 0.50 | 16.87 | 14.97 | 11.96 | 52.38 |
| | ORLM-LlaMa3-8B | 1.87 | 1.28 | 18.92 | 100.00 | 35.71 | 2.25 | 8.44 | 62.05 | 32.74 | 52.17 | 73.33 |
| **Bare LLMs (Direct Prompting)** | DeepSeek-V3.2-685B | 5.61 | 1.10 | 1.80 | 11.11 | 9.52 | 7.87 | 2.48 | 20.48 | 7.87 | 15.22 | 41.62 |
| | DeepSeek-V3.2-Think-685B | 7.48 | 2.57 | 3.60 | 5.56 | 14.29 | 1.12 | 3.97 | 42.17 | 9.14 | 6.52 | 42.20 |
| | Qwen3-Max (No-Thinking) | 1.87 | 2.02 | 3.60 | 11.11 | 9.52 | 1.12 | 0.50 | 12.65 | 7.36 | 10.87 | 35.84 |
| | Qwen3-Max-Preview (Thinking) | 1.87 | 12.11 | 40.54 | 22.22 | 23.81 | 2.25 | 0.99 | 21.08 | 4.31 | 5.43 | 38.15 |
| | Claude-Sonnet-4.5 (No-Thinking) | 0.93 | 0.00 | 0.00 | 0.00 | 7.14 | 1.12 | 0.74 | 16.27 | 7.61 | 6.52 | 36.42 |
| | Claude-Sonnet-4.5 (Thinking) | 0.93 | 0.00 | 0.00 | 0.00 | 4.76 | 1.12 | 0.74 | 16.27 | 8.12 | 5.43 | 35.26 |
| | Gemini-3-Pro-Preview | 1.40 | 0.18 | 0.90 | 5.56 | 11.90 | 0.00 | 0.74 | 6.02 | 5.84 | 14.13 | 18.50 |
| | GPT-5.1 | 0.47 | 2.75 | 8.11 | 11.11 | 4.76 | 0.56 | 1.74 | 11.45 | 5.58 | 10.87 | 34.68 |
| | GPT-5.1-CodeX | 0.00 | 0.37 | 0.90 | 22.22 | 4.76 | 0.00 | 0.00 | 6.02 | 8.63 | 13.04 | 25.43 |
| | Avg | 2.17 | 1.76 | 6.37 | 25.40 | 14.28 | 1.73 | 2.54 | 22.20 | 12.67 | 17.47 | 47.57 |

**(i) Strong performance on existing benchmarks does not transfer to MIPLIB-NL.** On established NL-to-Opt benchmarks, most recent systems achieve moderate-to-high Pass@1 accuracy on average (Table 1, last row), including NLP4LP (90.97%), MAMO-E (91.78%), and RESOCRATIC (83.63%). In contrast, MIPLIB-NL is substantially more challenging: the average Pass@1 accuracy drops to 17.85%. This gap is also evident at the model level. For example, strong general-purpose LLMs that score 70–95% on many prior benchmarks achieve only 20–40% Pass@1 on MIPLIB-NL (e.g., Gemini-3-Pro-Preview at 36.19%, GPT-5.1 at 39.05%, DeepSeek-V3.2-685B at 23.81%). These results indicate that prior evaluations, dominated by small or weakly structured instances, substantially overestimate current NL-to-Opt capability.

**(ii) Scaling and specialization provide limited gains on MIPLIB-NL.** Specialized training and larger model scales improve performance on several benchmarks (Table 1). For instance, SIRL-QWEN2.5-32B performs strongly on MAMO-C (87.40%) and INDUSTRYOR (61.90%), while OPTMATH-QWEN2.5-32B consistently outperforms its 7B counterpart across multiple datasets (e.g., BENCH4OPT: 61.36% vs. 48.30%). However, these gains do not close the gap on MIPLIB-NL. Among fine-tuned systems, Pass@1 accuracy remains below 6% in most cases, indicating that industrial-scale structure, indexing, and long-range consistency pose challenges that are not addressed by existing supervision or fine-tuning distributions.

**(iii) Failures on MIPLIB-NL are dominated by execution-level breakdowns.** Table 9 reports the overall execution-level failure rate for $N=1$, defined as the fraction of cases that fail to produce a valid solver outcome. Averaged across methods, execution failures are rare on many benchmarks (e.g., NLP4LP: 1.73%, NL4OPT: 2.17%, RESOCRATIC: 2.54%), but rise sharply on MIPLIB-NL to 47.57%. This pattern is consistent across model families: GPT-5.1-CodeX (25.43%), GPT-5.1 (34.68%), Claude-Sonnet-4.5 variants (35–36%), DeepSeek-V3.2 models (41–42%), and Qwen3 variants (35–38%). Notably, some fine-tuned systems exhibit even higher failure rates on MIPLIB-NL (e.g., OPTMATH-QWEN2.5-7B at 90.48%), suggesting that specialization can be brittle when exposed to large, heterogeneous industrial instances. These execution-level failures largely explain the collapse of Pass@1 accuracy on MIPLIB-NL, as many attempts fail before semantic correctness can be meaningfully assessed.

**(iv) Pass@8 improves absolute accuracy but does not eliminate the gap.** Table 8 shows that allowing multiple samples improves performance across all datasets. Nevertheless, MIPLIB-NL remains the most challenging regime: the average

Pass@8 accuracy reaches only 21.87%, compared to 49–97% on other benchmarks. This suggests that naive resampling offers limited recoverability when models struggle with global structure, indexing, and constraint consistency.

**(v) Additional fine-tuning check.** Beyond pure evaluation, we also test whether difficult in-domain supervision helps smaller models. We conduct a leakage-controlled family-held-out LoRA experiment on Qwen2.5-7B-Instruct, using a split with 179 training instances and 44 test instances and zero family overlap (Table 10).

*Table 10.* Family-held-out LoRA experiment on Qwen2.5-7B-Instruct. **Base-7B** is the original Qwen2.5-7B-Instruct model; **LoRA-7B** is the same model after LoRA fine-tuning on the train-side in-domain supervision set; **GPT-5.1** is the best single frontier-model result on the same held-out test set; and **Union** counts instances solved by at least one available frontier-model run. The split contains 179 training instances and 44 test instances with zero family overlap.

| Metric | Base-7B | LoRA-7B | GPT-5.1 | Union |
|---|---|---|---|---|
| Executability | 1/44 | 14/44 | 31/44 | 36/44 |
| Correctness | 0/44 | 5/44 | 20/44 | 22/44 |

LoRA fine-tuning substantially improves both executability and end-to-end correctness over the base 7B model, showing that MIPLIB-NL can provide useful supervision for training better modeling agents. However, a large gap to frontier models remains, suggesting that model scale and higher-level modeling ability are still important for industrial-scale optimization modeling.

**Takeaway.** Together, Tables 1, 8, 9, and 10 demonstrate that MIPLIB-NL introduces a qualitatively different difficulty profile: substantially lower success rates, sharply higher execution-level failure probabilities, limited gains from increased sampling, and a persistent gap between small fine-tuned models and frontier models. These results support our central claim that realistic NL-to-Opt evaluation must stress large-scale, structured industrial models, where correctness depends on recovering indexed variable groups, repeated constraint families, and globally consistent data bindings rather than isolated local semantics.

## I. Coverage of MIPLIB 2017 Instances

MIPLIB 2017 comprises two components: a curated *Benchmark* set with 240 instances and a broader *Collection* of 1,065 real-world models, covering a wide range of difficulty, scale, and modeling styles. In principle, applying structure-aware reverse generation to the entire repository would be desirable. In practice, however, translating MIPLIB instances into high-quality natural-language-to-optimization specifications requires substantial expert effort and is not uniformly feasible across all models.

We therefore focus on a subset of 223 instances that can be reliably translated under our validation criteria. The remaining instances fall into several categories that currently pose significant challenges:

**Time and human resource constraints.** This is a primary factor limiting MIPLIB-NL to 223 instances. Although LLMs possess strong capabilities, the raw input for this task consists of MPS files. These files often exceed the context window of current models, or otherwise saturate the attention mechanism with fragmented, low-level data, thereby impairing task performance. While agentic workflows can mitigate context limitations to some extent as successfully reverse-engineering and validating medium-scale instances with simple forms, they still necessitate intensive human-in-the-loop interaction to ensure accuracy and prevent hallucinations. Purely manual reverse engineering, conversely, is prohibitively time-consuming. Future work will focus on automating the verification and reverse-generation pipeline to maximize dataset scale, retaining human experts only for critical decision-making on complex problems. However, we estimate that even with abundant time and manpower, only approximately 40% of the repository (400–500 instances) could be feasibly processed due to the intrinsic limitations described below. The main bottleneck is verification rather than generation. LLMs can already help with candidate scaffold discovery, metadata aggregation, script writing, and equivalence checks, and they can generate many plausible descriptions once a structure is fixed. What remains difficult is guaranteeing that a recovered scaffold, data binding, and NL description are exactly faithful to the original industrial formulation. We therefore view the next scalable path as verification-first semi-automation, with human experts reserved for final structural decisions on difficult or ambiguous cases.

**Solver-oriented reformulation and presolving.** MPS files are designed to facilitate efficient solving rather than human readability. They prioritize mathematical equivalence (or equivalence under transformation, such as translation or negation) over semantic clarity. Furthermore, MPS files frequently embody aggressive preprocessing and data transformations. While a presolved MPS file is advantageous for a solver, it presents a significant obstacle for reverse engineering natural language descriptions. Consider the problem `ex9`, which represents a $9 \times 9$ tiling puzzle restoration. Its variables are named $b_{i,j,k,l}$, ostensibly representing the placement of tile $i$ at position $(j, k)$ with rotation $l$. However, the MPS formulation incorporates high-level pruning: certain tiles are algebraically restricted to specific regions (edges, corners, or centers) based on their shape, and specific rotations are eliminated a priori. Consequently, the variable count is significantly lower than the expected $81 \times 81 \times 4$, and the constraint matrix becomes a tangled collection of conditional logic to account for these "holes" in the state space. This is evidenced by the fact that the Gurobi solver processes the presolved MPS file significantly faster than it solves a model generated from a clean, high-level description of the puzzle. For our task, reconstructing the original semantic intent from such sparse, irregular, and heavily presolved structures is exceptionally difficult.

**Absence of semantic variable names.** While MPS files store variable names, a significant portion utilize generic indexing (e.g., $x_1, x_2, \ldots, x_n$) rather than meaningful descriptors. This lack of semantic information accounts for over 50% of the instances we could not reverse engineer. Our methodology relies heavily on variable naming patterns to group variables, which in turn allows for the grouping of constraints and the identification of loop structures (compressing massive data into compact natural language). While constraint names and matrix sparsity patterns (spy plots) provide some cues, they are often insufficient without the context of variable names. Using `ex9` as an example again: without the variable names $b_{i,j,k,l}$, the problem appears merely as a set of Boolean implications ("if generic binary var A is 1, then generic binary vars B, C, or D must be 0"). It is nearly impossible to infer that this structure arises from physical puzzle piece matching. When combined with the presolving issues mentioned above, the spy plot appears as an irregular, "potholed" pattern with no discernible logic. For problems with complex relations or high-dimensional indices (like the 4-index `ex9`), the absence of variable names renders the instance effectively indecipherable.

**Extreme scale and structural complexity.** Even when variable names are preserved, certain instances remain difficult to reverse engineer due to their sheer scale and the heterogeneity of their constraints. Many complex scheduling, production planning, and timetabling problems fall into this category. For example, `comp07-2idx-hard` is a course timetabling problem involving capacity thresholds and complex temporal dependencies between daily slots. Recovering a coherent natural language description required significant effort to disentangle the constraint file. Similarly, for instances like `brazil3` and `highschool1-aigio`, human experts eventually abandoned the reverse-engineering process after initial inspection. While we believe semantic reconstruction is theoretically possible for these cases, the cognitive load and effort required to map the entangled algebraic structure back to a human-readable blueprint were deemed too high for the current iteration of the dataset.

**Lack of real-world interpretability.** Some instances can be technically reconstructed but were excluded from MIPLIB-NL because they represent mathematical relaxations or variations that lack self-consistent real-world semantics. In these cases, we cannot justify the "physical reality" of the reversed problem (analogous to describing a "Platform $9\frac{3}{4}$" in a strictly realistic context). For instance, `dano3_3` is a partial continuous relaxation of a binary transportation problem. The ground truth in the MPS file implies that a specific road is connected with a value of $0.87$. While mathematically valid for the relaxation, translating this into a natural language context results in semantic absurdity. Consequently, such instances were omitted to maintain the linguistic and logical quality of the dataset.

**Verification challenges with infeasible and open instances.** Although MIPLIB-NL includes a small number of infeasible and open instances, they are almost exclusively members of problem families that also contain tractable or easily verifiable counterparts. This proximity allows us to confirm the correctness of the natural language description and the resulting model via cross-validation with the solved instances. However, when an infeasible or open instance appears as an isolated case, or when an entire problem class is comprised solely of such instances, independent verification becomes intractable. Furthermore, open problems are inherently ill-suited for a ground-truth dataset; without a known optimal solution, it is impossible to convincingly validate that the code generated from the reversed description yields the correct result.

Importantly, exclusion from MIPLIB-NL does not indicate that an instance is unimportant or uninteresting. Rather, it reflects current limitations of structure recovery and validation at extreme scale or under heavy solver-oriented obfuscation. We view extending coverage to a larger fraction of MIPLIB 2017 as an important direction for future work as tools and

methodologies improve.

Although the current release focuses on mixed-integer linear programming (MILP) in order to validate the pipeline under a solver-grounded setting, the methodology is not tied to MILP alone. The same reverse-construction principle can be extended to other optimization classes and public model libraries, such as CBLIB for conic optimization instances including second-order cone programming (SOCP) and semidefinite programming (SDP), MINLPLib for mixed-integer nonlinear and nonlinear programming (MINLP/NLP), XCSP3 for constraint programming (CP), and multi-objective optimization collections. For multi-objective optimization, the validation step can be adapted by checking reference behavior under multiple scalarizations or weighted-sum settings. Early trials suggest that nonlinear and multi-objective instances can often be translated into natural scenarios as well, such as risk-buffer rules for SOCP project selection, semidefinite stiffness constraints for structural design, or multi-metric configuration selection for encoder tuning.

# J. Prompt Templates

In this section, we present the prompt templates used for generating optimization models with Gurobi. We employ four distinct data formats: Bench4Opt, MIPLIB-NL, ComplexOR, and OptMATH. While Bench4Opt, MIPLIB-NL, and ComplexOR each utilize a dedicated prompt template tailored to their specific structure, all other datasets follow the unified OptMATH prompt template. These templates ensure that LLMs receive precise instructions regarding the problem description, data format, and required output structure.

---

**OptMATH Prompt Template**

You are a professional mathematical optimization expert. Please use Python and gurobi to solve the following optimization problem.
# Question: [Problem Description]
# Note: - The Code must include:

```
import gurobipy as gp
from gurobipy import GRB
```

- Make sure the model variable is named 'model'.
- Avoid using $<$ and $>$ in Gurobi constraints; instead, use $\leq$ or $\geq$ as appropriate.
- Carefully determine whether the variable is an integer or a continuous variable.

---

**Bench4Opt Prompt Template**

You are a professional mathematical optimization expert. Please use Python and gurobi to solve the following optimization problem from the BENCH4OPT dataset.
# Problem Description: [Problem Description]
# Data File Path: [Full Data Path] # Data Preview: [Data Preview Content]
## Requirements:
1. Formulate the mathematical model and write Python code to solve it.
2. Use gurobi as the solver.
3. Read data from the JSON file using the exact path provided above.
4. Print the results in the specified format.
5. The code must be self-contained and runnable, with complete error handling.
## Mathematical Model
- **Key Parameters**: [numerical values from JSON data]
- **Decision Variables**: [description and types with justification]
- **Objective Function**: [mathematical expression]
- **Constraint Set**: [all constraints in math form]
- **Problem Classification**: [LP/IP/MIP]
# Output Format Provide your response in this EXACT structure:
## Mathematical Model - **Key Parameters Definition**
- **Decision Variables**: [description and types with justification]

---

- **Objective Function**: [mathematical expression]
- **Constraint Set**: [all constraints in math form]
- **Problem Classification**: [LP/IP/MIP]
## Python Implementation

```python
import gurobipy as gp
from gurobipy import GRB
import json

# Create model
model = gp.Model("optimization_problem")

# Data Loading from JSON file
with open(r'[Full Data Path]', 'r') as f:
    data = json.load(f)

# Decision variables
# [Define variables based on mathematical model with proper types]
# REMEMBER: Use INTEGER variables (GRB.INTEGER) when modeling countable
quantities

# Use CONTINUOUS (GRB.CONTINUOUS) only for inherently fractional quantities
like time/weight

# Objective function
# [Set objective based on mathematical model]

# Constraints
# [Add constraints based on mathematical model]

# Solve
model.optimize()

# Extract results
if model.Status == GRB.OPTIMAL:
    print(f"optimal_value = {model.ObjVal}")
else:
    print(f"Optimization status: {model.Status}")
    if model.Status == GRB.INFEASIBLE:
        print("optimal_value = INFEASIBLE")
    elif model.Status == GRB.UNBOUNDED:
        print("optimal_value = UNBOUNDED")
    else:
        print("optimal_value = ERROR")
```

*Note*: Avoid including executable code blocks like `if __name__ == '__main__':` in your response. Write your code directly without wrapping it in functions. # Response: (Give your response here)

---

## MIPLIB-NL Prompt Template

You are a professional mathematical optimization expert. Please use Python and gurobi to solve the following optimization problem.
# Problem Description: [Problem Description]

# Parameters: [Parameters JSON]
# Data Description: [Data Files Info]
## IMPORTANT: File Paths **Base Path for Data Files**: [Working Directory]
**CRITICAL INSTRUCTIONS FOR FILE READING:**
1. All data files are located relative to the base path specified above
2. When reading CSV files, use the COMPLETE RELATIVE PATH from base path
3. Example: If base_path is '/path/to/benchmark' and file is './data/station.csv',Use: `pd.read_csv('/path/to/benchmark/data/station.csv')` # NOT '.data/station.csv' or just 'station.csv'
4. DO NOT use absolute paths or relative paths with './'
Requirements:
1. Formulate the mathematical model and write Python code to solve it.
2. Use gurobi as the solver.
3. Read data files using the CORRECT relative paths as specified above.
4. print format: {"objective_value": float, "variables": dict, "status": str}
# Output Format Provide your response in this EXACT structure: ## Python Implementation

```python
import gurobipy as gp
from gurobipy import GRB

# Create model
model = gp.Model("optimization_problem")

# Data Loading,reading data from data files

# Decision variables
# [Define variables based on mathematical model with proper types]
# REMEMBER: Use INTEGER variables (GRB.INTEGER) when modeling countable
quantities
# Use CONTINUOUS (GRB.CONTINUOUS) only for inherently fractional quantities
like time/weight

# Objective function
# [Set objective based on mathematical model]

# Constraints
# [Add constraints based on mathematical model]

# Solve
model.optimize()

# Extract results(Must print the optimal_value)
if model.Status == GRB.OPTIMAL:
    print(f"optimal_value = {model.ObjVal}")
    print(f"Just print the best solution: {model.objVal}")

else:
    print(f"Optimization status: {model.Status}")
    if model.Status == GRB.INFEASIBLE:
        print("optimal_value = INFEASIBLE")
    elif model.Status == GRB.UNBOUNDED:
        print("optimal_value = UNBOUNDED")
    else:
```

```
        print("optimal_value = ERROR")
```

Note: Avoid including executable code blocks like `if __name__ == '__main__':` in your response. Write your code directly without wrapping it in functions. # Response: (Give your response here)

---

## ComplexOR Prompt Template

You are a professional mathematical optimization expert. Please use Python and gurobi to solve the following optimization problem from the ComplexOR dataset.
# Problem Description: [Problem Description]
# Problem Data Path: [Data Path]
# Code Example Template:

`[Code Example]`

## Requirements:
1. Complete the provided code example to solve the optimization problem.
2. Use gurobi as the solver.
3. Use the sample input data to test your implementation.
4. Print the optimal objective value in the format: `optimal_value = <value>`.
5. The code must be self-contained and runnable, with complete error handling.

---

[1]Great Bay University [2]Peking University [3]Huawei Technologies Co., Ltd [4]**AUTHORERR: Missing \icmlaffiliation.** . Correspondence to: Zaiwen Wen <wenzw@pku.edu.cn>.

*Proceedings of the 43$^{rd}$ International Conference on Machine Learning*, Seoul, South Korea. PMLR 306, 2026. Copyright 2026 by the author(s).

