# OpenReview forum: "Constructing Industrial-Scale Optimization Modeling Benchmark"
_ICML.cc/2026/Conference — ICML 2026 regular_

### Official Review · Reviewer_Djin · 2026-03-13

**Soundness:** 3
**Presentation:** 3
**Significance:** 3
**Originality:** 3
**Overall Recommendation:** 4
**Confidence:** 4

**Summary:**

This paper introduces MIPLIB-NL, a benchmark for automatic optimization modeling. It is reverse-engineered from the MIPLIB 2017 dataset and contains 223 instances. Compared with existing benchmarks, its instances are substantially larger and more challenging. Experiments show that the performance of current methods drops sharply on this benchmark, especially for open-source methods relying on task-specific fine-tuning. In contrast, strong closed-source and open-source foundation models still retain a certain level of accuracy.

**Compliance With Llm Reviewing Policy:**

Affirmed.

**Final Justification:**

The rebuttal addressed my previous concerns, I will keep my positive score.

**Key Questions For Authors:**

1. Could the authors discuss whether and how the construction process could be extended or automated in future work?
2. Would it be possible to split the 223 instances into fine-tuning and test subsets to examine whether limited fine-tuning on challenging examples could improve the performance of smaller models?
3. Could the authors provide more details on the validation process — specifically, how many instances passed the primary validation stage and how many passed the secondary validation — so as to better assess the contribution of each construction step?

**Limitations:**

Yes

**Strengths And Weaknesses:**

Strengths

1. The benchmark is well-motivated. It directly addresses a notable gap in the optimization modeling literature, and its difficulty is clearly and substantially higher than that of existing benchmarks.
2. The dataset construction process is rigorous and reflects considerable human effort, including secondary validation procedures.
3. The experimental findings are interesting. In particular, the observation that strong foundation models outperform smaller task-specifically fine-tuned models is a noteworthy result.

Weaknesses

1. From a methodological standpoint, the proposed pipeline reads more as an engineering effort than a research contribution. Its novelty is fairly limited, and the construction process relies heavily on manual labor, which constrains generalizability. That said, given the scale and difficulty of the instances, the absence of full automation is understandable.

2. The analysis is somewhat shallow. For example, the experiments show that foundation models outperform fine-tuned models, but it remains unclear whether this is mainly because the fine-tuning datasets lack sufficiently difficult examples, or because of model scale itself, i.e., whether this advantage is attributable to "emergent capabilities" arising from greater model scale alone. This point could be investigated further. For instance, the 223 instances could be split into subsets for simple fine-tuning and testing, in order to examine whether even limited fine-tuning on such difficult data would enable smaller models to acquire some useful modeling capability.

3. The related-work coverage could be broadened. In particular, Table 2 summarizes existing datasets and benchmarks from prior work, several relevant studies are missing, such as OptiTrust and Step-Opt. These should be included for completeness.

---

> ### Author Rebuttal · Authors · 2026-03-30
>
> We thank the reviewer for the constructive feedback and for recognizing the value of a harder industrial-scale benchmark. We respond point-by-point below.
>
> **[Response to Weak Point 1 and Question 1: research contribution, novelty, and future automation]**
>
> We emphasize that this work is *data-centric* paper rather than an *algorithm-centric* paper. We view its contribution as methodological rather than engineering-only. Concretely, the paper contributes: (i) a structure-aware reverse-construction methodology for turning solver-level industrial formulations into NL-to-Opt benchmark instances; (ii) a benchmark grounded in real MIPLIB models rather than toy or synthetic formulations; and (iii) empirical evidence that methods which appear strong on prior benchmarks degrade sharply once evaluated on industrial-scale benchmarks. In this sense, the scientific contribution is data-centric: the paper identifies a missing evaluation regime, proposes a methodology to construct it, and shows that current systems behave very differently under this regime.
>
> We acknowledge that the current workflow is not fully automated. As discussed in our responses to **Reviewer nX6S [Weak Point 1]** and **Reviewer joVh [Weak Point 1 and Question 1]**, the limiting factor here is verification rather than generation. For industrial MPS files, subtle mistakes in recovered loop structure, variable roles, or data alignment can silently invalidate an instance, which is why the current release still relies on expert-guided construction and validation. At the same time, we do not view the future path as purely manual: LLMs are already increasingly useful for candidate scaffold discovery, metadata aggregation, and equivalence checking. We therefore see semi-automation, rather than unconstrained full automation, as the most realistic next step, with human experts remaining responsible for final structural decisions on difficult cases.
>
>
> **[Response to Weak Point 2 and Question 2: foundation models vs. fine-tuned models]**
>
> To test whether smaller (fine-tuned) models lag behind mainly because they were not exposed to sufficiently difficult in-domain training examples, we conducted an additional controlled LoRA experiment on Qwen2.5-7B-Instruct using a small curated train-side supervision set.
>
> *Setting:* Before any data curation, we fixed a family-held-out split of the 223 retained instances into 179 training and 44 test instances, with zero family overlap (to prevent label leakage). As a coarse sanity check, strong-LLM historical accuracy is similar on the two sides; we use this only as a rough difficulty proxy. The final LoRA training set contains 108 curated train-side samples covering 45 training families: 87 historically verified in-domain solutions and 21 additional cleaned train-side references. No test-side artifact was used during curation.
>
> *Results:* Evaluation is execution-based on the original held-out instances. The base 7B model solves `0/44` instances correctly and produces executable code on only `1/44`. After LoRA, this improves to `5/44` correct and `14/44` executable. Thus, limited difficult in-domain supervision clearly helps small models under a strict leakage-controlled setting, especially in executability. However, the residual gap remains substantial: on this same held-out test set, the strongest historical single-model result reaches `20/44` correct, and the reference upper bound from the available strong-model results is `22/44` correct.
>
> *Takeaway:* This additional experiment supports a mixed but clear conclusion: insufficient difficult in-domain supervision is an important part of the bottleneck, but it is unlikely to be the whole explanation. Even after adding carefully curated in-domain supervision, a substantial gap to frontier LLMs remains, suggesting that model scale and higher-level modeling ability are likely still important on this benchmark.
>
>
> **[Response to Weak Point 3: related-work coverage]**
>
> OptiTrust and Step-Opt are already included in Table 2  under Appendix A. We will also mention them explicitly in the main related-work discussion for completeness.
>
> **[Response to Question 3: primary vs. secondary validation details]**
>
> In our current construction logs, 245 candidate instances went through the full validation pipeline. Among them, 163 passed the primary validation stage, i.e., they were successfully recovered under independent NL-to-Opt reconstruction with Pass@8-style screening. The remaining 82 cases then entered the secondary stage, which involves expert-driven human-LLM validation and revision. Of these 82 cases, 60 were retained after secondary validation, while 22 were ultimately not retained. In addition, a substantially larger number of MIPLIB instances did not enter this full validation pipeline at all, for the reasons already summarized in Appendix I. We will add this breakdown explicitly in the revision so that readers can better assess the contribution of each validation stage.

---

> > ### Author Rebuttal · Reviewer_Djin · 2026-04-02
> >
> > Thank you for the detailed response. The additional LoRA fine-tuning experiment is a valuable addition and helps address my concern regarding analysis depth. I will keep my positive score.

---

### Official Review · Reviewer_nX6S · 2026-03-13

**Soundness:** 3
**Presentation:** 2
**Significance:** 3
**Originality:** 2
**Overall Recommendation:** 4
**Confidence:** 2

**Summary:**

The paper introduces a structure-aware reverse construction method, a dataset called MIPLIB-NL, and a comprehensive evaluation of SOTA LLM-based optimization modeing methods. MIPLIB-NL is reverse-constructed from MIPLIB 2017 through three stages: Stage 1, Structural abstraction, where experts recover indexed variable groups and repeated constraint families; Stage 2, Structure-aware Opt-to-NL generation, where variables, constraints, and so on are translated into NL descriptions; and Stage 3, Semantic validation, which involves validating via independent NL-to-Opt reconstructions and expert reviews. The authors conducted experiments on MIPLIB-NL, and one interesting observation is that the error mode can be divided into three categories, providing some practical insights to the community.

**Compliance With Llm Reviewing Policy:**

Affirmed.

**Final Justification:**

The answers are detailed and clear, and I'd like to keep my positive score.

**Key Questions For Authors:**

1. The expert-driven loop abstraction stage is quite labor-intensive and could be subjective. Are there any possible methods to automate this stage or reduce subjectivity?

**Limitations:**

See “Weaknesses”.

**Strengths And Weaknesses:**

Strengths:

1. The paper detects the evaluation limitations of NL-to-optimization systems, which is important to address.

2. The proposed solution is systematic and comprehensive, including a dataset, a dataset construction method, and an evaluation.

3. The experiments revealed three model failure modes, which provide practical guidance for the community.

Weaknesses:

1. Stage 1 of the proposed construction method relies on OR experts and is labor-intensive, which can be a significant bottleneck for its implementation in real-world scenarios.

2. The 223 instances used are all from MIPLIB, limiting the dataset to MIP and excluding datasets with non-linear or multi-objective problems.

---

> ### Author Rebuttal · Authors · 2026-03-30
>
> We thank the reviewer for the positive assessment and for highlighting the value of the dataset, the construction methodology, and the failure-mode analysis. We respond point-by-point below.
>
> **[Response to Weak Point 1: labor intensity, automation in Stage 1]**
>
> We thank the reviewer for highlighting the labor-intensive nature of Stage 1. While this was a significant bottleneck during our initial late-2025 experiments due to agent context limits and hallucinations in long-horizon tasks, recent advancements have drastically reduced this burden. In our March 2026 re-tests using Codex and GPT-5.4, agents autonomously handled instances with simple constraint structures and significantly reduced human effort on complex ones by automating tedious validation steps.
>
> However, human OR experts remain crucial for deep structural insights. For instance, in the puzzle problem (`ex9`, Appendix C), GPT-5.4 required human prompting to discover the underlying edge-matching logic rather than relying on a naive lookup table. Experts are still needed for quality control and guiding mathematical intuition.
>
> Importantly, an LLM's ability to reverse-engineer problems using ground-truth solutions for closed-loop correction does not translate directly to forward-modeling success, where no ground truth exists. Thus, even as pipeline automation improves, MIPLIB-NL remains a rigorous and essential benchmark for evaluating real-world modeling capabilities.
>
> **[Response to Weak Point 2: limitation to MIP / future extension to non-linear and multi-objective settings]**
>
> We acknowledge this limitation. However, we emphasize that our primary contributions are not limited to the released dataset *MIPLIB-NL*, and also include the reusable *reverse-construction methodology* (structural abstraction, model-data separation, and solver-grounded validation). Due to the labor-intensive nature of expert review and human-LLM interaction, we prioritized MIPs because they dominate industrial applications and provide a strong testbed for validating the methodology.
>
> Crucially, our pipeline is paradigm-agnostic. Preliminary investigations confirm its applicability to other repositories: CBLIB (3000+ SOCP and SDP instances), MINLPLib (1600+ MINLP and NLP instances), XCSP3 (23000+ CP instances), and MOOT (120+ MOO instances). For multi-objective optimization, solver-grounded validation can be adapted by verifying solutions across multiple weighted-sum combinations. Specifically, we have already successfully drafted reverse-generated NL descriptions for non-linear and MOO instances using our pipeline. For example, an SOCP project selection problem (`ck_n25_m10_o1_1`) naturally translates to "risk buffer rules involving standard deviation," an SDP framework design (`5x5_1bar`) translates to "global stiffness matrix semi-definiteness under multiple loads," and an `x264` encoder tuning problem abstracts to "multi-metric configuration selection."
>
> In summary, while the current benchmark focuses on MIPs to ensure high-quality, expert-validated data, extending this methodology to non-linear and multi-objective domains is an immediate direction for future work.
>
> **[Response to Question 1: reducing subjectivity in Stage 1]**
>
> We address the labor intensity and automation concerns in **[Response to Weak Point 1: labor intensity, automation in Stage 1]**. Regarding subjectivity, we'd like to emphasize that while the notation of loop abstraction may vary by expert, the mathematical semantics are objective, dictated entirely by the raw MIPLIB data. The abstraction is a necessary dimensionality reduction to prevent experts and LLMs from parsing MB/GB-sized raw files. For example, in the `wachplan` instance (Appendix C), 56 flat constraints are compressed into the loop equation:
>
> `2.0 <= sum_{1<=j<=8} x#{j}#{i} <= 3.0, for all 0<=i<=27.`
>
> The algebraic syntax might be subjective, but the underlying data structure is not.
>
> To automate and reduce subjectivity in this stage, we employ an asymmetric LLM verification approach. While generating perfect loops from flat files is hard for current LLMs, verifying them is easier. LLM agents can write scripts to expand the human-drafted loops and mathematically verify their exact equivalence against the original raw data, ensuring objective correctness. Besides, please see our response to **Reviewer joVh [Question 2]** regarding subjectivity in terms of problem background selection/wording choice.

---

> > ### Author Rebuttal · Reviewer_nX6S · 2026-04-03
> >
> > Thank you for the detailed response. The answers about how to deal with labor intensity and subjectivity are clear. I’ll keep my positive score.

---

### Official Review · Reviewer_tk3P · 2026-03-13

**Soundness:** 2
**Presentation:** 4
**Significance:** 1
**Originality:** 2
**Overall Recommendation:** 3
**Confidence:** 4

**Summary:**

This paper presents a new benchmark called MIPLIB-NL for evaluating the capability of LLMs to construct optimization models from natural language descriptions.
They created it using a reverse construction methodology consisting of three steps.
First, they aggregated a variety of individual constraints into constraint families with a loop-based structure.
Then, they reverse-engineered natural language descriptions based on this mathematical structure and contextual information, optionally applying LLM-based polishing.
Finally, they validated the correctness of the datasets by testing whether state-of-the-art LLMs could reconstruct the problem and generate solver codes that produce the correct solutions.

**Compliance With Llm Reviewing Policy:**

Affirmed.

**Final Justification:**

With the additional clarifications, I updated my final evaluation. There are still key concerns remaining regarding RQ3.

**Key Questions For Authors:**

- Can the main focus of this paper be clarified, and its scientific contributions verified and validated?

**Limitations:**

Yes

**Strengths And Weaknesses:**

- Their arguments are well-structured and well-supported by the experimental results. The overall flow of the paper is easy to understand and follow in the main body.

- In addition, although Appendix I distinguishes cases that prohibit NL instantiation, the 'application context identification' section on Page 5 states that natural language scenarios were assigned even to artificially generated MIPLIB data. Therefore, the distinction between these cases is not clearly defined, which could hinder reproducibility.

- The comparison with existing NL-to-Optimization benchmarks should become clearer and more concise. For example, in 'structural fidelity' of Section 2.2, not all benchmarks rely on flattened individual constraints; however, despite the existence of benchmarks constructed under loop-based constraint aggregation, the distinction between them remains ambiguous.

- This work represents the first attempt to reverse-engineer the MIPLIB dataset into a natural language benchmark, which holds significant value as it introduces a novel methodological approach. The need for an industry-level benchmark to evaluate the optimization formulation capabilities of LLMs has been widely recognized, and this paper effectively addresses this important problem.

- However, my main concern is that this paper does not contain enough science to justify its publication in a major conference proceedings. The overall reverse-engineering approach seems interesting, but lacks verification and validation. The research questions raised in Section 4.1 are neither the central theme of this paper nor answered in depth.

- Related to the above comment, it is quite vague what the main focus of this paper is. As a new benchmark dataset, it will make an excellent resource for researchers in the field. As a standalone research work, its significance and scientific value are unclear.

- Since it relies exclusively on MIPLIB 2017, the dataset is limited to Mixed-Integer Linear Programming problems. It would be highly beneficial for future work to explore methodologies that can encompass a broader range of optimization problems, such as Second-Order Cone Programming or Semi-Definite Programming.

---

> ### Author Rebuttal · Authors · 2026-03-30
>
> We thank the reviewer for the careful reading and the constructive concerns. We respond point-by-point below.
>
> **[Response to Weak Point 1: application-context identification vs. Appendix I exclusions]**
>
> The distinction for assigning NL descriptions is based on *structural clarity* rather than whether the data is real or synthetic.
>
> In MIPLIB, “artificially generated” usually means random parameters applied to clear combinatorial structures. For these instances (roughly 20 in our dataset), we can still map the underlying math to tight NL scenarios. For example, the synthetic `cvs` family (hypergraph vertex separator) aligns naturally with a real-world “workstation assignment” scenario. For generic random IPs such as `gen-ip`, a generic “resource allocation” NL is mathematically faithful.
>
> In contrast, Appendix I excludes instances whose structural semantics are obfuscated, even if they have real-world tags. For example, `buildingenergy` has a genuine industrial background, but its formulation is too complex or mathematically transformed to reliably recover the original business logic. We therefore chose not to force generic “maximize profit subject to constraints” templates onto such cases.
>
> We will make this clearer in the revised version.
>
> **[Response to Weak Point 2: comparison with existing benchmarks]**
>
> Please see our **[Additional comparison with prior benchmarks]** in the response to **Reviewer joVh** for the full audit and external figure. In brief, some prior benchmarks do preserve loop-based structure, so the distinction should not be framed as a binary “loop vs. no loop” claim. Our intended point is that richer NL-preserved structural complexity is rare in existing benchmarks and, when present, usually appears in much smaller numbers and at much smaller scales than in MIPLIB-NL.
>
> **[Response to Weak Points 3 and 4, and the Question: main focus, scientific value, and verification/validation]**
>
> We thank the reviewer for raising this central concern. We respectfully disagree that the paper “does not contain enough science.” In modern ML/LLM research, benchmark construction is itself a scientific contribution when it identifies a missing capability dimension, proposes a principled methodology, and materially changes the empirical picture of current systems. This data-centric paradigm is already established in top venues. In optimization modeling, examples include OptMATH (ICML 2025), ReSocratic (ICLR 2025), EOR (ICLR 2025), and ORQA (AAAI 2025). In broader mathematical reasoning, examples include MiniF2F (ICLR 2022), PutnamBench (NeurIPS 2024), OlympiadBench (ACL 2024), Omni-MATH (ICLR 2025), FATE (ICLR 2026), MATH-Perturb (ICML 2025), RBench (ICML 2025), EMMA (ICML 2025), and RE-IMAGINE (ICML 2025). These works have shaped their fields precisely because these "data-centric" papers can reveal gaps that algorithm papers alone do not expose. Our paper follows this "data-centric" paradigm. Our main focus is not only to release a dataset, but also to present a methodology for constructing realistic NL-to-Opt benchmarks from industrial-scale problems, and to expose limitations of current NL-to-Opt systems that are invisible on prior benchmarks.
>
> Regarding “verification and validation,” we respectfully disagree that the pipeline lacks them. Our workflow is not a free-form generate-and-trust process: reverse construction is anchored to the original solver-level formulation, with recovered variable groups, constraint families, and model--data bindings checked against the raw MPS structure and coefficients. The released instances are also made auditable through an explicit model--data separation format, and Section 3.3 adds an explicit semantic validation stage consisting of *independent NL-to-Opt reconstruction* and *expert-driven human--LLM interaction*. Cases whose semantics cannot be recovered with sufficient confidence are excluded rather than forced into the benchmark (Appendix I). We will revise the paper to make this logic clearer.
>
> Regarding Section 4.1, we also respectfully disagree that these research questions are “neither the central theme nor answered in depth.” In benchmark papers, targeted evaluation questions are a standard way to show what the new benchmark reveals about current systems; this is part of validating the contribution, not separate from it. In our case, these questions test whether preserving industrial structure changes the evaluation picture, which is one of the paper’s central claims. Thus, the Section 4.1 experiments are not the whole paper, but they are a major component showing why this benchmark matters and what limitations of current systems become visible under this regime.
>
>
> **[Response to Weak Point 5: limitation to MILP]**
>
> We acknowledge this limitation and refer the reviewer to our response to **Reviewer nX6S [Weak Point 2]**. While we validated the pipeline on MILP first, the reverse-construction methodology can be extended beyond MILP in future work.

---

> > ### Author Rebuttal · Reviewer_tk3P · 2026-04-04
> >
> > Thanks for the detailed explanation and claims. While my other comments and questions are reasonably addressed, I do have reservations about the research question parts and the main scientific contribution part.
> >
> > I agree that the targeted evaluations are important. I would suggest changing the wording of "Research Questions" to "Evaluation Criteria" or a similar term with a limited, specific scope.
> >
> > Those questions themselves could also be improved for the purpose of evaluation. Since earlier in the paper, the authors pointed out the current limitations and the new method to create a large-scale benchmark set, the evaluation criteria should focus on those factors, rather than the current broader questions. For example, the authors could consider the current RQ1 and RQ2 as the evaluation criteria, while making RQ3 a more central research question. The nature of RQ1-2 and RQ3 looks very different.
> >
> > If RQ3 deserves a central research question, which I believe it does, Figure 7 can be improved with other supporting evidence. Are there different patterns of error modes across the different scales? Are they statistically significant?
> >
> > Related to this question, my main concern still remains. I am still struggling to understand the main scientific contribution of this paper. I understand that the main motivation of the authors is the lack of an industry-scale benchmark set, which I agree with. But what new scientific findings does this new industry-scale benchmark set make?
> >
> > If the main point is that current LLM-based modeling is not yet ready for industry-scale problems, that conclusion is already well supported by prior benchmarks such as OptiMathBench, LogiOR, and other recent modeling studies.
> >
> > If the main point is that modeling error patterns differ significantly in industry-scale problems, and that current modeling research should therefore shift toward those factors, then the current analysis of error modes is not yet sufficient to support that claim.
> >
> > If the main goal of the authors is to help train a better modeling agent by providing an industry-scale training dataset, which is probably already in their roadmap, I fully support this approach. But, in its current form, the manuscript does not show how helpful this benchmark set is for such a task.
> >
> > Hopefully, the authors can clarify these points.

---

> > > ### Author Response · Authors · 2026-04-07
> > >
> > > We thank the reviewer for the thoughtful follow-up. In ML benchmark papers, benchmark-level questions are often written as research questions (RQs), so the issue is hierarchy rather than the label. We will make this explicit in revision: RQ1 and RQ2 mainly validate the new benchmark regime and establish the capability picture, while RQ3 is the main analysis question.
> > >
> > > More importantly, the paper’s contribution should not be reduced to any one of the three “main point” alternatives in the follow-up. The core contribution is data-centric: a solver-grounded methodology for constructing realistic NL-to-Opt benchmarks from industrial-scale optimization problems and creating an evaluation regime largely absent from prior work. RQ1 and RQ2 show that this regime changes the empirical picture; RQ3 analyzes which failures it reveals.
> > >
> > > So the claim is not simply that industrial-scale instances are harder. Rather, prior benchmarks underrepresent the structure-preserving industrial regime that materially changes the evaluation picture; this is also consistent with our broader audit of prior benchmarks ([d.png](https://anonymous.4open.science/r/MIPLIB-NL-CF6F/assets/d.png)).
> > >
> > > RQ3 is intended to support this point. Relative to the original Figure 7, which separates failures into execution, modeling, and timeout errors, (as suggested by the reviewer) we now extend the analysis to finer subtypes and summarize the results in two external figures: an across-benchmark comparison ([b.jpg](https://anonymous.4open.science/r/MIPLIB-NL-CF6F/assets/b.jpg)) and a within-benchmark relative-scale analysis ([r.png](https://anonymous.4open.science/r/MIPLIB-NL-CF6F/assets/r.png)). Modeling failures are split into Variable Domain / Type, Constraint / Global Logic, Data / Index Coupling, Objective / Bound / Coefficient, and Template / Incomplete Model; execution failures into Python / API, Data Binding, Solver Status, Environment / Resource, and Other Runtime.
> > >
> > > First, across benchmarks, MIPLIB-NL exhibits a qualitatively different error profile. Prior benchmarks are typically dominated by Variable Domain / Type or Constraint / Global Logic errors, whereas MIPLIB-NL is dominated by Data / Index Coupling Error (66.0%) and Template / Incomplete Model (26.4%). This is the kind of failure expected when models must preserve heterogeneous data relations, index mappings, and cross-entity coupling at industrial scale. The corresponding pairwise statistical tests remain significant  with the contrast being especially strong on the modeling side. So the point is not just that MIPLIB-NL is harder; it reveals a distinct cross-benchmark failure regime.
> > >
> > > Second, within benchmarks, the relative-scale analysis shows that failure composition also shifts systematically as instances become larger relative to others from the same benchmark. For execution errors, the dominant pattern moves from data-binding failures at smaller relative scales to environment/resource failures at larger relative scales. For modeling errors, Variable Domain / Type remains the largest subtype across all relative-scale buckets, but its share declines at the largest scale, while Data / Index Coupling and Template / Incomplete Model become more prevalent. These shifts are also statistically significant. So relative scale does not merely increase the failure rate; it changes the composition of failures.
> > >
> > > The benchmark is also useful beyond pure evaluation. As discussed in our response to **Reviewer Djin [Weak Point 2 and Question 2]**, we conducted a family-held-out LoRA fine-tuned experiment on Qwen2.5-7B-Instruct with a leakage-controlled split (179 train / 44 test, zero family overlap):
> > >
> > > | Metric | Base-7B| FineTuned-7B | GPT-5.1 | Union |
> > > | --- | ---: | ---: | ---: | ---: |
> > > | Executability | 1/44 | 14/44 | 31/44 | 36/44 |
> > > | Correctness | 0/44 | 5/44 | 20/44 | 22/44 |
> > >
> > > FineTuned-7B on a curated train-side in-domain supervision set substantially improves both executability and end-to-end correctness, showing that difficult in-domain supervision helps smaller models. However, a large gap to frontier models (e.g., GPT-5.1 at writing time) remains. So the benchmark is useful for training better modeling agents, while also indicating that model scale and higher-level modeling ability remain important.
> > >
> > > In short, the three alternatives in the follow-up are better understood as complementary consequences and uses of a new benchmark regime, not as mutually exclusive definitions of the paper’s contribution. The revision will make this hierarchy explicit while keeping the standard RQ terminology, and will strengthen the paper’s presentation of RQ3 with the deeper subtype-level analyses described above.  We hope this clarifies the points the reviewer raised.

---

### Official Review · Reviewer_joVh · 2026-03-13

**Soundness:** 4
**Presentation:** 4
**Significance:** 3
**Originality:** 3
**Overall Recommendation:** 5
**Confidence:** 5

**Summary:**

While current datasets mostly rely on toy or synthetic problems (only 10 or 20 variables),this benchmark pushes the boundary to the millions, that is a scale that finally reflects the messy reality of industrial solvers. It offers a solid baseline for testing what LLMs can actually achieve in real world optimization problems. The contribution in this work is building a bridge between natural language and actual solver code for industrial problems, going beyond the usual toy examples, to contribute the advancement of the field of automated modeling.

**Compliance With Llm Reviewing Policy:**

Affirmed.

**Final Justification:**

My questions have been fully resolved, and I will maintain my positive score.

**Key Questions For Authors:**

While this dataset is highly valuable, my main concern is its future scalability which is also mentioned in the part of weakness. I am curious if the authors have experimented with using the LLMs to generate new instances (in large scale)? Maybe the results are poor, but I want to know how powerful the LLMs are in this problem these days. Even a semi-automated pipeline would significantly boost the long-term viability of this work. Could we used some LLMs to significantly reduce the high costs of manual annotation? Addressing this would clarify the benchmark's long-term sustainability and impact.

I also have another question. Since some different optimization can be modeling in the same formula (e.g. if a box contains seven balls, taking four balls is equivalent to taking three balls) , the natural language or modeling of an instance maybe vary. How did you make choice? Do different choices result in different levels of difficulty for the automated modeling algorithm?

**Limitations:**

yes

**Strengths And Weaknesses:**

This paper presents a dataset that addresses a major bottleneck in the field. Aligning natural language with reference formulations is a solid and much-needed contribution to the community. However, this work is the reliance on manual curation. While this ensures high quality, it does raise questions about how the authors plan to keep the benchmark fresh as industrial needs evolve.

---

> ### Author Rebuttal · Authors · 2026-03-30
>
> We thank the reviewer for the positive evaluation and the thoughtful questions.
>
> **[Response to Weak Point 1 and Question 1: manual curation, long-term freshness, and scalability]**
>
> We appreciate the suggestion regarding LLM-driven scalability. We categorize LLM generation into two directions: (1) *contextual augmentation* (altering background, preserving math), which is detailed in our response to **[Question 2]**. This effectively enriches semantic diversity and can help keep the benchmark fresh, though SOTA models are often quite robust to such changes; (2) *structural* or *de novo generation* (creating new mathematical models). While LLMs can easily generate thousands of new problems, verifying their mathematical correctness without a reliable ground truth is currently an intractable bottleneck that breaks our validation pipeline. This is precisely why the current release still relies on expert curation: for a benchmark, correctness and auditability are more important than raw scale. Developing verifiable, automated LLM generation frameworks therefore remains a critical focus for our future work. In other words, we do see a realistic path toward semi-automated expansion, but we believe it must remain verification-first rather than generation-first.
>
> **[Response to Question 2: realization choice and difficulty sensitivity]**
>
> We agree that the same underlying optimization model may admit multiple natural-language realizations, and that these choices may affect the difficulty of automated modeling.
>
> In our dataset construction, once the structural scaffold of an instance was fixed, we used one canonical realization whose main goal was semantic faithfulness and verifiability, rather than selecting wording that would be especially easy for LLMs. To directly test the reviewer’s concern, we conducted an additional controlled study in which we kept the optimization model, data files, and reference optimal value unchanged, and only varied the natural-language realization. We tested 14 source instances from 4 families (`ab`, `air`, `beasleyC`, `graph`), each with three versions: the original text (`v0`), a same-domain paraphrase (`v1`), and a scenario-reskinned but semantically equivalent rewrite (`v2`).
>
> We evaluated 3 models (`qwen3-max`, `gpt-5.1`, `claude-sonnet-4-5`). Each cell below reports aggregated Pass@1 over all `#instances × 3 models` runs for that family. Thus,  `9/9` means 9 correct predictions out of `3 instances × 3 models`. Under this controlled setup, we observe only slight variation, confined to a small subset of families:
>
> | Family | # source instances | `v0` (original) | `v1` (paraphrase) | `v2` (reskinned) | Interpretation |
> |---|---:|---:|---:|---:|---|
> | `ab` | 5 | `0/15` | `0/15` | `0/15` | Highly stable |
> | `air` | 3 | `9/9` | `9/9` | `8/9` |Minor variation |
> | `beasleyC` | 3 | `6/9` | `6/9` | `6/9` |Highly stable |
> | `graph` | 3 | `0/9` | `2/9` | `0/9` |Minor variation |
>
> Thus, for several families (`air`, `beasleyC`), performance is quite stable across realizations, suggesting that our benchmark conclusions are not driven by a particular wording choice. At the same time, the `graph` family suggests that some abstract problem classes may exhibit limited sensitivity to semantic surface form in this small-scale study. We will clarify this point in the revision and note that robustness to equivalent realizations is an important direction for future benchmark extensions.
>
> Taken together, our answer is that LLMs are already useful for benchmark extension when the underlying mathematics is fixed and verifiable, but much less reliable for unconstrained creation of new benchmark instances. Likewise, our controlled study suggests that differences across equivalent realizations are limited and mostly confined to a small subset of problem families, so the main conclusions of our benchmark are unlikely to be driven by a particular wording choice.
>
> **[Additional comparison with prior benchmarks]**
>
> Following Reviewer **tk3P**’s suggestion, we audited prior NL-to-Opt benchmarks using four levels of **NL-preserved structure complexity**: `L0` = flat/enumerative NL with no explicit family structure; `L1` = simple indexed families; `L2` = coupled or multi-index families; `L3` = advanced temporal/routing/sequencing scaffolds. This measures how much structure is preserved in the NL itself; see https://anonymous.4open.science/r/MIPLIB-NL-CF6F/assets/d.png for a figure showing the `L0/L1/L2/L3` distribution for each benchmark.
>
> Under this criterion, some prior benchmarks do preserve loop-based structure, so our claim is not a binary “loop vs. no loop” distinction. However, the overall landscape remains dominated by `L0` (`92.1%`). These structurally complex cases are also scarce in absolute number: most benchmarks contribute only `0–13`. They are also much smaller in average `#vars + #cons` (typically tens to hundreds) than MIPLIB-NL ( with a median of `10^4` and upper tail of  `10^7`).

---

> > ### Author Rebuttal · Reviewer_joVh · 2026-04-03
> >
> > Thank you for the thorough rebuttal. My questions have been perfectly resolved, and I will happily maintain my score of "Accept".

---

### Decision · Program_Chairs · 2026-04-30

**Decision:**

Accept (regular)

**Comment:**

This submission is a benchmark/dataset paper that starts from MIPLIB 2017, and reverse engineers Natural Language descriptions of the models, as a new benchmark to evaluating NL-to-Opt pipelines. This new benchmark allows for evaluation for industry-scale optimization problems, as opposed to toy problems that currently dominate evaluation methodology in the literature. The paper demonstrates the use of the benchmark on some existing models.

Reviewers agree that such a benchmark is valuable, but note some concerns about the scalability of the benchmark construction approach, since it involves a lot of human intervention and manual curation. Another reviewer also questions the scientific value of the submission, but I (the AC) agree with the authors that benchmarks themselves can already be a good research contribution.

The authors are encouraged to incorporate rebuttal responses into a revision of the paper.